

# Nitrogen budgets following a Lagrangian strategy in the Western Tropical South Pacific Ocean: the prominent role of N₂ fixation (OUTPACE cruise).

Mathieu Caffin[1], Thierry Moutin[1], Rachel Ann Foster[2], Pascale Bouruet-Aubertot[3], Andrea Michelangelo Doglioli[1], Hugo Berthelot[1,4], Olivier Grosso[1], Sandra Helias-Nunige[1], Nathalie Leblond[5], Audrey Gimenez[1], Anne Alexandra Petrenko[1], Alain de Verneil[1] and Sophie Bonnet[6].

[1] Aix Marseille Université, CNRS, Université de Toulon, IRD, OSU Pythéas, Mediterranean Institute of Oceanography (MIO), UM 110, 13288, Marseille, France
[2] Stockholm University, Department of Ecology, Environment and Plant Sciences. Stockholm, Sweden
[3] Sorbonne Universités, UPMC Univ. Paris 06, LOCEAN, France
[4] Laboratoire des sciences de l'environnement marin, IUEM, Université de Brest-UMR 6539 CNRS/UBO/IRD/ Ifremer, Plouzané, France
[5] Observatoire Océanologique de Villefranche, Laboratoire d'Océanographie de Villefranche, UMR 7093, Villefranche-sur-mer, France
[6] Aix Marseille Université, CNRS, Université de Toulon, IRD, OSU Pythéas, Mediterranean Institute of Oceanography (MIO), UM 110, 98848, Nouméa, New Caledonia

*Correspondence to*: Mathieu Caffin (mathieu.caffin@mio.osupytheas.fr)

**Abstract.**

We performed N budgets at three stations in the western tropical South Pacific (WTSP) Ocean during austral summer conditions (Feb. Mar. 2015) and quantified all major N fluxes both entering the system (N₂ fixation, nitrate eddy diffusion, atmospheric deposition) and leaving the system (PN export). Thanks to a Lagrangian strategy, we sampled the same water mass for the entire duration of each long duration (5 days) station, allowing to consider only vertical exchanges. Two stations located at the western end of the transect (Melanesian archipelago (MA) waters, LD A and LD B) were oligotrophic and characterized by a deep chlorophyll maximum (DCM) located at $51 \pm 18$ m and $81 \pm 9$ m at LD A and LD B. Station LD C was characterized by a DCM located at $132 \pm 7$ m, representative of the ultra-oligotrophic waters of the South Pacific gyre (SPG water). N₂ fixation rates were extremely high at both LD A ($593 \pm 51$ µmol N m$^{-2}$ d$^{-1}$) and LD B ($706 \pm 302$ µmol N m$^{-2}$ d$^{-1}$), and the diazotroph community was dominated by *Trichodesmium*. N₂ fixation rates were lower ($59 \pm 16$ µmol N m$^{-2}$ d$^{-1}$) at LD C and the diazotroph community was dominated by unicellular N₂-fixing cyanobacteria (UCYN). At all stations, N₂ fixation was the major source of new N (> 90 %) before atmospheric deposition and upward nitrate fluxes induced by turbulence. N₂ fixation contributed circa 8-12 % of primary production in the MA region and 3 % in the SPG water and sustained nearly all new primary production at all stations. The e-ratio (e-ratio = PC export/PP) was maximum at LD A (9.7 %) and was higher than the e-ratio in most studied oligotrophic regions (~1 %), indicating a high efficiency of the WTSP to export carbon relative to primary production. The direct export of diazotrophs assessed by qPCR of the *nifH* gene in sediment traps represented up to 30.6 % of the PC export at LD A, while there contribution was 5 and < 0.1 % at LD B and



LD C, respectively. At the three studied stations, the sum of all N input to the photic layer exceeded the N output through organic matter export. This disequilibrium leading to N accumulation in the upper layer appears as a characteristic of the WTSP during the summer season, although the role of zooplankton in export fluxes should be further investigated.

## 1 Introduction

Biological nitrogen fixation, the reduction of atmospheric di-nitrogen ($N_2$) to ammonia is performed by a diverse group of prokaryotic organisms, commonly called diazotrophs. It provides the major external source of bio-available nitrogen (N) to the ocean, before riverine and atmospheric inputs (Deutsch et al., 2007; Gruber, 2008; Gruber and Sarmiento, 1997). In the oligotrophic ocean, N availability often limits phytoplankton growth (e.g. Moore et al., 2013) and $N_2$ fixation sustains a significant part of new primary production (PP) such as in the North (Karl et al., 1997) and South Pacific Ocean (Moutin et

al., 2008), the western Mediterranean Sea (Garcia et al., 2006), or the tropical North Atlantic (Capone et al., 2005), the new PP being the production unrelated to internal recycling of organic matter in the photic layer. New N input by $N_2$ fixation has thus been recognized as a significant process influencing global oceanic productivity, and can eventually fuel $CO_2$ sequestration through the $N_2$-primed prokaryotic carbon (C) pump (Karl et al., 2003).

Low $\delta^{15}N$ signatures of particles from sediment traps in the tropical North Pacific (Karl et al., 1997, 2012, Scharek et al.,

1999a, 1999b) and Atlantic (Altabet, 1988; Bourbonnais et al., 2009; Knapp et al., 2005; Mahaffey et al., 2003) suggest that at least part of the recently fixed N is ultimately exported out of the photic zone. Böttjer et al. (2017) revealed that $N_2$ fixation supports 26-47 % of particulate N (PN) export over a 9 year time series (2005-2013) period at station ALOHA (North Pacific Subtropical Gyre). Export efficiency may depend on the diazotroph community composition present in surface waters. Blooms of diatom-diazotroph associations (DDAs) systematically observed in late summer at station

ALOHA are thought to be directly responsible for the concomitant pulses of particulate export (Karl et al., 2012). High export associated with DDAs has also been observed in the Amazon River plume (Subramaniam et al., 2008) suggesting a high direct export efficiency associated with DDAs. *Trichodesmium* is one of the main contributors to global $N_2$ fixation (Mahaffey et al., 2005), but is rarely recovered in sediment traps (Chen et al., 2003; Walsby, 1992), suggesting a low direct export efficiency. However, the fixed $N_2$ by *Trichodesmium* is efficiently transferred to large non-diazotrophic

phytoplankton such as diatoms (Berthelot et al., 2016; Bonnet et al., 2016a), which can be subsequently exported (Nelson et al., 1995), suggesting a potential indirect export pathway. A recent mesocosm study performed in New Caledonia unexpectedly revealed that the production sustained by unicellular diazotrophic cyanobacteria (hereafter referred to as UCYN) was much more efficient at promoting particle export than the production sustained by DDAs (Berthelot et al., 2015). However, the export efficiency of UCYN is poorly studied (White et al., 2012) in the open ocean despite the fact that

they contribute as much as *Trichodesmium* to $N_2$ fixation rates in many parts of the ocean (Bonnet et al., 2009; Martínez-Pérez et al., 2016; Moisander et al., 2010; Montoya et al., 2004). More studies are thus needed to further investigate the ability of different diazotroph communities to fuel direct or indirect particle export in the oligotrophic ocean.



Studying the impact of $N_2$ fixation on PN export in the ocean and the relative role of each diazotroph group in this process are technically challenging. It requires the measurement of all major N fluxes both entering the system ($N_2$ fixation, nitrate ($NO_3^-$) eddy diffusion, atmospheric deposition) and leaving the system (PN export) with an adequate time frame under contrasting diazotroph communities' composition. Most importantly, such N budgets must be performed in the same water

mass to ensure that the particulate matter recovered in the sediment traps corresponds to the production that occurred just above in the photic layer. This is what we performed during the OUTPACE (Oligotrophy to UlTra-oligotrophy PACific Experiment) cruise in the Western Tropical South Pacific (WTSP) in summer 2015, during which we used a Lagrangian strategy (see below).

The WTSP Ocean has recently been identified as a hot spot of $N_2$ fixation, harbouring $N_2$ fixation rates >500 µmol N m$^{-2}$ d$^{-1}$

(Bonnet et al., 2017). The region covered by the OUTPACE cruise is characterized by trophic and $N_2$ fixation gradients (Moutin et al., 2017), with oligotrophic waters characterized by high $N_2$ fixation rates ($631 \pm 286$ µmol N m$^{-2}$ d$^{-1}$) mainly associated with *Trichodesmium* in the western part (i.e. within the hot spot around Melanesian archipelago waters, hereafter named MA), and ultra-oligotrophic waters characterized by low $N_2$ fixation rates ($85 \pm 79$ µmol N m$^{-2}$ d$^{-1}$) mainly associated with UCYN in the eastern part (South Pacific gyre, hereafter named SPG) (Bonnet et al., this issue; Stenegren et al., this

issue). This region therefore provides ideal conditions to study the potential role of $N_2$ fixation on particulate export under contrasting situations.

In the present study we focus on (i) the contribution of $N_2$ fixation to new N inputs in the WTSP during the summer season; (ii) the coupling between $N_2$ fixation and export; and (iii) the equilibrium versus disequilibrium between $N_2$ fixation and particulate N export in the WSTP.

**2 Material and methods**

**2.1 Station sampling strategy**

The OUTPACE cruise was carried out during austral summer conditions (strong thermal stratification, 18 Feb.-3 Apr. 2015) along a west-east 5500 km transect from New-Caledonia to French Polynesia. We performed a N budget at three stations hereafter named long duration (LD) stations that were chosen according to three criteria: (1) local minima of surface current

intensity, (2) different trophic regimes, i.e. oligotrophic vs. ultra-oligotrophic, and (3) different diazotrophs communities, i.e. *Trichodesmium* vs. UCYN.

To locate these three stations, we used a Lagrangian strategy developed during previous cruises such as LATEX (Doglioli et al., 2013; Petrenko et al., 2017) and KEOPS2 (d'Ovidio et al, 2015). Briefly, the regions of interest along the vessel route were firstly characterized at large scale through the analysis of satellite data. The altimetry-derived currents were processed

by SPASSO (Software Package for an Adaptive Satellite-based Sampling for Ocean campaigns; http://www.mio.univ-amu.fr/SPASSO/) to derive Eulerian and Lagrangian diagnostics of ocean circulation : Okubo-Weiss parameter, particle retention time and advection, Lagrangian Coherent Structures (d'Ovidio et al., 2015), together with maps of the sea surface

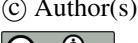



temperature and chlorophyll-a (Chl *a*) concentrations. The satellite data were processed on land in near real time and transmitted to the ship together with a daily bulletin proposing LD station positions (the complete series of 43 bulletins is available on the OUTPACE website at https://outpace.mio.univ-amu.fr/OUT_Figures/Bulletins/). We also performed onboard quantitative Polymerase Chain Reaction (qPCR) analyses on the *nif*H gene to measure the abundance of six groups

of diazotrophs (Stenengren et al., this issue). Thus, we located the stations in regions where either *Trichodesmium* or UCYN dominated the diazotroph community. Then, the exact locations of the three LD stations were determined on board in real time from a rapid survey using a Moving Vessel Profiler (MVP) equipped with conductivity-temperature-depth (CTD) and fluorimeter sensors, accompanied by the hull-mounted thermosalinograph and acoustic Doppler current profiler (de Verneil et al., this issue). Finally, Surface Velocity Program (SVP) drifters were deployed in order to study the relative dispersion at

the surface (de Verneil et al., this issue) during the station occupation.

By using this strategy, LD A station (19°12.8'S - 164°41.3'E, 25 Feb. - 2 Mar.) was positioned in MA waters in the western part of the transect (Figure 1) offshore New Caledonia. LD B station (18°14.4'S - 170°51.5'W, 15-20 Mar.) was positioned in MA waters near Niue Island and LD C station (18°25.2'S - 165°56.4'W, 23-28 Mar.) was positioned in the eastern part of the transect, in the SPG near the Cook Islands.

Each LD station was investigated for 5 days. The sequence of operations was the following: a drifting array equipped with three PPS5 sediment traps, current meters, oxygen sensors, and high frequency temperature sensors (see https://outpace.mio.univ-amu.fr/spip.php?article75 for details) was deployed at each station the first day. Then, a series of CTD (SBE 911+ SeaBird) casts (0-500 m) were performed every 3-4 h near the actual position of the drifting array to study the high frequency evolution of temperature, salinity, Photosynthetically Available Radiation (PAR) and fluorescence during

the station's occupation. Small-scale turbulence was characterized in the first 800 m from microstructure measurements using a Vertical Microstructure Profiler (VMP1000) that was typically deployed prior to or following each CTD cast (Bouruet-Aubertot et al, this issue). Nutrient concentration measurements (0-200 m) were performed everyday on the midday CTD casts (hereafter named 'nut. CTD'). In addition to the 0-500 m casts every 3-4 h, production casts (0-150 m) (hereafter named 'prod. CTD') were performed three times at each LD station (on day 1, 3 and 5) to quantify $N_2$ fixation and

primary production rates. Incubations with tracers ($^{14}C$ and $^{15}N_2$, see below) to quantify $N_2$ fixation and primary production were performed on an in situ drifting production line deployed for 24 h from sunrise to sunrise. The drifting array with the traps was recovered at the end of each LD station. The inputs of new N to the photic layer were induced by three different sources: atmospheric deposition at the air-sea interface, $N_2$ fixation as an interior source and $NO_3^-$ input by vertical diffusion. The N output was driven by PN sedimentation. The methods for the determination of each parameter are given below.



## 2. 2 Experimental procedures

### Physico-chemical parameters, nutrient concentrations and C:N ratios

In situ Chl $a$ concentration was derived from fluorescence measurements performed with a AquaTraka III (Chelsea Technologies Group Ltd) sensor. PAR was measured on each CTD profile. Phosphate ($PO_4^{3-}$) and $NO_3^-$ concentrations were

measured daily at 12 depths from the surface to 200 m on each nutrient CTD cast using standard colorimetric procedures (Aminot and Kérouel, 2007) on a AA3 AutoAnalyzer (Seal-Analytical). The quantification limits were 0.05 µmol $L^{-1}$ for $PO_4^{3-}$ and $NO_3^-$.

### Primary production and associated N uptake

PP was measured using the $^{14}C$ tracer method (Moutin and Raimbault, 2002). Samples were incubated on the in situ drifting

production line (Aquamout: https://outpace.mio.univ-amu.fr/spip.php?article75) for 24-h from dusk-to-dusk at 9 depths (75 %, 54 %, 36 %, 19 %, 10 %, 3 %, 1 %, 0.3 % and 0.1 % of surface irradiance levels), corresponding to the sub-surface (5 m) down to 105 m, 80 m and 180 m for stations LD A, LD B and LD C, respectively. A N-derived PP (N-PP) was obtained at each depth by dividing PP by the classical C:N Redfield ratio (6.625). Integrated N-PP (iN-PP) over the studied layer was calculated by using the trapezoidal method, assuming that surface N-PP was identical to N-PP measured in subsurface (5 m)

and considering that N-PP 20 m below the deepest sampled depth was zero (JGOFS, 1988). The theoretical percentage of production associated with regeneration in the photic layer was estimated as: 100 x (1 – (∑New N input / integrated N-PP)) as already proposed by Moutin and Prieur (2012).

### Atmospheric deposition

N atmospheric deposition ($NO_3^-$ and $NO_2^-$ (nitrite), hereafter called DIN) was quantified along the transect by aerosols

sampling as described in Guieu *et al.* (Submitted). Briefly, at each station, sample was collected on parallel filters: one filter was used to determine the total N mass per volume of air and another filter was used to perform dissolution experiments to measure the atmospheric DIN released in seawater that was then converted to atmospheric N inputs to the ocean.

### Nitrogen fixation rates

$N_2$ fixation rates ($\rho N_2$) were measured using the $^{15}N_2$ tracer method (Montoya et al. (1996), modified (see below)) on day 1, 3

and 5 at each LD station (hereafter named in situ_1, in situ_3 and in situ_5). Seawater was collected from the Niskin bottles in duplicate 4.5 L polycarbonate bottles at 9 depths (same depths as for NPP). 5 mL of $^{15}N_2$ gas (99 atom% $^{15}N$, Eurisotop) were injected in each bottle through the septum cap using a gas-tight syringe. All bottles were shaken 20 times to facilitate the $^{15}N_2$ dissolution and incubated in situ on the production line at the same depth of sampling for 24 h from dusk to dusk (hereafter called 'in situ incubation method'). It has been previously shown that the bubble method was potentially

underestimating $N_2$ fixation rates (Großkopf et al., 2012; Mohr et al., 2010) due to incomplete equilibration of the $^{15}N_2$ gas




with surrounding seawater compared to methods consisting in adding the $^{15}N_2$ as dissolved in a subset of seawater previously $N_2$ degassed (Mohr et al., 2010), whereas other studies did not find any difference between both methods (Bonnet et al., 2016b; Shiozaki et al., 2015). In the present study, we intentionally decided to use the bubble method both due to the high risk of organic matter and trace metal contamination during the $^{15}N_2$-enriched seawater preparation (Klawonn et al., 2015)

that has been seen to enhance $N_2$ fixation in this area (Benavides et al., 2017; Moisander et al., 2010). However, to avoid any possible rate underestimation due to equilibration of the $^{15}N_2$ gas with surrounding seawater, final $^{15}N$-enrichment in the $N_2$ pool was quantified on each profile in triplicates at 5 m and at the deep chlorophyll maximum (DCM). After incubation, 12 mL of each 4.5 L bottle were subsampled in Exetainers, fixed with $HgCl_2$ (final concentration 20 µg mL$^{-1}$) and stored upside down at 4°C in the dark until analyzed onshore within 6 months after the cruise according to Kana et al. (1994) using a

Membrane Inlet Mass Spectrometer.

For each LD station, in parallel with the last $N_2$ fixation profile (the in situ_5 profile), we performed a replicate $N_2$ fixation profile in which bottles were incubated in on-deck incubators equipped with circulating seawater at the specified irradiances using blue screening (hereafter called 'deck incubation method'). For these profiles, samples were collected in triplicates at 6 of the 9 depths reported above (75 %, 54 %, 36 %, 10 %, 1 % and 0.1 % surface irradiance level) in 2.3 L polycarbonate

bottles amended with 2.5 mL of $^{15}N_2$ gas (99 atom% $^{15}N$, Eurisotop) for 24 h.

In both cases, incubations were stopped by gentle filtration of the samples onto pre-combusted (450 °C, 4 h) Whatman GF/F filters (25 mm diameter, 0.7 µm nominal porosity). Filters were stored in pre-combusted glass tubes at -20 °C during the cruise, then dried at 60 °C for 24 h before analysis onshore. $^{15}N$-enrichments of PN collected on filters were determined using an Elemental Analyzer coupled to an Isotope Ratio Mass Spectrometer (EA-IRMS, Integra2 Sercon Ltd). The accuracy

of the EA-IRMS system was systematically controlled using International Atomic Energy Agency (IAEA) reference materials, AIEA-N-1 and IAEA-310A. In addition, the $^{15}N$-enrichement of the ambient (unlabeled) PN was measured at each station at the DCM and the subsurface and was used as the "initial" $^{15}N$-enrichment as termed in Montoya et al. (1996). Integrated $N_2$ fixation rates ($i$-$\rho N_2$) over the studied layer were calculated by using the same method as for the iN-PP.

**Nitrate diffusion**

$NO_3^-$ inputs from deep waters by turbulent mixing were estimated at the top of the nitracline. The top of the nitracline was found to fall along an isopycnal surface ($\rho=\rho_{NO3}$) whose density was determined at each station (Table 4). The $NO_3^-$ flux along the isopycnal surface $\rho=\rho_{NO3}$ was defined as:

$$Flux_{\rho_{NO3}}(t) = -Kz(z_{\rho_{NO3}}, t) \times \frac{d[NO_3]}{d\rho}(\rho_{NO3}, t) \times \frac{d\rho}{dz}(z_{\rho_{NO3}}, t), \qquad (1)$$

with Kz, the turbulent diffusion coefficient along the isopycnal $\rho=\rho_{NO3}$ inferred from VMP1000 measurements performed

every 3-4 h during the LD station occupation as described in Bouruet-Aubertot et al. (this issue) ; $\frac{d[NO_3]}{d\rho}$, the constant isopycnal slope of the nitracline, calculated for each station (Table 4), and $\frac{d\rho}{dz}$, the vertical density gradient measured by the



VMP1000 at the $\rho_{NO3}$ isopycnal depth $z_{\rho NO3}$. The time series of the $NO_3^-$ diffusive flux was calculated using an hourly temporal interpolation of Kz, daily averages and 5-day averages over the entire duration of each LD station were also computed.

**Particulate matter export**

Particulate matter export was quantified with three PPS5 sediment traps (1 m$^2$ surface collection, Technicap, France) deployed for 5 days at 150, 330 and 520 m at each LD station (Fig. 1). Particle export was recovered in polyethylene flasks screwed on a rotary disk which allowed automatically changing the flask every 24-h to obtain a daily material recovery. The flasks were previously filled with a buffered solution of formaldehyde (final conc. 2 %) and were stored at 4 °C after collection until analysis to prevent degradation of the collected material. The flask corresponding to the fifth day of sampling

on the rotary disk was not filled with formaldehyde in order to collect 'fresh particulate matter' for further diazotroph quantification as described below. Thus, this last flask was not used in the particulate export computations reported in Table 2. Onshore, swimmers were handpicked from each sample, quantified and genera identified. Exported particulate matter and swimmers were both weighted and analyzed separately on EA-IRMS (Integra2, Sercon Ltd) to quantify exported PC and PN. Particulate phosphorus (PP) was analyzed by colorimetric method (880 nm) after mineralization according to Pujo-Pay and

Raimbault (1994).

**Diazotroph abundance in the traps**

Triplicate aliquots of 2 to 4 mL from the flask dedicated to diazotroph quantification were filtered onto 0.2 µm Supor filters, flash frozen in liquid nitrogen and stored at -80 °C until analysis. Nucleic acids were extracted from the filters as described in Moisander et al., (2008) with a 30 seconds reduction in the agitation step in a Fast Prep cell disruptor (Thermo, Model

FP120; Qbiogene, Inc. Cedex, Frame) and an elution volume of 70 µl. Diazotroph abundance for *Trichodesmium* spp., UCYN-B, UCYN-A1, het-1, and het-2 were quantified by qPCR analyses on the *nif*H gene using previously described oligonucleotides and assays (Foster et al. 2007; Church et al. 2005). The qPCR was conducted in a StepOnePlus system (applied Biosystems, Life Technologies, Stockholm Sweden) with the following parameters: 50 °C for 2 min, 95 °C for 10 min, and 45 cycles of 95 °C for 15 s followed by 60 °C for 1 min. Gene copy numbers were calculated from the mean cycle

threshold (Ct) value of three replicates and the standard curve for the appropriate primer and probe set. For each primer and probe set, duplicate standard curves were made from 10-fold dilution series ranging from $10^8$ to 1 gene copies per reaction. The standard curves were made from linearized plasmids of the target *nif*H or from synthesized gBLocks gene fragments (IDT technologies, Cralville, Iowa USA). Regression analyses of the results (number of cycles=Ct) of the standard curves were analyzed in Excel. Two µl of 5 KDa filtered nuclease free water was used for the no template controls (NTCs). No

*nifH* copies were detected for any target in the NTC. In some samples only 1 or 2 of the 3 replicates produced an amplification signal; these were noted as detectable but not quantifiable (< DL). A 4$^{th}$ replicate was used to estimate the reaction efficiency for the *Trichodesmium* and UCYN-B targets as previously described in Short et al., (2004). Seven and





two samples were below 95% in reaction efficiency for *Trichodesmium* and UCYN-B, respectively. The detection limit for the qPCR assays is 1-10 copies.

To determine the biovolume and C content of diazotrophs, cell sizes of *Trichodesmium* and UCYN-B were determined in samples from the photic layer of each LD station. Briefly, 2.3 L of surface (5 m) seawater were gently filtered onto 2 µm

nominal porosity (25 mm diameter) polycarbonate filters, fixed with paraformaldehyde (final conc. 2 %) and stored at -80 °C. Cells length and width measurements were performed on 25 to 50 cells per station at x400 magnification with a Zeiss Axio Observer epifluorescence microscope. The biovolumes (BV) of *Trichodesmium* and UCYN were estimated using the equation for a cylinder and a sphere, respectively (Sun and Liu, 2003). The cellular C contents were determined by using the relation between BV and C content according to Verity et al. (1992). Given that our work was performed in the field on wild

populations, we preferred using the biovolume estimate for C content rather than previously measured values based on published culture data. The C contents estimated here are within the range of those previously reported (Dekaezemacker and Bonnet, 2011; Dron et al., 2013; Hynes et al., 2012; Knapp et al., 2012; Luo et al., 2012). As DDAs were not easily identified on the filters, we used a C content of 1400 pg C cell$^{-1}$ for *Rhizosolenia* spp. (determined for the OUTPACE cruise, pers. comm. K. Leblanc) and assumed 6 cells per trichome of *Richelia intracellularis* (Foster and Zehr, 2006; Villareal,

1989), and 1 *Richelia* per diatom *Rhizosolenia (*both *Rhizosolenia* spp. and *R. intracellularis* are rarely reported as asymbiotic).

**Statistical analyses**

Spearman correlation coefficients were used to examine the relationships between DCM and nitracline depths during the station occupations ($\alpha = 0.05$).

A non-parametric Mann–Whitney test ($\alpha = 0.05$) was used to compare N$_2$ fixation rates obtained using in situ and on-deck incubation methods.

**3 Results**

**3.1 Hydrological background**

NO$_3^-$ concentrations in the photic layer were below quantification limit (0.05 µmol L$^{-1}$) during the station occupation at the

three LD stations. They became quantifiable below 70 m, 100 m and 120 m depth at LD A, B and C, respectively (Fig. 2) and at depths corresponding to a density anomaly of 23.59, 24.34 and 24.66 kg m$^{-3}$, respectively. The latter values correspond to the top of the nitracline, the isopycnal $\rho = \rho_{NO3}$ (Fig. 3; Table 4). The corresponding $\frac{d[NO_3]}{d\rho}$ were 3573, 5949 and 8888 µmol kg$^{-1}$ for stations LD A, LD B and LD C, respectively.

Averaged PO$_4^{3-}$ concentrations were close or below the quantification limit (0.05 µmol L$^{-1}$) from the surface to 20 m at LD A

and LD B, and then increased with depth to reach 0.46 and 0.36 µmol L$^{-1}$ at 200 m at LD A and LD B, respectively (Fig. 2).



At LD C, $PO_4^{3-}$ concentrations were always above the quantification limit and varied from 0.11 to 0.17 µmol L$^{-1}$ in the 0-120 m layer with the minimum concentration located at 60 m. Below 120 m, $PO_4^{3-}$ concentrations increased with depth to reach 0.43 µmol L$^{-1}$ at 200 m.

At LD A, the DCM and the $\rho_{NO3}$ depths were located between 60 and 100 m (Fig. 3). At LD B, the $\rho_{NO3}$ was located between 100 and 120 m, while the DCM was deepening from 25 m to 70 m along the station occupation. At LD C, the DCM and the $\rho_{NO3}$ were below the bottom of the photic layer (110-120 m), varying simultaneously between 115 and 155 m. The DCM and $\rho_{NO3}$ depths were significantly correlated ($p<0.05$) at LD A and LD C and not correlated ($p > 0.05$) at LD B. The depths of the photic layers, corresponding to 1 % of the surface PAR at midday, were 80-90 m at LD A, 60-70 m at LD B and 110-120 m at LD C.

### 3.2 Atmospheric deposition

The average of atmospheric deposition along the transect was below 0.2 µmol N m$^{-2}$ d$^{-1}$ (Table 4).

### 3.3 N$_2$ fixation rates

N$_2$ fixation rates measured using the in situ incubation method ranged 0.0 - 19.3 nmol N L$^{-1}$ d$^{-1}$ at LD A, 0.1 - 45.0 nmol N L$^{-1}$ d$^{-1}$ at LD B and 0.0 - 2.6 nmol N L$^{-1}$ d$^{-1}$ at LD C (Fig. 2). At LD A and LD B, maximum rates were measured near the surface (5 m) where they reached 19.3 and 45.0 nmol N L$^{-1}$ d$^{-1}$, and decreased with depth down to 0.5 nmol L$^{-1}$ d$^{-1}$ at 70 and 55 m, respectively at LD A and LD B. At LD C, N$_2$ fixation rates were 2 to 20 times lower than at LD B and LD A with a maximum of 2.6 nmol N L$^{-1}$ d$^{-1}$ located around 40 m. Close to the surface (5 m), rates were below quantification limit. At LD A and LD C, the three profiles measured on days 1, 3 and 5 at each station were similar to each other, while at LD B rates measured in the 0-40 m layer were different over the three sampling dates (Fig. 2), with rates decreasing over time.

N$_2$ fixation rates measured using the deck incubation method were not statistically different (Mann-Whitney paired test, $p<0.05$) than those measured using the in situ mooring line method (Fig. 2). They ranged 0.1 to 21.0 nmol N L$^{-1}$ d$^{-1}$ at LD A, 0.1 to 30.3 nmol N L$^{-1}$ d$^{-1}$ at LD B, and were below 1.2 nmol N L$^{-1}$ d$^{-1}$ at LD C. Overall, the profiles were similar between both methods, except the maximum at 40 m at LDC, which was not sampled with the on-deck incubation method.

Integrated N$_2$ fixation ($i$-$\rho$N$_2$) rates were 593 ± 51, 706 ± 302 and 59 ± 16 µmol N m$^{-2}$ d$^{-1}$ at LD A, LD B and LD C, respectively using data from the in situ incubation method (Fig. 2; Table 4) and 628 ± 156, 942 ± 253 and 56 ± 31 µmol N m$^{-2}$ d$^{-1}$ at LD A, LD B, and LD C, respectively, using data from the deck incubation method (Fig. 2). 80 % of $i$-$\rho$N$_2$ was reached at 36 m at LD A, 82 % was reached at 27 m at LD B and 78 % was reached at 60 m at LD C.

### 3.4 Vertical diffusive fluxes of nitrate

The averaged NO$_3^-$ input through vertical diffusion showed a strong contrast between the western station LD A and the two other stations, with mean values equal to 24.4 ± 24.4 µmol N m$^{-2}$ d$^{-1}$ at LD A and 6.7 ± 5.3 and 4.8 ± 2.2 µmol N m$^{-2}$ d$^{-1}$ at LD B and LD C, respectively (Fig. 4). Strong time variability was also observed with a typical standard deviation of the



same order as the mean value (Table 1). At LD A, a $NO_3^-$ peak input of 50 µmol N $m^{-2}$ $d^{-1}$ was observed on day 1 (26/02), while during days 2 and 3 (27/02 and 28/02) the daily average input was smaller than the average value for the station, between 5-10 µmol N $m^{-2}$ $d^{-1}$ without any peak input. At the end of LD A (days 4 and 5 - 01/03 and 02/03) the strongest $NO_3^-$ input variability was observed with instantaneous peaks reaching 46 to 89 µmol N $m^{-2}$ $d^{-1}$. When averaged per day,

daily input was minimum on day 2 (27/02) with 5 µmol N $m^{-2}$ $d^{-1}$ and maximum on day 5 (02/03) with 65 µmol N $m^{-2}$ $d^{-1}$ (red lines in Fig. 4a). At LD B, the mean daily $NO_3^-$ input varied within a factor of ~5, from 2 µmol N $m^{-2}$ $d^{-1}$ on day 5 (20/03) to 11 µmol N $m^{-2}$ $d^{-1}$ on day 2 (17/03). The largest daily averages obtained on days 2 and 3 were explained by the occurrence of $NO_3^-$ input peaks. At LD C, $NO_3^-$ input heterogeneously varied between 2 and 10 µmol N $m^{-2}$ $d^{-1}$ with minimum daily average on day 4 (27/03; 3 µmol N $m^{-2}$ $d^{-1}$) and maximum daily average on day 1 (24/03; 8 µmol N $m^{-2}$ $d^{-1}$).

Similarly as for the other stations, $NO_3^-$ input peaks were observed during the days of larger mean daily $NO_3^-$ input.

This time variability in $NO_3^-$ input was strongly influenced by the one of the vertical diffusion coefficient. Indeed the Kz time series showed a strong variability (Table 1) with peak values occurring intermittently during periods of enhanced turbulence, thus leading to peaks in $NO_3^-$ diffusive flux.

## 3.5 Particulate matter export

Mass fluxes recovered in the sediment traps at the three stations over the three sampling depths (150, 330 and 520 m) ranged from 13.6 to 87.2 mg of Dry Weight (DW) $m^{-2}$ $d^{-1}$ (Table 2). At LD A and LD C, fluxes decreased with depth, which was not observed at LD B. Maximum mass fluxes were measured at LD A, with 87.2 mg DW $m^{-2}$ $d^{-1}$ at 150 m, 23.9 mg DW $m^{-2}$ $d^{-1}$ at 330 m and 22.3 mg DW $m^{-2}$ $d^{-1}$ at 520 m. LD B presented the lowest export at 150 m (14.1 mg DW $m^{-2}$ $d^{-1}$) over the three stations. At LD C, 19.6 mg DW $m^{-2}$ $d^{-1}$ of particulate matter was exported at 150 m and the lowest export was recorded at

330 m.

Particulate C (PC) and PN recovered in the sediment traps followed the same patterns as the mass fluxes (Table 2), with a maximum export at 150 m at LD A, a minimum export at LD B and intermediate export at LD C. However, as PN and PC were not always in the same proportion in the exported matter, it induced variations of C:N ratios at the three stations, with averaged C:N ratios of 8.2 at LD A, 9.1 at LD B and 6.2 at LD C.

25 The mass of swimmers (zooplankton) recovered in the traps ranged from 10.5 to 376.1 mg DW $m^{-2}$ $d^{-1}$ (Table 3) and accounted from 36 to 94 % of total DW. The maximum was found at 330 m at LD A, and 150 m at LD B and LD C. As for particulate matter, zooplankton C (Zoo-C), N (Zoo-N) and P (Zoo-P) mass measured at each depth of each station followed the same pattern as the mass of swimmers recovered. Zoo-C ranged from 4.9 to 129.2 mg C $m^{-2}$ $d^{-1}$, Zoo-N ranged from 1.1 to 19.5 mg N $m^{-2}$ $d^{-1}$, and Zoo-P ranged from 0.03 to 0.41 mg P $m^{-2}$ $d^{-1}$.

## 3.6 Direct export of diazotrophs

*Trichodesmium* abundance measured in the sediment traps at the three stations ranged from below detection limit (< DL) to 2.67 $10^4$ *nifH* gene copies $mL^{-1}$ of sediment material (< DL at LD A 150 m and LD C 330 m) and represented less than 0.1 %



of *Trichodesmium* abundance integrated over the water column at the three stations based on data of Stenegren et al., (This issue). UCYN-B abundance measured in the traps ranged from < DL to 4.27 x10$^3$ *nifH* gene copies mL$^{-1}$. It accounted from 0.1 to 10.5 % of UCYN-B abundance integrated over the water column at LD A, and <0.5% at LD B and LD C. DDAs abundance, restricted to het-1 (*Richelia* associated with *Rhizosolenia* diatoms), ranged from < DL to 1.99 x10$^4$ *nifH* gene copies mL$^{-1}$ (< DL at LD A 150 and 520 m, LD B 150 m and LD C 330 m) and accounted up to 72.6 %, 2.9 % and 0.1 % of DDAs abundance integrated over the water column at LD A, LD B and LD C, respectively. While het-2 (*Richelia* associated with *Hemiaulus* diatoms) were observed in the water column (Stenegren et al., This issue), they were only detected in one sediment trap sample (LD B, 330 m) and < DL in 330 m from LD A and 500 m from LD B. When converted to C, diazotrophs represented between 5.4 % and 30.6 % (Fig. 5) of the total PC measured in the traps (Table 2) at LDA, from < 0.1 to 5.0 % at LD B and < 0.1 % at LD C. *Trichodesmium* and het-1 were the major contributors to diazotroph export at LD A and LD B (note that *Trichodesmium* data were not available for LDA (150 m) and UCYN-B and Het-1 were the major contributors at LD C.

## 4 Discussion

### Towards a daily N-budget

The analysis of hydrographic tracers and velocity structures present during our study at the three stations reveals that horizontal variability due to advection was important at spatial scales larger than the ones sampled during each station (de Verneil et al., this issue). Thus, we consider that we sampled the same water mass during each station and only vertical exchanges controlled input and output of N in the upper water column, which allow us to perform a daily N-budget at the three stations as summarized in Table 4.

### Contribution of N$_2$ fixation to new N input in the WTSP

The daily N-budget (Table 4) indicates that N$_2$ fixation was the major external source of N to the WTSP whatever the degree of oligotrophy, and represents more than 90 % of new N to the surface ocean at every station. This contribution is higher than in previous studies performed in other oligotrophic regions impacted by N$_2$ fixation (Table 5) such as the tropical North Atlantic (50 %, Capone et al., 2005) and Pacific (30-50 %, Dore et al., 2002; Karl et al., 2002), and higher than the average contribution at the global scale (Gruber, 2008). This previously unreported high contribution of N$_2$ fixation may have several origins.

- Firstly, atmospheric deposition measured during OUTPACE is in the lower end of fluxes measured in the world ocean (Guieu et al., Submitted), which is in agreement with the low atmospheric input estimates reported for the South Pacific by Wagener et al. (2008). This induces a negligible contribution of atmospheric input to the overall N budget (less than 1 %), therefore inducing a stronger contribution of other terms, such as N$_2$ fixation. At LD C, for



example, $N_2$ fixation was the major source of N even if rates were in the same range as those reported in the western Mediterranean Sea during the stratification period (Bonnet et al., 2011), where they represent a minor contribution.

- Then, $NO_3^-$ input by vertical diffusion appeared as the second source (1 to 8 %) of new N at the three stations. This contribution was lower than in previous studies in other oligotrophic regions (Table 5) where $NO_3^-$ input by vertical diffusion contributes ~ 18 % of new N in the Indian South Subtropical Gyre (Fernández-Castro et al., 2015),  and ~ 50 % in the Tropical North Atlantic (Capone et al., 2005). In most studies (Fernández-Castro et al., 2015; Moutin and Prieur, 2012; Painter et al., 2013), an average Kz value is used (i.e. averaged over the cruise, over a station or over depth) to determine $NO_3^-$ input by turbulence in the photic layer. In this study we performed high frequency direct measurements of Kz and highlighted the importance of turbulent event pulses on diffusive $NO_3^-$ input. For example, pulses occurred for only 30 % of the time at LD A, but represented circa 80 % of the total input of $NO_3^-$ to the photic layer. Using a constant Kz of $10^{-5}$ $m^2$ $s^{-1}$ at the 3 stations decreases the $NO_3^-$ input down to 22.9 µmol N $m^{-2}$ $d^{-1}$ at LD A and increases $NO_3^-$ input up to 19.9 and 25.5 µmol N $m^{-2}$ $d^{-1}$ at LD B and LD C, that is 2.7 and 4.8 times higher than using a high frequency Kz for the latter two stations. The contrasted $NO_3^-$ input observed at the three stations results from the high variability in turbulence along the west-east transects (Bouruet-Aubertot et al., this issue). Thus, using a constant Kz removes the contrasted $NO_3^-$ input between the 3 stations (~ 4 times higher at LD A than at LDB and LD C). Consequently, using average Kz values for the diffusive flux computation can lead to significant bias. In our study, $NO_3^-$ input was calculated at the top of the nitracline. Painter et al. (2013) have demonstrated the variability that may be introduced into the estimated $NO_3^-$ input by the depth of the defined nitracline. With a constant Kz in the 2 cases, they estimated that $NO_3^-$ input was 5 times lower at the top of nitracline depth than at the maximum gradient depth. In our study, the $NO_3^-$ input would also be ~ 3-4 times higher if calculated at the maximum gradient depth rather than at the top nitracline, mainly due to the increase of the nitracline gradient up to 48 µmol N $m^{-4}$ (Table 1). However, in all cases, the $NO_3^-$ input by turbulence was always found to provide a subordinate contribution to the N budget.

- Finally, the high contribution of $N_2$ fixation to new N input in the photic layer results from the intrinsically high $N_2$ fixation rates we measured in the WTSP (especially in MA waters), that are part of the hot spot of $N_2$ fixation reported by Bonnet et al., (2017), with rates being in the upper range of rates reported in the global $N_2$ fixation Marine Ecosystem Data (MAREDAT) database (Luo et al., 2012). Those high $N_2$ fixation rates are as high as westward in the Salomon Sea (Berthelot et al., 2017; Bonnet et al., 2015), extending the hot spot of $N_2$ fixation to the whole WTSP (Bonnet et al., 2017).

The contribution of $N_2$ fixation to PP was around 15-21 % in MA waters and 4 % in SPG waters. The high contribution measured in the MA region is an order of magnitude higher than the one reported in previous studies performed in the Pacific Ocean (Moutin et al., 2008; Raimbault and Garcia, 2008; Shiozaki et al., 2014), the Atlantic Ocean (Fonseca-Batista et al., 2017; Rijkenberg et al., 2011) and the Mediterranean Sea (Moutin and Prieur, 2012) where they never exceed 5%, and also slightly higher than the contribution reported from a mesocosm experiment in the New Caledonia lagoon during a



UCYN bloom ($10.8 \pm 5.0$ %; Berthelot et al., 2015). As there was low supply of $NO_3^-$ through vertical diffusion (< 8 %) and atmospheric deposition (< 1 %), $N_2$ fixation sustains nearly all new production during the austral summer in the WTSP.

**Coupling between $N_2$ fixation and export in the WTSP**

Previous studies have used different methods for coupled measurements of $N_2$ fixation and export (Berthelot et al., 2015;
Dore et al., 2008; Karl et al., 2012; Scharek et al., 1999a; Subramaniam et al., 2008; White et al., 2012). The Lagrangian strategy used here was designed to sample the same water mass during the experiment and therefore minimize the methodological issues associated with particulate export flux measurements using sediment traps in the open ocean (Monroy et al., 2017). The severe meteorological conditions due to the development of tropical cyclone PAM (a category 5 storm) that hit the Vanuatu Islands on March 2015, obliged us to perform the LD B station at a more easterly location than initially
planned (Moutin et al., 2017). LD B was therefore sampled in a surface bloom with a DCM close to the surface (Fig. 3), contrary to LD A and LD C sampled in a zone with a DCM near the bottom of the photic layer (Fig. 3). Thus, data from LD B, although presented together with LD A and LD C, will be discussed apart.

Stations LD A and LD C were considered as oligotrophic and ultra-oligotrophic, respectively. PP was twice higher and the DCM shallower at station LD A compared to LD C. Furthermore, the diazotroph community composition was contrasted
between both stations, with a clear domination of *Trichodesmium* at LD A ($6.6 \times 10^4$ nifH copies $L^{-1}$ at 5 m) and lower abundance of diazotrophs and a clear domination of UCYN-B ($3.6 \times 10^3$ nifH copies $L^{-1}$ at 5 m) and het-1 ($3.0 \times 10^3$ nifH copies $L^{-1}$ at 5 m) (Stenegren et al., this issue) at LD C. The e-ratio (e-ratio = PC export/PP) calculated at LD A (9.7 %) was higher than the e-ratio in most studied oligotrophic regions (Karl et al., 2012; Moutin and Prieur, 2012; Raimbault and Garcia, 2008), where it rarely exceeds 1 %, indicating a high efficiency of the WTSP to export C relative to PP. Moreover, the e-
ratio was higher at LD A (characterized by high $N_2$ fixation rates (593 µmol N m$^{-2}$ d$^{-1}$)) than at LD C (characterized by low $N_2$ fixation (59 µmol N m$^{-2}$ d$^{-1}$)). This is in agreement with previous studies reporting typical e-ratios of 1 % in ultra-oligotrophic regions characterized by low $N_2$ fixation rates (like LD C) such as the eastern SPG (Moutin et al., 2008; Raimbault and Garcia, 2008) or the Mediterranean Sea (Bonnet et al., 2011), and typical e-ratios of 5 % in regions characterized by high $N_2$ fixation rates such as station ALOHA (Karl et al., 2012). Taken together, these results suggest that
$N_2$ fixation would enhance particle export. This is supported by Knapp et al. (this issue) who showed that nearly all exported production was supported by $N_2$ fixation in MA waters during the OUTPACE cruise.

At station ALOHA, the e-ratio varies between 2 to 15 % and is maximum during summer export fluxes of PN, that are attributed to the direct export of DDAs (Karl et al., 2012). In the present study, we investigated the potential direct export of diazotrophs by measuring the abundance of each diazotroph group in the traps. We reveal that the export efficiency of
*Trichodesmium*, i.e. the percentage of organisms present in the water column recovered in the traps (< 0.1 %) was lower than that of other diazotrophs, which is in agreement with Walsby (1992) and Chen et al. (2003), who revealed that *Trichodesmium* are rarely recovered in the sediment traps. The export efficiency of UCYN-B (2.3 % on average) and het-1 (4.0 % on average) was higher than that of *Trichodesmium*, which is in agreement with Bonnet et al., (2016b), who reported



that UCYN-C were efficiently exported thanks to aggregation processes in a mesocosm experiment in New Caledonia and Karl et al., (2012). In this study, the contribution of diazotrophs to PC export was up to 30.6 % at LD A, and was mainly driven by het-1 as the estimated het-1 C content was higher than that of UCYN-B and *Trichodesmium*. This suggests that DDAs were efficiently exported, which is in agreement with previous studies (Karl et al., 2012; Subramaniam et al., 2008).

At LD C, less than 0.1 % of the total PC measured in the traps was associated with diazotrophs, which is probably due to lower abundances of het-1 in the traps (< DL) than at LD A and the dominance of UCYN-B (at 330 m) having low cellular C contents. The contribution of diazotrophs to PC export at LD A (up to 30.6 % at 330 m) was high compared to what has been measured in a much smaller water column (15 m-high mesocosms) in New Caledonia (ca. 20 %; Bonnet et al, 2016a) and suggests that the direct export of diazotrophs should be further investigated in oligotrophic open ocean regions as to date as

few qPCR data on *nifH* in traps are available (Karl et al., 2012) to compare with our study. However, it has to be noted that *Trichodesmium* dominated the diazotroph community at LD A where we measured the highest export and e-ratio; this suggests that most of the export was likely indirect, i.e. after the transfer of diazotroph-derived N (DDN) to the surrounding bacterial, phytoplankton and zooplankton communities as revealed by Caffin et al., (this issue) during the same cruise, that are subsequently exported.

Station LD B was studied during a surface *Trichodesmium* bloom; however observations of poor cell integrity were reported (Stenegren et al., this issue) and other evidence indicated the senescence of the bloom (de Verneil et al., this issue). Higher N$_2$ fixation and integrated PP rates than those measured at LD A and C together with lower PN export resulted in an e-ratio less than 0.8 % (Table 4) at LD B. This very low export efficiency is probably related to the fact that we sampled station LD B during a collapsing *Trichodesmium* bloom (Stenegren, this issue) triggered by PO$_4^{3-}$ starvation (de Verneil, 2017) as

already reported in the WTSP (Moutin et al., 2005). The collapse of *Trichodesmium* blooms can possibly result from viral lysis (Hewson et al., 2004), mainly leading to the release of dissolved N in surrounding waters, or programmed cell death, mainly leading to rapid sinking of biomass and may influence C export (Bar-Zeev et al., 2013). PCD was detected at LD B (Berman-Frank et al., this issue), indicating that this process cannot be excluded, while *Trichodesmium* were not recovered in the traps (they dominated the export of diazotrophs at 150 m but altogether the direct export of diazotrophs never exceed

1.1 % of total export). It is thus likely that most of the N accumulated in the phytoplankton pool (including *Trichodesmium*) was released to the dissolved pool due to grazing and viral lysis, then quickly remineralized due to high microbial activity at LD B (Van Wambeke et al., this issue), and thus would explain the low export measured at this station. This result is supported by the efficient transfer of DDN to the surface planktonic food web at this station as described in Caffin et al., (This issue) and previously by Bonnet et al. (2016b) and Berthelot et al. (2016) in the WTSP. Thus, elevated Chl *a* patches as

we sampled at LD B may be more productive areas than the ambient oligotrophic waters but less efficient in terms of export, corresponding to the concept of "high-biomass, low-export" (HBLE) environments initially reported for the Southern Ocean (Lam and Bishop, 2007) where surface waters with high biomass were associated with low particle export at depth.



**Disequilibrium of new vs. exported production**

The daily N budget computed here reveals that N input into the photic layer through atmospheric deposition, $N_2$ fixation and vertical $NO_3^-$ diffusion exceeded N output through organic matter export at the three studied stations. This imbalance between new and exported production is also observed in different oligotrophic regions of the ocean as the SPG (Raimbault
and Garcia, 2008), the Barents Sea (Olli et al., 2002; Reigstad et al., 2008; Wexels Riser et al., 2008), the North and South Atlantic gyres (Thomalla et al., 2006), and the Equatorial Pacific (Bacon et al., 1996). It has to be noted that our budget was performed at the daily scale but at the annual time scale or longer time scales, the PN export from the photic layer is supposed to balance new N input (Dore et al., 2002; Eppley and Peterson, 1979).

This imbalance between new and exported N frequently reported in the oligotrophic ocean may result from (1) a spatial
decoupling between production and export, (2) a temporal decoupling between production and export and/or (3) processes other than particle export such as DON and/or zooplankton export. As we used the Lagrangian strategy described above and confirmed that we sampled the same water at all LD stations during our surveys (de Verneil et al., this issue), the first option (spatial decoupling) can be excluded. The second option (2) would mean that we performed our budget during a period corresponding to production of organic matter that was dissociated from the export that would have happened later. Such
temporal lag has already been reported in the Southern Ocean (Nodder and Waite, 2001), accompanied by biomass accumulation in the photic layer; therefore, we cannot exclude this hypothesis here. Regarding the third possible explanation, the primary process by which organic matter is exported out of the photic layer is the gravitational sinking of particles to the deep ocean (Karl et al., 1996; Knauer et al., 1990). However, two other main processes can contribute to export: physical mixing resulting in export of dissolved organic matter (Carlson et al., 1994; Carlson and Ducklow, 1995; Copin-Montégut
and Avril, 1993; Toggweiler, 1989) and zooplankton diel-migrations that actively transport organic matter out of the photic layer (Longhurst et al., 1989, 1990; Longhurst and Glen Harrison, 1988; Vinogradov, 1970). In the present study, the DON export flux was limited to eddy diffusion as we performed our survey during the stratification period, and the low downward flux of DON estimated by Moutin et al., (this issue) was unable to explain the observed imbalance. However, zooplankton might play a significant role, although hard to quantify. Zooplankton living below the photic layer migrate to the surface at
night and when going down, can increase the export of dissolved and particulate organic and inorganic N through defecation, excretion or mortality (Atkinson et al., 1996; Le Borgne and Rodier, 1997; Dam et al., 1993, 1995, Longhurst et al., 1989, 1990; Longhurst and Williams, 1992; Zhang and Dam, 1998) if it happens below a barrier to vertical mixing (i.e. nitracline or pycnocline (Longhurst et al., 1989, 1990)). Zooplankton has been reported to represent between 4.9 and 38 % of the total export flux (Table 6) in several ecosystems, and therefore should be considered in export calculations. Here we estimated the
maximum contribution of zooplankton using the higher value reported in Table 6 (38 %). It reached a maximum of 106 μmol N $m^{-2}$ $d^{-1}$ at LD A, 12 μmol N $m^{-2}$ $d^{-1}$ at LD B and 18 μmol N $m^{-2}$ $d^{-1}$ at LD C at 150 m. By applying this correction to our export values, it cannot explain the observed disequilibrium between new and exported N.



Finally, the zooplankton itself is sampled by the traps but cannot be distinguished from swimmers. In the present study, the zooplankton contribution to PN export (Table 3) was high (68 % on average) and above that measured in other oligotrophic areas such as the Mediterranean Sea (Moutin and Prieur, 2012). All zooplankton recovered in traps were considered as swimmers and were therefore discarded (Tables 1 and 3), which may have lead to underestimate the PN export and could

also partly explain the observed disequilibrium between new and export production. Further studies should be investigated to assess the contribution of living versus dead zooplankton to PN.

In summary, we suggest that zooplankton plays a key role on the export in the WTSP and its contribution would increase the particulate export. Moreover, zooplankton activity can transfer N accumulated in the phytoplankton pool to the dissolved pool following grazing and related trophic processes. N from the dissolved pool is then remineralized by microbial activity

and accumulates in the photic layer, thus N is not recovered in sediment traps

## 5 Conclusion

In this study, we successfully used a Lagrangian strategy in the WTSP to follow the same water mass during 5 days in order to perform N budgets during the stratification period (Feb. - March 2015) at 3 stations. $N_2$ fixation appeared as a substantial biogeochemical process providing the major external source of N in the photic layer. *Trichodesmium* was the major

diazotroph in the oligotrophic MA waters (LD A and LD B), while UCYN dominated the diazotroph community in the ultra-oligotrophic waters of the gyre (LD C). $N_2$ fixation contributed to ~15-21 % of the estimated PP in the MA region where $N_2$ fixation rates were high, and to ~3 % in the SPG water where $N_2$ fixation rates were low. As there was limited supply of $NO_3^-$ through vertical eddy diffusion (< 8 %) and atmospheric deposition (< 1 %), $N_2$ fixation accounted for nearly all new production (more than 90 % of new N). The current coupling between typical high $N_2$ fixation rates of the WTSP, with the

high PN and PC export measured in this region associated with high e-ratios (up to ~10 %) suggests that $N_2$ fixation plays an important role in export during austral summer conditions in the WTSP, either directly or indirectly. The export efficiency measured here in the WTSP (LD A) is comparable to the one measured in the Southern Ocean (Rembauville et al., 2015) considered as an efficient ecosystem for C export. The oligotrophic ocean represents 60 % of the global ocean surface, and therefore may play a more significant role in C export than initially considered (Baines et al., 1994; Wassmann, 1990). Even

if the link between $N_2$ fixation rates and export is obvious, the possible temporal decoupling between these processes and the potential role of zooplankton need to be further investigated.

## Acknowledgements

This is a contribution of the OUTPACE (Oligotrophy from Ultra-oligoTrophy PACific Experiment) project (https://outpace.mio.univ-amu.fr/) funded by the French research national agency (ANR-14-CE01-0007-01), the LEFE-
30 CyBER program (CNRS-INSU), the GOPS program (IRD) and the CNES (BC T23, ZBC 4500048836). The OUTPACE




cruise (http://dx.doi.org/10.17600/15000900) was managed by MIO (OSU Institut Pytheas, AMU) from Marseilles (France). The authors thank the crew of the RV *L'Atalante* for outstanding shipboard operations. G. Rougier and M. Picheral are warmly thanked for their efficient help in CTD rosette management and data processing, as well as C. Schmechtig for the LEFE-CyBER database management. The satellite-derived data of Sea Surface Temperature, Chl *a* concentrations and currents have been provided by CLS in the framework of the CNES funding; we warmly thank I. Pujol and G. Taburet for their support in providing these data. We acknowledge NOAA, and in particular R. Lumpkin, for providing the SVP drifters. All data and metadata are available at the following web address: http://www.obs-vlfr.fr/proof/php/outpace/outpace.php. Argo DOI (http://doi.org/10.17882/42182).

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



**Table 1:  Mean Kz, mean nitracline gradient, mean NO$_3^-$ flux and associated standard deviations over the station occupation at LD A, LD B and LD C at the top nitracline and at the maximum gradient.**

| Station | Kz $m^2 s^{-1}$ | Nitracline gradient $\mu mol\ N\ m^{-4}$ | NO$_3^-$ flux $\mu mol\ N\ m^{-2}\ d^{-1}$ |
|---|---|---|---|
| *Top nitracline* | | | |
| LD A | $1.11\ 10^{-5} \pm 1.00\ 10^{-5}$ | 23 ± 13 | 24.4 ± 24.4 |
| LD B | $3.59\ 10^{-6} \pm 3.11\ 10^{-6}$ | 21 ± 12 | 6.7 ± 5.3 |
| LD C | $2.04\ 10^{-6} \pm 1.11\ 10^{-6}$ | 27 ± 13 | 4.8 ± 2.2 |
| *Maximum gradient* | | | |
| LD A | $1.69\ 10^{-5} \pm 1.15\ 10^{-5}$ | 53 ± 10 | 79 ± 56 |
| LD B | $4.52\ 10^{-6} \pm 3.22\ 10^{-6}$ | 48 ± 6 | 19 ± 14 |
| LD C | $2.96\ 10^{-6} \pm 1.84\ 10^{-6}$ | 48 ± 14 | 21 ± 11 |

**Table 2: Sediment trap data at the three LD stations. Depth of collection, mean mass flux of DW matter, particulate C, N and P flux, mean C:N:P molar ratio. No data was collected at LD C at 520 m.**

| Station | depth m | Mass flux mg DW m$^{-2}$ d$^{-1}$ | PC mg C m$^{-2}$ d$^{-1}$ | PN mg N m$^{-2}$ d$^{-1}$ | PP mg P m$^{-2}$ d$^{-1}$ | C:N:P ratio 106:x:y |
|---|---|---|---|---|---|---|
| | 150 | 87.2 ± *41.1* | 27.1 ± *12.1* | 3.9 ± *1.8* | 1.38 ± 0.71 | 106:13:2.1 |
| LD A | 330 | 23.9 ± *15.2* | 5.8 ± *4.0* | 0.9 ± *0.8* | 0.13 ± 0.16 | 106:14:0.9 |
| | 520 | 22.3 ± *4.6* | 4.7 ± *1.0* | 0.6 ± *0.2* | 0.15 ± 0.23 | 106:12:1.3 |
| | 150 | 14.1 ± *6.5* | 3.5 ± *1.3* | 0.4 ± *0.2* | 0.06 ± 0.07 | 106:10:0.7 |
| LD B | 330 | 16.8 ± *9.0* | 3.2 ± *2.0* | 0.4 ± *0.2* | 0.20 ± 0.37 | 106:11:2.6 |
| | 520 | 17.9 ± *10.5* | 3.7 ± *1.4* | 0.5 ± *0.2* | 0.04 ± 0.03 | 106:12:0.4 |
| | 150 | 19.6 ± *6.9* | 3.8 ± *0.8* | 0.7 ± *0.2* | 0.03 ± 0.01 | 106:17:0.3 |
| LD C | 330 | 13.6 ± *6.4* | 2.6 ± *0.9* | 0.5 ± *0.2* | 0.04 ± 0.02 | 106:17:0.6 |
| | 520 | - | - | - | - | - |

**Table 3: Zooplankton sediment traps data at the three LD stations. Depth of sampling, mean DW zooplankton flux, C and N zooplankton fluxes.**

| Station | depth m | Zooplankton DW (Swimmers) mg m$^{-2}$ d$^{-1}$ | Zoo-C mg C m$^{-2}$ d$^{-1}$ | Zoo-N mg N m$^{-2}$ d$^{-1}$ | Zoo-P mg P m$^{-2}$ d$^{-1}$ | C:N:P ratio 106:x:y |
|---|---|---|---|---|---|---|
| | 150 | 82.1 ± *18.0* | 42.3 ± *7.6* | 8.2 ± *1.2* | 0.31 ± *0.28* | 106:18:0.3 |



| | | | | | | | | | |
|---|---|---|---|---|---|---|---|---|---|
| LD A | 330 | 376.1 | ± | *26.1* | 129.2 ± *15.5* | 19.5 ± *4.0* | 0.41 ± *0.36* | 106:14:0.1 |
| | 520 | 14.1 | ± | *8.0* | 5.3 ± *2.2* | 1.3 ± *0.4* | 0.03 ± *0.03* | 106:22:0.2 |
| | 150 | 112.9 | ± | *38.1* | 57.1 ± *21.7* | 11.3 ± *4.7* | 0.39 ± *0.25* | 106:18:0.3 |
| LD B | 330 | 62.3 | ± | *31.6* | 27.3 ± *15.5* | 4.9 ± *2.8* | 0.17 ± *0.19* | 106:16:0.3 |
| | 520 | 10.5 | ± | *3.0* | 4.9 ± *1.2* | 1.1 ± *0.3* | 0.03 ± *0.01* | 106:20:0.3 |
| | 150 | 121.3 | ± | *37.5* | 41.7 ± *14.8* | 10.3 ± *5.0* | 0.38 ± *0.13* | 106:22:0.4 |
| LD C | 330 | 31.0 | ± | *5.1* | 14.3 ± *5.0* | 2.9 ± *1.3* | 0.10 ± *0.02* | 106:18:0.1 |
| | 520 | - | | | - | - | - | |

**Table 4: Mean daily N-budget at the three stations LD A, LD B and LD C.**

| Characteristics | Units | LD A | LD B | LD C |
|---|---|---|---|---|
| $\rho_{NO3}$ | kg m$^{-3}$ | 23.59 | 24.34 | 24.66 |
| $\dfrac{d[NO_3]}{d\rho}$ | µmol kg$^{-1}$ | 3573 | 5949 | 8888 |
| Atmospheric deposition | µmol N m$^{-2}$ d$^{-1}$ | 0.2 | 0.2 | 0.2 |
| N$_2$ Fixation | µmol N m$^{-2}$ d$^{-1}$ | 593 | 706 | 59 |
| NO$_{3-}$ diffusion | µmol N m$^{-2}$ d$^{-1}$ | 24 | 7 | 5 |
| $\sum$ N inputs | µmol N m$^{-2}$ d$^{-1}$ | 617 | 713 | 64 |
| Integrated N-PP | µmol N m$^{-2}$ d$^{-1}$ | 2885 | 4709 | 1678 |
| Export N -150 m | µmol N m$^{-2}$ d$^{-1}$ | 279 | 31 | 47 |
| Export N - 330 m | µmol N m$^{-2}$ d$^{-1}$ | 64 | 29 | 36 |
| Export N - 520 m | µmol N m$^{-2}$ d$^{-1}$ | 44 | 37 | - |
| e-ratio -150 m | % | 9.7 | 0.7 | 2.8 |
| e-ratio - 330 m | % | 2.2 | 0.6 | 2.1 |
| e-ratio - 520 m | % | 1.5 | 0.8 | - |
| Theoretical regenerated production | % | 79 % | 85 % | 96 % |

**Table 5: Contribution of N$_2$ fixation and NO$_3^-$ vertical diffusion to new N inputs in oligotrophic region.**





| Location | Contribution to new N | | Source |
|---|---|---|---|
| | N$_2$ fixation | NO$_3^-$ diffusion | |
| Tropical North Atlantic | 50 % | 50 % | Capone et al., 2005 |
| Subtropical North Atlantic | 2 % | - | Mourino-Carballido et al., 2011 |
| Subtropical South Atlantic | 44 % | - | Mourino-Carballido et al., 2011 |
| South Atlantic Gyre | 21 % | 24 % | Fernández-Castro et al., 2015 |
| Indian South Subtropical Gyre | 12 % | 18 % | Fernández-Castro et al., 2015 |
| Mediterranean Sea | 0 – 32 % | 21 – 53 % | Moutin and Prieur, 2012 |
| | | | Bonnet et al., 2011 |
| North Pacific subtropical Gyre | 30 – 50 % | - | Karl et al., 2003 |
| North Pacific subtropical Gyre | 48 % | 52 % | Dore et al., 2002 |
| Western Tropical South Pacific | 92 – 99 % | 1 – 8 % | This study |

**Table 6: Contribution of N export by active zooplankton migration to total PN export.**

| Location | % of PN export | Source |
|---|---|---|
| Subtropical and tropical Atlantic | 7.6 | Longhurst et al., 1989, 1990 |
| North Atlantic BATS | 37.3 | Dam et al., 1995 |
| Equatorial Pacific | 4.9 | Le Borgne and Rodier, 1997 |
| Equatorial Pacific | 19.8 | Le Borgne and Rodier, 1997 |
| North Atlantic BATS | 19.9 | Al-Mutairi and Landry, 2001 |
| North Pacific subtropical Gyre | 38 | Hannides et al., 2009 |
| California Current Ecosystem | 20 | Stukel et al., 2013 |
| Cost Rica Dome | 38 | Stukel et al., 2015 |





**Figures caption**

**Figure 1: Position of the long duration stations sampled in this study (OUTPACE cruise): LD A in green, LD B in red and LD C in blue on a quasi-Lagrangian surface Chl $a$ concentrations map. The in situ production lines were deployed in the photic layer (from 5 to 105, 80 and 180 m for LD A, LD B and LD C, respectively) and the PPS5 sediment traps were deployed at 150, 330 and 520 m.**

**Figure 2: Vertical profiles of net $N_2$ fixation rates (nmol N $L^{-1}$ $d^{-1}$) estimated using in situ incubations at day 1 (red circles), day 3 (orange circles) and day 5 (purple) and using on-deck incubations (purple filled area) at stations LD A (left), LD B (middle) and LD C (right). The $NO_3^-$ concentrations x10 averaged over the 5 days of station occupation are also reported (light blue squares : μmol $L^{-1}$), as well as $PO_4^{3-}$ concentrations x100 (dark blue squares: μmol $L^{-1}$), and fluorescence/chlorophyll x100 (green dots : μg $L^{-1}$)**

**Figure 3: Temporal evolution of PAR, DCM (green line), $\rho_{NO3}$ (red line), and 1 % of surface PAR (yellow dots and crosses) during the three stations' occupation period (LD A: top panel, LD B: middle panel, LD C: bottom panel).**

**Figure 4: Temporal evolution of upward vertical $NO_3^-$ flux (μmol N $m^{-2}$ $d^{-1}$) calculated at the top of the nitracline for each station (LD A: top panel, LD B: middle panel and LD C: bottom panel) after temporal interpolation (blue). Daily mean from noon to noon in dash orange line and occupation period mean in green line.**

**Figure 5: Relative contribution of each diazotroph (*Trichodesmium* in red, UCYN-B in orange and Het-1 in yellow) to the total PC associated with diazotrophs (diazotroph-PC) in the sediment traps at 150 m (top), 330 m (middle) and 520 m (bottom), at the three stations LD A (left panel), LD B (middle panel) and LD C (right panel). Values in blue correspond to the contribution of diazotroph-PC to total PC measured in the traps. No *Trichodesmium* valid data available at LD A 150 m.**



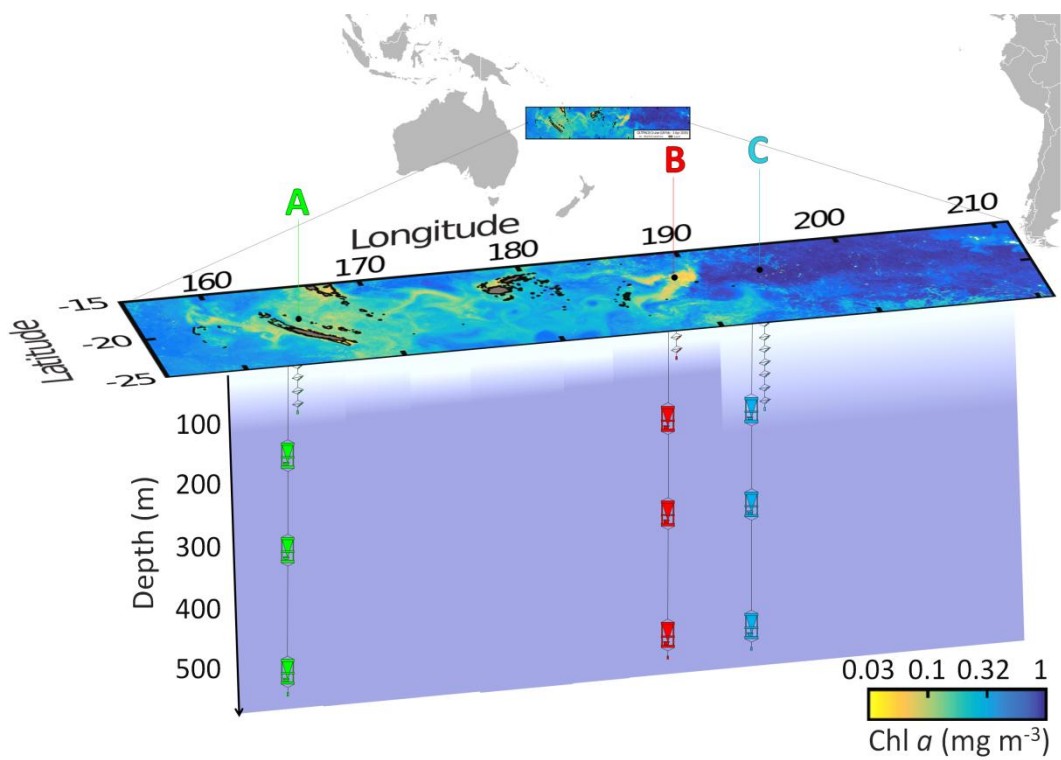

**Figure 1**





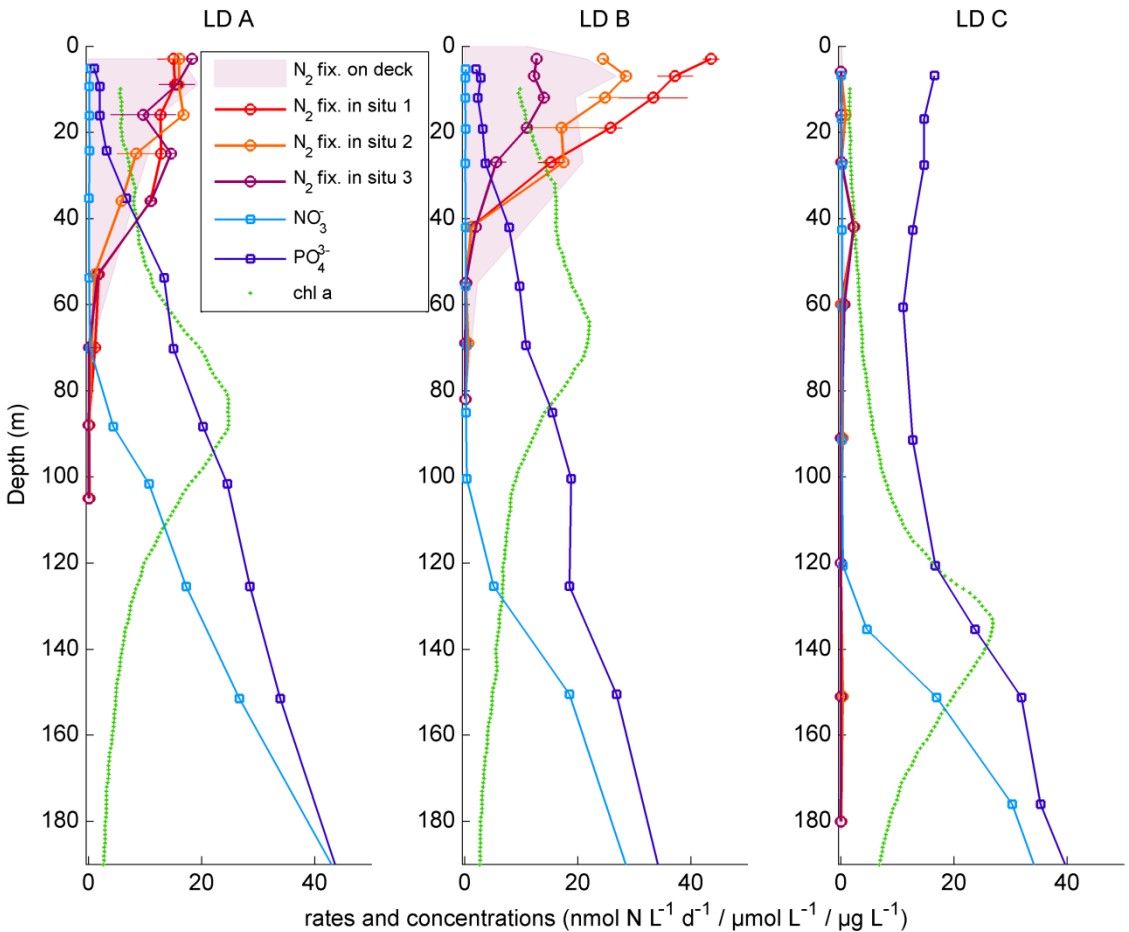

**Figure 2**



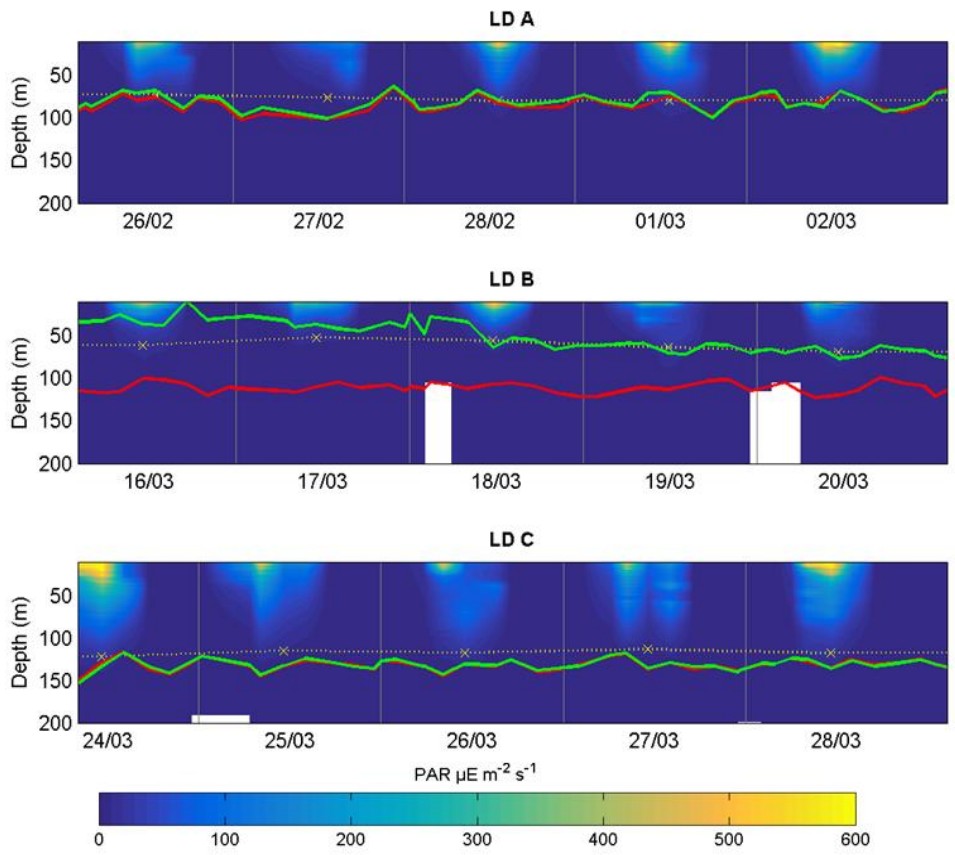

**Figure 3**



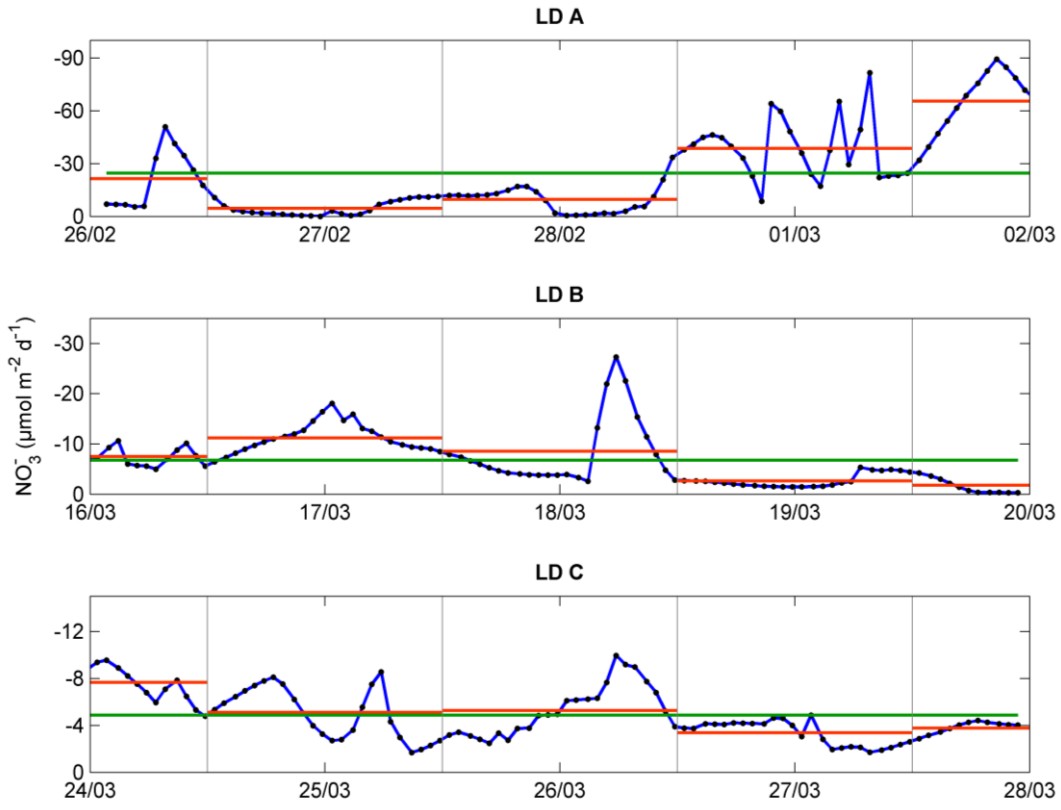

**Figure 4**

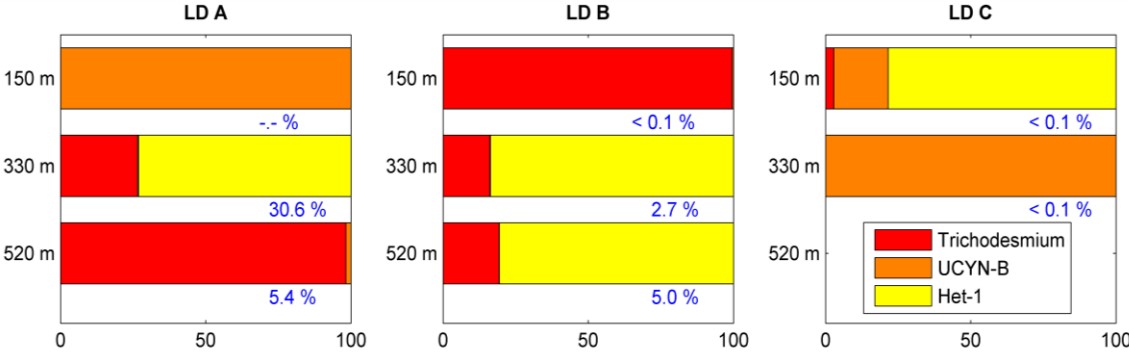

5    **Figure 5**