# Peer review of "N2 fixation as a dominant new N source in the Western Tropical South Pacific Ocean (OUTPACE cruise)"

_Biogeosciences, 2017_

## Referee Comment (RC1) · Anonymous Referee #1 · 10 Nov 2017

**General comments:**
In their study, Caffin *et al.* use a Lagrangian approach to determine N sources ($N_2$ fixation rates, $NO_3^-$ supply from vertical diffusion and N from atmospheric depositions) and sinks (particulate N export) at 3 stations in the Western South Tropical Ocean. They also measured primary production using $^{14}C$ tracer incubations. Their main findings are that 1) $N_2$ fixation is the major source of new N (>90%) at all stations, regardless of whether the diazotroph community was dominated by *Trichodesmium* or unicellular cyanobacteria, 2) carbon export relative to primary production is high in this region, and 3) the sum of N input in the photic zone exceeds particulate N export. Overall, this study is interesting and timely as this region has recently been identified as a $N_2$ fixation hotspot. However, I have some concerns. First, they report a low input from atmospheric depositions but did not consider atmospheric sources other than $NO_3^-$ and $NO_2^-$ (e.g., $NH_4^+$ DON, PON). They also measured relatively high $N_2$ fixation rates and I am wondering if they considered the possibility of a contamination of their $^{15}N-N_2$ stock with $^{15}N-NO_3^-$ and $NH_4^+$, which would artificially increase their $N_2$ fixation rates, as recently reported by Dabundo *et al.* (2014). Some of the references cited (this issue) were unavailable on the *Biogeosciences Discussions* online forum at the time of this review, making it impossible to evaluate these parts of the manuscript. I was also a bit confused regarding the novelty of their dataset: were the same $N_2$ fixation rates, qPCR or any other data collected at the same stations during the OUTPACE cruise already published in previous studies? The authors should make a clear distinction of the new data contributed by their study versus the data already published elsewhere in other manuscripts in the special issue.

**Specific comments:**

**Title**
The title is a bit long and not focused on the main point of the study. I suggest changing for: "$N_2$ fixation as the dominant new N source in the Western Tropical South Pacific Ocean (OUTPACE cruise)"

**Introduction**
Page 2, line 15: Knapp *et al.* (2008) and Bourbonnais *et al.* (2009) also observed a low $\delta^{15}N$ of $NO_3^-$ (relative to $\delta^{18}O-NO_3^-$) in surface waters in the western and eastern subtropical Atlantic Ocean, supporting the role of $N_2$-fixers in these regions.

Page 4, lines 6: What factors influence the distribution of *Trichodesmium* or UCYN? I believe temperature is an important factor (see Moisander *et al.*, 2010). This point should be discussed a bit more.

**Experimental procedures**
Page 5, line 19: They only considered $NO_3^-$ and $NO_2^-$ when quantifying N atmospheric depositions. They should also consider $NH_4^+$ or organic nitrogen (particulate or dissolved). For instance, Cornell *et al.* (1995) estimated that organic nitrogen was a significant component of atmospheric N depositions even in remove marine regions.

Page 5, line 26: Did they check their commercial $^{15}$N Eurisotop gas for possible contamination with $^{15}$N-labeled dissolved inorganic nitrogen ($NO_3^-$, $NO_2^-$ and $NH_4^+$)? Microbial assimilation of contaminant $^{15}$N labeled dissolved inorganic nitrogen would artificially increase $N_2$ fixation rates. Dabundo *et al*. (2014) recently reported significant concentrations of $^{15}$N contaminants in $^{15}$N-labelled $N_2$ gas supplied by Sigma-Aldrich and Campro Scientific.

Page 6, lines 5-7: A better way to assess whether equilibration was complete would be to try different treatments in triplicate, i.e., shake the bottles for different times and intensity before the *in-situ* incubations.

Page 6, line 6: Add "incomplete" before equilibration.

Page 6, lines 7-10: I assume the 12 mL subsample was collected without contact with the atmosphere?

Page 6, lines 16-24: What is the detection limit for their $N_2$ fixation rates?

Page 7, line 21: Define UCYN-B, UCYN-A1, het-1 and het-2. Which bacteria are represented by these different groups?

Page 8, line 1: Were samples with qPCR reaction efficiency below 95% reported? Why not repeat analysis for these samples?

**Results**

Page 9, lines 24 to 27: How are these rates different from the one measured in Bonnet *et al*., 2017 (This issue). Are they the same rates as reported in Bonnet *et al*. (2017)? The Bonnet *et al.* paper was not yet available at the time of this review, making it impossible to effectively evaluate this part of the manuscript.

Page 11, line 11: Were *Trichodesmium* data for LD A (150 m) not available or below detection limit (as stated on page 10, line 32)?

**Discussion**

Page 11, lines 27-29: Perhaps atmospheric deposition measured during OUTPACE are low because they neglected contributions from organic nitrogen and $NH_4^+$. This possibility should be discussed (see my previous comment, page 5, line 19).

Page 12, lines 24-27: Again, it would be relevant to check for possible contamination of their $^{15}N_2$ Eurisotop stock by $^{15}$N-labelled dissolved inorganic nitrogen (see Dabundo *et al*., 2014).

Page 14, line 22-23: The Berman-Frank paper was not submitted at the time of this review. Also, define PCD.

Page 16, lines 1-6: This paragraph is not clear. Do they mean the dead and live "swimmers" zooplankton were not distinguishable? Rewrite accordingly.

**References**

The following cited references were not accessible on the *Biogeosciences Discussion* online forum at the time of this review:
Berman-Frank *et al.* (This issue)
Bonnet *et al.* (This issue)
Bouruet-Aubertot *et al.* (This issue)
Caffin *et al.* (This issue)
Moutin *et al.* (This issue) – there is a Moutin *et al.* submitted but with a different title
Van Wambeke *et al.* (This issue)

**Tables**

Table 5: Include contributions from atmospheric depositions in this table.

**Figures**

Figure 3: Why PAR and DCM are decoupled at station LD B?

**Technical considerations:**

Review the manuscript for grammatical errors and typos. Here are a few examples:

Page 1, line 21: replace "Thanks to a Lagrangian…" for "Using a Lagrangian…".

Page 1, line 34: replace "while there contribution…" for "while their contribution…"

Page 9, line 13: replace for "… in-situ incubation method ranged from 0.0 – 19.3 nmol N $L^{-1}$ $d^{-1}$ …"

Page 10, line 11: replace by "was strongly influenced by the vertical diffusion coefficient".

Page 11, line11: replace "LDA" by "LD A".

Page 11, line 21: replace "whatever" for "regardless of", i.e., "… $N_2$ fixation was the major external source of N to the WTSP regardless of the degree of oligotrophy, …"

Page 12, lines 22-3: change for "… $NO_3^-$ input by turbulence always represented a minor contribution to the N budget."

Page 13, line 16: replace for "… and a clear dominance of …"

Page 14, lines 9-10: Add a period after "in oligotrophic open ocean regions." Start a new sentence with "To date, few qPCR *nifH* data from sediment traps are available…". Perhaps the proportion of dead versus live zooplankton could be estimated from the flask not filled with formaldehyde collected on the fifth day of sampling and used for diazotroph quantification.

Page 15, line 4: change for: "… in different oligotrophic regions of the ocean, for instance, the SPG …"

**Additional references:**

Cornell, S., Randell, A. and Jickells, T., 1995. Atmospheric inputs of dissolved organic nitrogen to the oceans. *Nature*, *376*(6537), pp.243-246.

Dabundo, R., Lehmann, M.F., Treibergs, L., Tobias, C.R., Altabet, M.A., Moisander, P.H. and Granger, J., 2014. The contamination of commercial $^{15}N_2$ gas stocks with $^{15}N$–labeled nitrate and ammonium and consequences for nitrogen fixation measurements. *PloS one*, *9*(10), p.e110335.

Knapp, A.N., DiFiore, P.J., Deutsch, C., Sigman, D.M. and Lipschultz, F., 2008. Nitrate isotopic composition between Bermuda and Puerto Rico: Implications for $N_2$ fixation in the Atlantic Ocean. *Global Biogeochemical Cycles*, *22*(3).

---

## Referee Comment (RC2) · Anonymous Referee #2 · 30 Nov 2017

Summary and Overall Impressions

This is a well-written, interesting, enjoyable paper that quantifies the source of "new" nitrogen to the euphotic zone as well as the flux of nitrogen derived from N2 fixation with respect to diazotroph community and other factors. Data was collected from three stations in the western tropical south Pacific ocean. Two of these stations were oligotrophic while the third was "ultra" oligotrophic. Nitrogen fixation was overwhelmingly the largest source of N input due in part to very high rates of nitrogen fixation and lower than typical rates of atmospheric deposition. N flux from nitrogen fixation was

uncoupled from N inputs, and possible reasons for this are well discussed.

This paper is well presented and the "story" is clearly told. Everything is well organized and easy to follow. The conclusions are interesting and important, and I really liked the 5 day averaging and well resolved vertical profiles. Daily variability is high and important to measure! In addition, the comparison between in situ and on deck incubations is useful and important, as most researchers are unable to use in situ arrays. The introduction and discussion sections are very well done, but I do have some concerns about some of the methods and the presentation of the results.

My primary concerns are methodological and regard 1) the nitrogen fixation rate calculations including the absence of a reported detection limit and a failure to completely address the failings of the "bubble method" and 2) propogation of error throughout calculations and some additional statistical comparisons. These issues can all using data already in hand, and I think that these changes are unlikely to impact the conclusions of the manuscript. Nevertheless statistical rigor and proper calculations are important. I have also listed some line item concerns below.

General Comments

Throughout the paper, nitrate transport is refered to as a "diffusive flux", but "diffusion" brings to mind molecular diffusion, whereas this flux is really a result of vertical mixing. Please rename this term or, if this is standard terminology, indicate clearly that this is not molecular diffusion but is a physical mixing process.

Regarding atmospheric deposition, was atmospheric deposition of NH4 considered? What about DON? - These would be "new" N that look like regenerated N. Also, throughout the manuscript, "atmospheric deposition" reflects dry deposition only (as opposed to wet). Please modify the text throughout so that this distinction is clear.

Some of the notation was a bit confusing. was used both for integrated N2 fixation rates (i-N2) and density and nitracline density (NOÂň3). I know that these are established

conventions, but use of the same abbreviation is a bit confusing and these should be clarified somehow.

The methods section could use a bit more detail and/or additional references. I have indicated specific problems below.

In section 2.1, a more clear description of which parameters led to the designation of the three stations would be useful, i.e. LD A was oligotrophic and Tricho dominated, etc. A map of station locations would be helpful, to be referenced on p. 4 lines 11-14, which could be Figure 1 or similar.

Regarding the nitrogen fixation rate calculations, while I understand the reluctance to add equilibrated seawater to the incubations, especially given the oligotrophic nature of the samples, the compromise proposed here (to measure 15N2 at the conclusion of the incubation) does not adequately reconcile the rates calculated here with the problems associated with the bubble method. It has been shown that the fraction label of the dissolved N2 "source pool" changes during the course of the incubation (Mohr 2010 and others). However, the method used in this manuscript assumes a steady fraction label of that pool, based on the fraction label measured at the end of the incubation, inevitably resulting in an underestimation of nitrogen fixation rates. This is problematic because a) different organisms fix at different times of day and b) the rate of change likely varies based on physical and chemical parameters. Problem (a) is especially relevant to this study as the researchers have specifically set out to compare regions with different diazotroph assemblages. At this stage, this problem cannot be addressed directly, but some discussion of the implications of this in light of the conclusions is warranted. Problem (b) can and should be directly addressed by correcting the rate calculations for the rate of bubble dissolution. Examples of this correction can be found in Figure 1 of Mohr (2010) and the Supplemental Figure in Jayakumar (2017). Since these were incubated on an in situ array, differences in temperature with depth could have variably impacted rates of 15N2 dissolution. This is addressed to some degree in those two publications and should be discussed in this manuscript.

Atmospheric deposition is really a flux to the mixed layer. Depending on mixed layer depth relative to the nitracline, mixing speeds, and biological uptake rates the upward flux of NO3 may reflect a flux to the sub-mixed layer euphotic zone only. Nitrogen fixation rates also show vertical structure. I assume that diazotroph communities have vertical structure (might be nice to put this on Figure 2 if these data were collected, as well as density to show mixed layer depth); Tricho floats (if much of the Tricho is floating on the surface, was it even sampled?). Diatoms can adjust their bouyancy. The 200 m over which everything is integrated is a fairly large region. How does one account for these mini-environments within that 200 m in the budget? Could the vertical structure in N fixation reflect higher NO3 concentrations (even though they're undetectable - but the d.l. in this study was pretty high for oligotrophic waters) closer to the nitracline? Some discussion of the vertical structure within the euphotic zone may be informative.

The abstract could use more contextual information and conclusions. In its current form it reads as a bit of a data dump. The interesting conclusions of the paper could be better showcased in this section.

Technical and Line Item Comments

Abstract

Line 20: "...all major fluxes..." could be "...all major vertical fluxes..."

Line 21: Instead of "Thanks to a Lagrangian..." this would be more clear as "Using a Lagrangian..."

Line 22: "...allowing to consider..." Should this read "allowing us to consider"?

Line 23: Might it be easier to refer to stations as A, B, and C without the 'LD', as it does not seem to have any necessary significance.

Line 26: The N2 fixation measurements appear to have been depth integrated (units umol N/m2/d), but what depth region were these integrated over? Also, what is meant by "extremely high"? In comparison to what, are these high?

Line 30-31: "N2 fixation...at all stations." Does this refer only to the LD A, B, and C stations? If not, please state how many stations were sampled in each region for these data.

Line 31: PC and PP have not been defined.

Line 34: "there contribution" should be "their contribution"

Line 36-37: "This disequilibrium...summer season..." I don't understand this sentence. Does this mean that this disequilibrium is generally held to be true (confirmed by other studies) or that you found this here and that it was consistent across all stations?

Line 37: The mention of zookplankton seems strange. I would delete it.

Introduction

p 2 Line 5: "di-nitrogen" Is this journal format? I have generally seen it written as "dinitrogen".

p 2 Line 5: "ammonia" At seawater and physiological pH, it is primarily "ammonium" that is present.

p 2 Line 7-11: "In the oligotrophic...photic layer." Run-on sentence.

p 3 Line 8: "(see below)" This is not necessary.

p 3 Lines 9-13: What region are these N2 fixation rates integrated over?

Materials and Methods

p 3 Line 22: "strong thermal stratification" Please provide data or a reference for this statement or delete it. What qualifies as "strong" thermal stratification?

p 3 Line 23: "...along a west-east...French Polynesia." Please also indicate the location using lat/long, as the precise locations of New Caledonia and French Polynesia do not spring immediately to mind.

p 3 Line 25: "diazotrophs" should be "diazotroph"

p 3 Line 29: "firstly" should be "first"

p 4 Line 19: Doesn't PAR stand for "Photosynthetically Active Radiation", not "Photosynthetically Available Radiation", as written here?

p 4 Line 19: Does "fluorescence" refer to chlorophyll a fluorescence, or some other set of wavelengths? Please be specific.

p 5 Line 3: "In situ Chl a concentration was..." Should be "In situ Chl a concentrations were..."

p 5 Line 3: Was the AquaTraka III an in situ sensor attached to and deployed with the CTD package? So, this was used instead of the SeaBird chl sensor? Or is the AquaTraka for shipboard measurements? This is a little unclear because I expected to see the SeaBird chl sensor used with the SeaBird package. Please clarify.

p 5 Line 4-7: Please include more details on collection of the NO3 and PO4 samples-were they filtered? Were they stored or run immediately at sea?

p 5 Line 9: Were these incubations in glass bottles? please indicate. How many replicates were used?

p 5 Line 13-17: Were these integrated over the upper 200 m? It looks like that's so, but please state it specifically.

p 5 Line 19: "DIN" should include ammonium, which can account for ∼40% of total N deposition (Dentener et al. 2006). I suggest renaming the combined NO3/NO2 term to NOx or similar to avoid confusion.

p 5 Line 19-22: Since the reference is "submitted", these methods should be explained in greater detail or additional references given.

p 5 Line 26: "same depths as for NPP" Were these 15N2 incubations performed in the

same bottles as the 14C incubations?

p 6 Line 4: The danger of "trace metal contamination" is mentioned. Was the water for incubations collected using trace metal clean methods?

p 6 Line 13: "blue screening" I assume that the purpose of blue screening was to alter the quality of incident light on the bottles. Please provide additional information on the change in quality to the incident light achieved with the blue screening. Also, was the quantity of incident light altered (i.e. by using different sizes mesh screen or some other neutral density filter) in the deckboard incubators?

p 6 Line 16: What is meant by "gentle filtration"?

p 6 Line 21-22: Why was the initial PN only measured at two depths rather than for each rate measurement? Has it been determined that two depths are sufficient? If so, please be specific. How exactly were these two measurements used as 'initial' measurements for each rate measurement? Were they averaged and then used for all rates at that stations or for all stations or some other method? please specify. How many replicates were collected per depth for the initial measurements?

p 6 Line 25: How was the nitracline depth calculated? I am unclear on what NO3 is. Is this the density where the nitracline occurs?

p 6 Line 25- p 7 Line 3: Is there a reference for this flux calculation?

p 7 Line 6: Why were these depths chosen?

p 7 Line 8: "...buffered solution of formaldehyde..." Please include a reference for this statement.

p 7 Line 12: Please specify what a "swimmer" is. Were these living organisms found in the 'fresh' trap, or were they in the preserved samples too? If the latter, how were they separated from the rest of the material? Were these a certain size class of organisms? Why were they analyzed separately?

p 8 Line 6: What is meant by "gently filtered"?

p 8 Line 11-12: "The C contents...Luo et al. 2012)." This belongs in the results section.

p 8 Line 11-16: "As DDAs...asymbiotic)." It is unclear why this information is present; I thought the cell C content was only determined for Tricho and UCYN. Please add a topic sentence to this paragraph explaining for which organisms biovolume and cell C content were determined and which were determined directly v indirectly.

Results

p 9 Line 2: "...with the minimum concentration located at 60 m." It's best to refrain from calling this a minimum because the difference in concentration is slight and there are no replicates.

p 9 Line 5: "...while the DCM was deepening from 25 m to 70 m..." I suggest restating this as "...the depth of the DCM increased from 25 to 70 m during the five days that the station was occupied..." or something similar

p 9 Line 6: "...varying simultaneously between 115 and 155 m." This wording is confusing.

p 9 Line 11: "...below 0.2..." Is it below 0.2 or equal to 0.2, as Table 4 indicates? Where is the comparison of total N deposition and $NO_3+NO_2$ deposition, as suggested in the Methods?

p 9 Line 13: "...0.0-19.3..." What was the detection limit of this method? Instead of reporting "0.0" please report rates as below the limit of detection if they are. Two recent publications depict methods for calculating the detection limit of these rates (Gradoville et al. 2017; Jayakumar et al. 2017). Accounting for a DL will be especially important for LDC where the rates were very low and may be undetectable.

p 9 Line 17: "...below the quantification limit..." So there is a quantification limit! How was it calculated and what was it?

p 9 Line 17-19: Please compare the N2 fixation rates from days 1-3 using the Mann Whitney test, as was done comparing the in situ and shipboard incubations.

p 9 Line 23: "...the maximum at 40 mat LDC..." Was this small rate actually above the detection limit though?

p 9 Line 24-27: It is nice to see this comparison! It makes me feel better about all the shipboard measurements in the literature!

p 9 Line 31-p10 Line 1: "Strong time...(table 1)." This is repetitive with the previous sentence.

p 10 Line 22: "...maximum export...at LD C." PC export for LDB and LDC do not look significantly different, based on the overlap in standard deviation (Table 2).

p 10 Line 21-24: Why is there a discussion of trends in PC but not PN and PP? Line 24: "...averaged C:N ratios...LDC." Are these significantly different?

p 10 Line 31: "below detection limit" What was the detection limit? Elsewhere in the paper, (i.e. for NO3 and PO4 concentrations), a "quantification limit" is referenced. What is the difference between that quantification limit and a detection limit? Please define these terms in the Methods section and be consistent with their usage.

Discussion

p 11 Line 27: "...atmospheric deposition...lower end of fluxes..." I am curious what the mixed layer depth was at these stations, as atmospheric deposition is really a flux only to the mixed layer, not the entire euphotic zone. Since the flux is so small, it likely doesn't matter for this study, but it would be nice to see those depths in a Table for readers who study atmospheric deposition to the mixed layer.

p 12 Line 8-17: "In this study...to significant bias." This is a really good point, and it's nice to see a snapshot of this variability. It does make one wonder, however, if the snapshot had been even longer, how would the results have varied, given the large

differences seen between the use of an average Kz and the instantaneous Kz values and the seeming randomness of the spikes. Is 5 days long enough? A sentence or two addressing this concern would be useful.

p 12 Line 22: "N m-4" What is a m-4?

p 15 Line 11: "such as DON and/or zooplankton export" Please elaborate. Dissolved compounds are not exported like particles are; they do not sink. Or is this a reference to conversion of fixed N to the DON pool, followed up uptake and subsequent export? Both a direct DON flux and a delayed N flux following DON uptake are worth mentioning. Also, what about fixed N that is released as ammonium? Ammonium and many simple DON compounds (amino acids, urea...) cycle very quickly and likely would be taken up before they could be mixed downward. Also, it is unclear why DON and zooplankton export are lumped together like this... Is this referring to an active zooplankton flux (which is a completely different process than the DON pathway) or to the sinking out of dead zooplankton later?

p 16 Line 1: "zooplankton itself is" should be "zooplankton themselves are" Please make similar corrections throughout.

Conclusion

p 16 Line 16-17: "contributed to ∼15-21%" and "...and to 3%..." Delete "to".

Tables and Figures

The tables seem to be out of the order that they are mentioned in the text. Please check this.

Table 1. Please define Kz in the figure legend. For this and all figures and tables, please indicate what the ± error is. Standard deviation? Standard error? n = ?

Table 2. What is "DW matter"? Please indicate this in the table legend or with a footnote to the table. Why are the errors italicized in this table and Table 3 but not in Table 1?

Please standardize this across tables. If PC, PN, and PP should be PC flux, PN flux, and PP flux, please indicate that. Please propogate the error of PC, N, and P into the C:N:P ratio calculation.

Table 3. Please apply the comments from Tables 1 and 2 to this table. Also, please use a standard number of significant digits for all measurements (i.e. LD A 300 m is inconsistent).

Table 4: Please include the standard deviation or error of these values by propogating the error from the measurements. please define all non-obvious terms (i.e. d[NO3]/dp, NO3) and the calculation for the e-ratio in a footnote so that the reader does not have to dig through the text to find them. Is N2 fixation the in situ rate or the shipboard rate? Please use consistent significant digits (i.e. for NO3 diffusion and export N 150 m).

Fig 1: This figure seems needlessly complicated and the vertical component does not seem spaceworthy. The advantage of the vertical component is to show where the production arrays and traps were deployed, but it's difficult to tell the exact depths in this figure. A simple map may be preferable.

Fig. 2: Please indicate on the figure itself which units correspond to which parameter, as it is a bit confusing in current form (particularly for phosphate). It may be instructive to use a different scale for the N fixation rates in the third panel, since they can't be seen on this scale.

Fig 3: This is a cool figure! Nice dataset! I am a little confused on how 1% of surface PAR was calculated at night. Should there be breaks in the dotted line for nighttime?

References

Dentener, F., J. Drevet, J. F. Lamarque, I. Bey, B. Eickhout, A. M. Fiore, D. Hauglustaine, L. W. Horowitz, and others. 2006. Nitrogen and sulfur deposition on regional and global scales: A multimodel evaluation. Global Biogeochem. Cycles. 20, GB4003, doi:10.1029/2005GB002672:

Gradoville, M. R., D. Bombar, B. C. Crump, R. M. Letelier, J. P. Zehr and A. E. White 2017. Diversity and activity of nitrogen‐fixing communities across ocean basins. Limnol. Oceanogr. 62: 1895-1909

Jayakumar, A., B. X. Chang, B. Widner, P. Bernhardt, M. R. Mulholland and B. B. Ward 2017. Biological nitrogen fixation in the oxygen-minimum region of the eastern tropical North Pacific ocean. The ISME Journal. 11: 2356-2367

Mohr, W., T. Grosskopf, D. W. Wallace and J. LaRoche 2010. Methodological underestimation of oceanic nitrogen fixation rates. PLOS one. 5: e12583

---

## Referee Comment (RC3) · C. Löscher (Referee) · 4 Dec 2017

The manuscript by Caffin et al. describes budgets of nitrogen at three stations in the oligotrophic western tropical South Pacific using a Lagrangian strategy thus being able to track the same water mass over time. The study reports exceptionally high N2 fixation rates and a corresponding high contribution of N2 fixation impacted material to export production. The study is very interesting to me particularly because of an approach that is more innovative than what is classically used when it comes to N budgets and N2 fixation. Overall, the paper doesn't need much changes to get into

shape for publication, the study is clear and well presented. I personally think the title is not the best choice, it could make a statement on what the prominent role of N2 fixation is.

In order to make the study entirely convincing I have some main aspects, which should be and easily could be clarified:

1. The good old topic on using the bubble method: It is not convincing to just measure the dissolved fraction and not give any ranges. There are concerns with that method, everyone knows that, if you claim it is ok to use it you should have done a comparative measurement at least for some of your samples using both methods. In this context, I either need to see the data on the dissolved vs. particulate phase, or the rates have to be presented as potential rates.

2. In the same context, I don't know the gas quality of the company you bought from, but I assume you checked for purity as recommended in the Dabundo paper. Otherwise the high rates may as well come from an ammonia incorporation or similar. Please present your quality check, here.

3. In addition, ammonia background measurements, fluxes and inputs are not mentioned- this would add enormous value to the stud, so please present if available. As you are making a suggestion on zooplankton moderated export, ammonia is a good part of this, too.

4. No sequencing was performed and no single cell rates were determined- how can you interpret on the key N2 fixers if you just look at 6 clusters via qPCR? What makes you conclude that Trichodesmium or UCYN clusters are important if you don't assess which diazotrophs are there?
* * *

---

## Referee Comment (RC4) · Anonymous Referee #4 · 7 Dec 2017

**Summary Statement**

Caffin et al. constructed a nitrogen budget for three stations in the western tropical Pacific Ocean by quantifying N2 fixation, NO3 diffusion, atmospheric deposition, and PN export. Overall, the study seems to be well-conducted, arguments are supported by data, and the paper is well-cited. There are some relatively minor issues, mostly with the presentation, as described below. The manuscript requires a thorough editing to correct awkward word choices, punctuation errors, and confusing text. The main point I found that was missing from the paper was a definition of the system being studied.

When the authors attempted to describe the system and site selection choices, the text was confusing and too vague, so this area of the paper could be improved. Some additional details are also missing from the methods and should be included. The conclusions section fell a bit flat and could be bolstered by putting the study findings into a better context relative to filling information and data gaps and describing the overall importance of the study results for our understanding of the global ocean. None of these issues represent serious barriers to publication, in my view, and only minor revisions are needed.

**Specific Comments**

Abstract — Overall, I found the Abstract was confusing. There is no clear direction, and the text jumps around from topic to topic without any clear context for the study or results. The concluding sentences do not place the study findings into any sort of importance relative to information and data gaps that we have for the WTSP (or other areas of the oligotrophic ocean). Why is the disequilibrium and apparent N accumulation important to describe?

P1, Lines 21-22 — Confusing sentence. Rewrite for clarity.

P1, Lines 24-25 — Is there more information on these locations, other than just DCM, that could be presented to give the reader a better idea of what these sampling locations are like?

Introduction

P3, Lines 1-8 — The authors need to define the "system" they are talking about. What are the boundaries of the "system"? Are sediments included? What does "...with an adequate time frame under contrasting diazotroph communities' composition" mean? Does "the same water mass" mean that horizontal water movement is not present/considered? Are there processes occurring within (or beyond) the boundaries of this "system" that could confound the approach?
P3, Lines 9-16 — The authors should provide more information on the trophic gradient and how 'oligotrophic' and 'ultra-oligotrophic' are defined. What are the physical factors causing the gradient?

P3, Lines 17-19 — The points of focus are great, but were there hypotheses to be tested? Why was it important to focus the study on these three points? What information/data gaps were being filled by conducting the study?

Methods

P3, Line 27 - P4, Line 14 — There are not enough details on the 3 criteria for site selection. What were the parameters of "local minima of surface current intensity" used to determine if conditions were suitable? How much surface current was considered acceptable? Were deeper currents considered? How was trophic status defined? In terms of chlorophyll or something else? If so, what were the thresholds used for oligotrophic, ultra-oligotrophic, etc.?

P5, Line 3 — Was the chlorophyll fluorescence sensor calibrated to simultaneous samples analyzed for chlorophyll using more conventional extraction techniques?

P5, Lines 4-7 — Were nutrient samples analyzed immediately, or filtered and stored for analysis later (if so, provide details on procedures used), or not filtered at all...? Why wasn't ammonium included in the nutrient measurements?

P5, Lines 9-17 — It is unclear where the "associated N uptake" part of this section is evaluated. More details are needed describing sample handling and analyses for the PP incubations.

P5, Lines 19-22 — More details are needed on the aerosols sampling, especially since the reference given for the method is only a submitted paper. Is there a reason why ammonia was not included in the atmospheric deposition measurements?

P5, Line 29 – P6, Line 3 — Very confusing sentence. Rewrite for clarity.

BGD
P6, Lines 7-10 — How were dissolved gas samples transferred from the bottles to Exetainers? Kana et al. (1994) does not cover 15N2 measurements/analyses using MIMS. Is there another citation for the 15N2 analyses using MIMS?

P7, Lines 1-3 — perhaps add "and" before daily? Something is missing in this sentence.

P7, Lines 12-15 — Define "swimmers". PP was previously defined as primary production, so also using it for particulate phosphorus is confusing.

**Results & Discussion**

P9, Lines 24-27 — What was the integration depth used for these rates? It is odd to see areal rates reported for a depth-integration that apparently does not include sediments.

P9, Line 30 — Is there really a strong contrast between LD A and the other two stations given the very large variability around the mean at LD A (24.4  $\pm$  24.4)?

P11, Line 11 — Perhaps the authors should use LD-A, LD-B, and LD-C to denote their stations, instead of LD A, LD B, and LD C. There have been a few cases like here (LDA) where the site abbreviations have not been consistent.

P12, Lines 3-23 — I found this narrative confusing. Perhaps the authors could streamline this text to focus it on the most important points?

- P13, Line 13 (and elsewhere) primary production or particulate phosphorus?
- P13, Lines 15-16 correct these scientific notations for gene copies
- P13, Lines 20-21 Why use the areal rates here instead of the volumetric rates?
- P13, Line 32 P14, Line 2 Awkward sentence. Rewrite for clarity.
- P14, Lines 7-14 Confusing text. Rewrite for clarity.

P14, Line 22 — What is "PCD"?

BGD
P16, Lines 12-26 — The conclusion section is a little flat. The authors could do a better job of placing their study into a better context in terms of the global N budget and C export in the oceans.

Table 1 — Are these Kz values supposed to be in scientific notation? Are the units for the nitracline correct?

**Technical Corrections**

- P1, Line 24 add "respectively" after "LD B"
- P1, Line 28 add a comma after "LD C"
- P1, Line 31 PC and PP not defined (or PN earlier)
- P1, Line 34 change "there" to "their"
- P2, Line 5 add comma after "ammonia"
- P2, Line 7 "before" is an awkward word choice
- P2, Lines 7-11 Long, run on sentence. Rewrite for clarity.
- P2, Line 21 add comma after "...et al., 2008)"
- P2, Line 24 add comma after "large"
- P2, Line 25 add comma after "phytoplankton"
- P2, Line 29 add comma after "ocean"
- P3, Line 9 "harbouring" is an awkward word choice
- P3, Lines 10-15 Long, run on sentence. Rewrite for clarity.
- P3, Line 17 add comma after "study"
- P4, Line 22 "every day"

BGD
P7, Line 13 — "weighed"

P9, Line 13 — add "from" after "ranged"

- P10, Line 32 "2.67 x 104"
- P11, Line 2 change "from" to "for"

The paper requires a thorough editing for grammar, word choice, and punctuation.

---

## Author Comment (AC1) · 23 Jan 2018

We thank Anonymous Referee #1 for the time and effort devoted to the review of the manuscript. Below, we reproduce the reviewer's comments and address their concerns point by point. The reviewer's comments are copied below in regular font with our responses in blue.

We are responding to this review a long time after it was published on the *Biogeosciences Discussion* online forum because we had to submit the companion paper by Caffin et al. (this issue), cited in this article, before the closure of OUTPACE's special issue on December 31, 2017.

**General comments:**

In their study, Caffin *et al*. use a Lagrangian approach to determine N sources ($N_2$ fixation rates, $NO_3^-$ supply from vertical diffusion and N from atmospheric depositions) and sinks (particulate N export) at 3 stations in the Western South Tropical Ocean. They also measured primary production using $^{14}C$ tracer incubations. Their main findings are that 1) $N_2$ fixation is the major source of new N (>90%) at all stations, regardless of whether the diazotroph community was dominated by *Trichodesmium* or unicellular cyanobacteria, 2) carbon export relative to primary production is high in this region, and 3) the sum of N input in the photic zone exceeds particulate N export. Overall, this study is interesting and timely as this region has recently been identified as a $N_2$ fixation hotspot. However, I have some concerns. First, they report a low input from atmospheric depositions but did not consider atmospheric sources other than $NO_3^-$ and $NO_2^-$ (e.g., $NH_4^+$ DON, PON). They also measured relatively high $N_2$ fixation rates and I am wondering if they considered the possibility of a contamination of their $^{15}N$-$N_2$ stock with $^{15}N$-$NO_3^-$ and $NH_4^+$, which would artificially increase their $N_2$ fixation rates, as recently reported by Dabundo *et al*. (2014). Some of the references cited (this issue) were unavailable on the *Biogeosciences Discussions* online forum at the time of this review, making it impossible to evaluate these parts of the manuscript. I was also a bit confused regarding the novelty of their dataset: were the same $N_2$ fixation rates, qPCR or any other data collected at the same stations during the OUTPACE cruise already published in previous studies? The authors should make a clear distinction of the new data contributed by their study versus the data already published elsewhere in other manuscripts in the special issue.

Reviewer #1 is right that we did not consider atmospheric sources other than $NO_3^-$ and $NO_2^-$. Our flux could be underestimated as it represents dry deposition and because gas and organic forms were not measured. At global scale, and depending on the location, organic nitrogen could represent up to 90 % of N atmospheric deposition (Kanakidou et al., 2012). Even if we double our estimated deposition flux, atmospheric deposition remained low (< 1.5 %) and consequently represented a minor contribution of the new N input. We are aware that Dabundo et al. (2014) report potential contamination of some commercial $^{15}N_2$ gas stocks with $^{15}N$-enriched $NH_4^+$, $NO_3^-$ and/or $NO_2^-$, and nitrous oxide ($N_2O$). To verify this, the Cambridge Isotopes batches that are routinely used by our team has been analyzed for potential contamination in Julie Granger and Richard Dabundo's lab, and this confirmed that the contamination of the $^{15}N_2$ gas stock was low: 1.4 x $10^{-8}$ mol of $^{15}NO_3^-$ per mol of $^{15}N_2$, and 1.1x$10^{-8}$ mol $NH_4^+$ per mol of $^{15}N_2$. The application of this contamination level to our samples using the model described in Dabundo et al. (2014) indicates that our rates could only be overestimated by 0.01 to 0.12 %. We thus confirmed that the stock contamination issue did not affect the results reported here. We are aware that some of the references cited here (this issue) were unavailable on the *Biogeosciences Discussions* online forum at the time of this review, but most of them (except Bouruet-Aubertot et al., this issue) are available now. The objective

of the OUTPACE special issue is to provide a unique opportunity for a group of researchers to focus on the "Interactions between planktonic organisms and biogeochemical cycles across trophic and $N_2$ fixation gradients in the western tropical South Pacific Ocean". It is a multidisciplinary approach with a tight time schedule, and with the main aim of sharing the data in order to provide best study in a relatively short time of a new (exceptional) set of data. The data may be used several times in different papers of the special issue focusing on different scientific questions. In this case, the method is given with all details only in one paper that is clearly referenced as the main one for the method.

Kanakidou, M., Duce, R. A., Prospero, J. M., Baker, A. R., Benitez-Nelson, C., Dentener, F. J., ... & Sarin, M. Atmospheric fluxes of organic N and P to the global ocean. Global Biogeochemical Cycles, 26(3), 2017

**Specific comments:**
**Title**
The title is a bit long and not focused on the main point of the study. I suggest changing for: "$N_2$ fixation as the dominant new N source in the Western Tropical South Pacific Ocean (OUTPACE cruise)"

We have chosen another title containing the Lagrangian term, but because two reviewers suggested the same change, we have changed the title in accordance with this suggestion.

**Introduction**
Page 2, line 15: Knapp *et al.* (2008) and Bourbonnais *et al.* (2009) also observed a low $\delta^{15}N$ of $NO_3^-$ (relative to $\delta^{18}O$-$NO_3^-$) in surface waters in the western and eastern subtropical Atlantic Ocean, supporting the role of $N_2$-fixers in these regions.

We have added these missing references in accordance with this suggestion

Page 4, lines 6: What factors influence the distribution of *Trichodesmium* or UCYN? I believe temperature is an important factor (see Moisander *et al*., 2010). This point should be discussed a bit more.

We are aware that the activity and distribution of diazotrophs have been hypothesized to be controlled by several environmental variables in the open ocean, such as light (Fu and Bell, 2003; Breitbarth et *al*., 2008; Levitan et *al*., 2010), temperature (Capone et *al*., 1997; Staal et *al*., 2003; Breitbarth et *al*., 2007; Moisander et *al*., 2010) or nutrient availability (Van den Broeck et *al*., 2004; Moutin et *al*., 2005; Mills et *al*., 2004; Ho, 2013). However, we did not focus on this in our study. In the context of the OUTPACE project, Bonnet et al. (this issue) have shown a correlation between diazotrophs abundance and temperature. A partition of niches between *Trichodesmium* and UCYN has been observed in this region (Moisander et al., 2010; Bonnet et al., 2015). Furthermore, this interesting scientific question has been studied in the WTSP by Stenegren et al. (this issue), who mentioned that the correlations between the different diazotroph communities and the environmental conditions observed in the WTSP were not always consistent with the meta-analysis of the external datasets.

**Experimental procedures**

Page 5, line 19: They only considered NO3- and NO2- when quantifying N atmospheric depositions. They should also consider NH4+ or organic nitrogen (particulate or dissolved). For instance, Cornell *et al*. (1995) estimated that organic nitrogen was a significant component of atmospheric N depositions even in remove marine regions.

> **We understand and accept this suggestion that $NH_4^+$ and organic N (particulate and dissolved) should also be considered when quantifying atmospheric deposition. Our quantification of the atmospheric deposition could be underestimated as it only represents dry deposition, and gas and organic forms were not measured. At global scale, and depending on the location, organic nitrogen could represent up to 90 % of N atmospheric deposition (Kanakidou et al., 2012). Even if we double our estimated deposition flux, atmospheric deposition remained low (< 1.5 %) and consequently represented a minor contribution of the new N input.**

Page 5, line 26: Did they check their commercial 15N Eurisotop gas for possible contamination with 15N-labeled dissolved inorganic nitrogen (NO3-, NO2- and NH4+)? Microbial assimilation of contaminant 15N labeled dissolved inorganic nitrogen would artificially increase N2 fixation rates. Dabundo *et al*. (2014) recently reported significant concentrations of 15N contaminants in 15N-labelled N2 gas supplied by Sigma-Aldrich and Campro Scientific.

> **We are aware that Dabundo et al. (2014) reports potential contamination of some commercial $^{15}N_2$ gas stocks with $^{15}N$-enriched $NH_4^+$, $NO_3^-$ and/or $NO_2^-$, and nitrous oxide ($N_2O$). In their study, Dabundo et al. (2014) analysed various brands of $^{15}N_2$ (Sigma, Cambridge Isotopes, Campro Scientific) and found that the Cambridge Isotopes brand (i.e., the one used in these studies) contained low concentrations of $^{15}N$ contaminants, and the potential overestimated $N_2$ fixation rates modeled using this contamination level would range from undetectable to 0.02 nmol N $L^{-1}$ $d^{-1}$. The rates measured in this study were on average ~10 nmol N $L^{-1}$ $d^{-1}$, suggesting that stock contamination would be too low to affect the results reported here.**
>
> **To verify this, the Cambridge Isotopes batches that are routinely used by our team has been analyzed for potential contamination in Julie Granger and Richard Dabundo's lab, and this confirmed that the contamination of the $^{15}N_2$ gas stock was low: 1.4 x $10^{-8}$ mol of $^{15}NO_3^-$ per mol of $^{15}N_2$, and 1.1x$10^{-8}$ mol $NH_4^+$ per mol of $^{15}N_2$. The application of this contamination level to our samples using the model described in Dabundo et al. (2014) indicates that our rates could only be overestimated by 0.01 to 0.12 %. We thus confirmed that the stock contamination issue did not affect the results reported here.**

Page 6, lines 5-7: A better way to assess whether equilibration was complete would be to try different treatments in triplicate, i.e., shake the bottles for different times and intensity before the *in-situ* incubations.

> **We agree that it would be a useful complementary study to try different treatments in triplicate in order to better assess whether the equilibration was complete. In the present study, we performed MIMS analyses to quantify the final $^{15}N$-enrichment in the $N_2$ pool and avoid any error due to an insufficient equilibration process.**

Page 6, line 6: Add "incomplete" before equilibration.

**This has been corrected in the new version**

Page 6, lines 7-10: I assume the 12 mL subsample was collected without contact with the atmosphere?

**The 12 mL subsample was collected rapidly from the 4.5 L to the Exetainers with contact with the atmosphere. We assume that the sampling was rapid enough to avoid sea-air exchanges that can affect the $^{15}N_2$ enrichment of the sample. In any case, this is more precise than using the theoretical $^{15}N_2$ enrichment, as revealed in Bonnet et al. (this issue).**

Page 6, lines 16-24: What is the detection limit for their N2 fixation rates?

**The minimum quantifiable rate calculated using standard propagation of errors via the observed variability between replicate samples, measured according to Gradoville et al. (2017), was 0.035 nmol N $L^{-1}$ $d^{-1}$. This has been specified in the new version of the paper.**

**Gradoville, M. R., D. Bombar, B. C. Crump, R. M. Letelier, J. P. Zehr and A. E. White Diversity and activity of nitrogen fixing communities across ocean basins. Limnol. Oceanogr. 62: 1895-1909, 2017**

Page 7, line 21: Define UCYN-B, UCYN-A1, het-1 and het-2. Which bacteria are represented by these different groups?

**The sentence has been changed to "Diazotroph abundance for *Trichodesmium* spp., UCYN-B (*Crocosphaera watsonii*), UCYN-A1 (*Candidatus Atelocyanobacterium thalassa*), het-1 (*Richelia intracellularis* and *Rhizosolenia*), and het-2 (*Richelia intracellularis* and *Hemiaulus*) were quantified by qPCR analyses on the nifH gene using previously described oligonucleotides and assays (Foster et al. 2007; Church et al. 2005)."**

Page 8, line 1: Were samples with qPCR reaction efficiency below 95% reported? Why not repeat analysis for these samples?

**The samples with qPCR reaction efficiency below 95% were excluded. As the amount of water was restricted for each parameter measurement, we were not able to repeat the analysis for these samples.**

**Results**

Page 9, lines 24 to 27: How are these rates different from the one measured in Bonnet *et al*., 2017 (This issue). Are they the same rates as reported in Bonnet *et al*. (2017)? The Bonnet *et al.* paper was not yet available at the time of this review, making it impossible to effectively evaluate this part of the manuscript.

**We understand that it was impossible for Reviewer #1 to evaluate this part of the manuscript as the Bonnet et al. paper was not available at the time of the review. The Bonnet el al. paper is now available in the *Biogeosciences Discussion* online forum. Bonnet et al. (this issue) reports rates from on-deck incubation at 15 short duration stations (that were not included in this study), and those of the 3 LD stations (presented here). Thus, the Bonnet et al. dataset is different and presented in a totally different perspective.**

Page 11, line 11: Were *Trichodesmium* data for LD A (150 m) not available or below detection limit (as stated on page 10, line 32)?

**Trichodesmium data for LD A (150 m) were not available as the qPCR efficiency was below 95 %. This has been changed on page 10 line 32 to: "… sediment material (< QL at LD C 330 m and not available at LD A 150 m) and …"**

**Discussion**

Page 11, lines 27-29: Perhaps atmospheric deposition measured during OUTPACE are low because they neglected contributions from organic nitrogen and NH4+. This possibility should be discussed (see my previous comment, page 5, line 19).

**We agree with this comment of Reviewer #1, thus we have discussed the possibility of underestimation of the atmospheric input in the new version of the manuscript as follows: "Extrapolated NOx deposition from the atmosphere during OUTPACE (range: $0.34 – 1.05$ μmol m$^{-2}$ d$^{-1}$) were one order of magnitude lower than predicted with major uncertainties by global models that include wet and gas deposition for that region (Kanakidou et al. (2012). Our flux could be an underestimation as it represents dry deposition and gas and organic forms were not measured. At global scale and depending on the location, organic nitrogen could represent up to 90 % of N atmospheric deposition (Kanakidou et al., 2012). Even if we double our estimated deposition flux, atmospheric deposition remained low (< 1.5 %) and consequently represented a minor contribution of the new N input (Table 4)"**

Page 12, lines 24-27: Again, it would be relevant to check for possible contamination of their 15N2 Eurisotop stock by 15N-labelled dissolved inorganic nitrogen (see Dabundo *et al*., 2014).

**As previously explained, to verify this, the Cambridge Isotopes batches that are routinely used by our team has been analyzed for potential contamination in Julie Granger and Richard Dabundo's lab and this confirmed that the contamination of the $^{15}N_2$ gas stock was low: $1.4 \times 10^{-8}$ mol of $^{15}NO_3^-$ per mol of $^{15}N_2$, and $1.1 \times 10^{-8}$ mol $NH_4^+$ per mol of $^{15}N_2$. The application of this contamination level to our samples using the model described in Dabundo et al. (2014) indicates that our rates could only be overestimated by 0.01 to 0.12 %. We thus confirmed that the stock contamination issue did not affect the results reported here**

Page 14, line 22-23: The Berman-Frank paper was not submitted at the time of this review. Also, define PCD.

**The Berman-Frank paper has been replaced by Spungin et al. (This issue), available in the *Biogeosciences Discussion* online forum at this time.**

**PCD (Programmed cell death) has been defined in the new version of the paper.**

**Spungin, D., Belkin, N., Foster, R., Stenegren, M., Caputo, A., Pujo-Pay, M., Leblond, N., Dupouy, C., Bonnet, S., and Berman-Frank, I.: Programmed cell death in diazotrophs and the fate of organic matter in the Western Tropical South Pacific Ocean during the OUTPACE cruise, Biogeosciences Discuss., in review, 2018.**

Page 16, lines 1-6: This paragraph is not clear. Do they mean the dead and live "swimmers" zooplankton were not distinguishable? Rewrite accordingly.

**This paragraph has been rewritten to explain more clearly that the dead and live "swimmers" zooplankton were not distinguishable.**

**References**

The following cited references were not accessible on the *Biogeosciences Discussion* online forum at the time of this review:

Berman-Frank *et al.* (This issue)
Bonnet *et al.* (This issue)
Bouruet-Aubertot *et al.* (This issue)
Caffin *et al.* (This issue)
Moutin *et al.* (This issue) – there is a Moutin *et al.* submitted but with a different title
Van Wambeke *et al.* (This issue)

**To date, the Bouruet-Aubertot et al. (this issue) paper is not available yet and the Berman-Frank et al. (this issue) paper was submitted as Spungin et al. (this issue); all the other references are available in the *Biogeosciences Discussion* online forum.**

**Tables**

Table 5: Include contributions from atmospheric depositions in this table.

**In most of the studies presented in Table 5, the N input associated to atmospheric deposition was not quantified. Only in the Mediterranean Sea, which is strongly affected by dust deposition, the atmospheric deposition was measured. Atmospheric models and global tropospheric budgets were performed (Duce et al., 2008; Kanakidou et al., 2012) that can give us a global picture of atmospheric deposition, however to be consistent with our study we have decided not to include those data in Table 5. In addition, the main objective of this table is to compare the contribution of $N_2$ fixation as a new N input in different regions of the world Ocean.**

**Figures**

Figure 3: Why PAR and DCM are decoupled at station LD B?

**The decoupling between PAR and DCM observed at LD B remains in significant chlorophyll a concentration observed at the surface that was stirred, deformed and transported by the mesocscale circulation, as explained in de Verneil et al. (this issue) who focused their study on the significant surface chlorophyll a bloom sampled at LD B.**

**de Verneil, A., Rousselet, L., Doglioli, A. M., Petrenko, A. A., and Moutin, T.: The fate of a southwest Pacific bloom: gauging the impact of submesoscale vs. mesoscale circulation on biological gradients in the subtropics, Biogeosciences, 14, 3471-3486, https://doi.org/10.5194/bg-14-3471-2017, 2017.**

**Technical considerations:**

Review the manuscript for grammatical errors and typos. Here are a few examples:

Page 1, line 21: replace "Thanks to a Lagrangian…" for "Using a Lagrangian…".

**This has been corrected in the new version**

Page 1, line 34: replace "while there contribution…" for "while their contribution…"

**This has been corrected in the new version**

Page 9, line 13: replace for "… in-situ incubation method ranged from 0.0 – 19.3 nmol N $L_{-1}$ $d_{-1}$

…"

**This has been corrected in the new version**

Page 10, line 11: replace by "was strongly influenced by the vertical diffusion coefficient".

**This has been corrected in the new version**

Page 11, line11: replace "LDA" by "LD A".

**This has been corrected in the new version**

Page 11, line 21: replace "whatever" for "regardless of", i.e., "… $N_2$ fixation was the major external source of N to the WTSP regardless of the degree of oligotrophy, …"

**This has been corrected in the new version**

Page 12, lines 22-3: change for "… $NO_{3-}$ input by turbulence always represented a minor contribution to the N budget."

**This has been corrected in the new version**

Page 13, line 16: replace for "… and a clear dominance of …"

**This has been corrected in the new version**

Page 14, lines 9-10: Add a period after "in oligotrophic open ocean regions." Start a new sentence with "To date, few qPCR *nifH* data from sediment traps are available…".

**This has been corrected in the new version**

Perhaps the proportion of dead versus live zooplankton could be estimated from the flask not filled with formaldehyde collected on the fifth day of sampling and used for diazotroph quantification.

**Reviewer #1's suggestion is interesting as a way to estimate the proportion of dead versus live zooplankton. In fact, all the zooplankton (dead and alive) were recovered from the flask filled with formaldehyde, while only dead zooplankton could be recovered from the flask that was not filled with formaldehyde. However, zooplankton were not recovered from the poisoned and unpoisoned flask on the same day; that is a problem, as we observed daily variability in the amount of zooplankton recovered from the flask.**

Page 15, line 4: change for: "… in different oligotrophic regions of the ocean, for instance, the SPG …"

**This has been corrected in the new version**

---

## Author Comment (AC3) · 23 Jan 2018

**Response to Carolin Löscher (Referee #3)**

We thank Carolin Löscher for the time and effort devoted to the review of the manuscript. Below, we reproduce the reviewer's comments and address her concerns point by point. The reviewer's comments are copied in regular font, with our responses in green.

We are responding to this review a long time after it was published on the *Biogeosciences Discussion* online forum because we had to submit the companion paper by Caffin et al. (this issue), cited in this article, before the closure of OUTPACE's special issue on December 31, 2017. We recently obtained a 3 month extension.

The manuscript by Caffin et al. describes budgets of nitrogen at three stations in the oligotrophic western tropical South Pacific using a Lagrangian strategy thus being able to track the same water mass over time. The study reports exceptionally high N2 fixation rates and a corresponding high contribution of N2 fixation impacted material to export production. The study is very interesting to me particularly because of an approach that is more innovative than what is classically used when it comes to N budgets and N2 fixation. Overall, the paper doesn't need much changes to get into shape for publication, the study is clear and well presented. I personally think the title is not the best choice, it could make a statement on what the prominent role of N2 fixation is.

In accordance with the comments by Anonymous Referee #1, we have changed the title to "N$_2$ fixation as a dominant new N source in the Western Tropical South Pacific Ocean (OUTPACE cruise)"

In order to make the study entirely convincing I have some main aspects, which should be and easily could be clarified:

1. The good old topic on using the bubble method: It is not convincing to just measure the dissolved fraction and not give any ranges. There are concerns with that method, everyone knows that, if you claim it is ok to use it you should have done a comparative measurement at least for some of your samples using both methods. In this context, I either need to see the data on the dissolved vs. particulate phase, or the rates have to be presented as potential rates.

The method mostly used here to measure N$_2$ fixation rates (addition of the $^{15}$N$_2$ tracer as a bubble in the incubation bottles -hereafter referred to as 'bubble addition method'- (Montoya et al., 1996)) has been seen to potentially underestimate rates (Großkopf et al., 2012; Mohr et al., 2010b; Wilson et al., 2012), compared to methods consisting in adding the $^{15}$N$_2$ as dissolved in a subset of seawater previously N$_2$ degased -hereafter referred to as '$^{15}$N$_2$-enriched seawater method'-, whereas other studies have not noted any difference between the two methods (Bonnet et al., 2016; Shiozaki et al., 2015).

In our lab, we have been using the bubble addition method for several years, e.g. (Bonnet et al., 2009; Bonnet et al., 2013; Bonnet et al., 2011; Dekaezemacker et al., 2013; Moutin et al., 2008) and the $^{15}$N$_2$-enriched seawater method in recent cruises and lab experiments (Benavides et al., 2015; Berthelot et al., 2015a; Berthelot et al., 2015b; Bonnet et al., 2016), and have compared the two methods. We have come to the conclusion that the associated drawbacks of degassing seawater and dissolving the $^{15}$N$_2$ are greater than the gain, considering the small differences observed between the two methods in Pacific waters (Bonnet et al. 2016, Shiozaki et al., 2015) and the high risk of sample

contamination involved when manipulating sample seawater to prepare dissolved $^{15}N_2$ (Klawonn et al., 2015).

In addition to the contamination issues, preparing dissolved $^{15}N_2$ on board represents additional time with samples sitting on the bench or rosette before incubation, which is especially critical in tropical environments. This can be solved using dissolved $^{15}N_2$ prepared beforehand in the lab using, for example, artificial seawater or seawater sampled from another station. Such an approach is suitable when the sampling is always performed within the same water mass (e.g. sampling inside an eddy for example), but using in situ seawater (from the same sample) is strongly preferred in long transect cruises that encompass productivity gradients, as was the case in the present studies.

For these reasons, the bubble addition method was used here, but we paid careful attention to accurately measuring the $^{15}N/^{14}N$ ratio of the $N_2$ pool in the incubation bottles, whatever the method used. The potential -but not systematic- underestimation when using the $^{15}N_2$ bubble method has indeed been attributed to the incomplete equilibration of $^{15}N_2$ in the incubation bottles (Mohr et al., 2010a), which results in a lower $^{15}N/^{14}N$ ratio of the $N_2$ pool as compared to the theoretical value that can be calculated on the basis of gas constants. Here, we systematically performed MIMS measurements of the $^{15}N/^{14}N$ ratio of the $N_2$ pool in the incubation bottles and provided $N_2$ fixation rates that are not underestimated due to this issue. Our MIMS results reveal a $^{15}N$ enrichment of the $N_2$ pool of 6.145 ± 0.798 atom% when bottles were incubated in on-deck incubators, and 7.548 ± 0.557 atom% when bottles were incubated on the in situ mooring line, which is clearly lower than the theoretical value of ~8.2 atom% based on gas constants calculation (Weiss, 1970).

We are aware the dissolution kinetics of $^{15}N_2$ in the incubation bottles may have been progressive along the 24 h of incubation (Mohr et al., 2010), therefore, the $N_2$ fixation rates provided here represent conservative values.

2. In the same context, I don't know the gas quality of the company you bought from, but I assume you checked for purity as recommended in the Dabundo paper. Otherwise the high rates may as well come from an ammonia incorporation or similar. Please present your quality check, here.

We are aware that Dabundo et al. (2014) reports potential contamination of some commercial $^{15}N_2$ gas stocks with $^{15}N$-enriched $NH_4^+$, $NO_3^-$ and/or $NO_2^-$, and nitrous oxide ($N_2O$). In their study, Dabundo et al. (2014) analysed various brands of $^{15}N_2$ (Sigma, Cambridge Isotopes, Campro Scientific) and found that the Cambridge Isotopes brand (i.e., the one used in these studies) contained low concentrations of $^{15}N$ contaminants, and the potential overestimated $N_2$ fixation rates modeled using this contamination level would range from undetectable to 0.02 nmol N $L^{-1}$ $d^{-1}$. The rates measured in this study were on average ~10 nmol N $L^{-1}$ $d^{-1}$, suggesting that stock contamination would be too low to affect the results reported here.

To verify this, the Cambridge Isotopes batches that are routinely used by our team has been analyzed for potential contamination in Julie Granger and Richard Dabundo's lab, and this confirmed that the contamination of the $^{15}N_2$ gas stock was low: 1.4 x $10^{-8}$ mol of $^{15}NO_3^-$ per mol of $^{15}N_2$, and 1.1x$10^{-8}$ mol $NH_4^+$ per mol of $^{15}N_2$. The application of this

**contamination level to our samples using the model described in Dabundo et al. (2014) indicates that our rates could only be overestimated by 0.01 to 0.12 %. We thus confirmed that the stock contamination issue did not affect the results reported here.**

3. In addition, ammonia background measurements, fluxes and inputs are not mentioned- this would add enormous value to the stud, so please present if available. As you are making a suggestion on zooplankton moderated export, ammonia is a good part of this, too.

**In our study, we focused on new N inputs (i.e. atmospheric deposition, $N_2$ fixation and vertical nitrate diffusion) thus associated with new production. Ammonium background was measured at the three LD stations (available on http://www.obs-vlfr.fr/proof/php/outpace/outpace.php) and was low (in the nM range) in the photic layer. We did not present ammonium data in our study, because ammonium fluxes were not measured during the cruise where the focus was essentially on new N budgets.**

4. No sequencing was performed and no single cell rates were determined- how can you interpret on the key N2 fixers if you just look at 6 clusters via qPCR? What makes you conclude that Trichodesmium or UCYN clusters are important if you don't assess which diazotrophs are there?

**We agree on this comment. The groups targeted by qPCR here are based on the companion paper of Stenegren et al. (this issue), who revealed that they were the major groups during the OUTPACE cruise.**

---

## Author Response (AR1)

Marseille, 27/02/2018

Dr Helge Niemann
Associated Editor – *Biogeosciences*

Dear Dr Niemann,

We are very grateful for the opportunity to revise our submission, entitled in its new version *'N$_2$ fixation as a dominant new N source in the Western Tropical South Pacific Ocean (OUTPACE cruise)'*, for publication in *Biogeosciences*. We followed the constructive and helpful feedbacks from four referees and the editor, incorporated the comments, and addressed the key concerns that were raised. Here we send you a new version of responses to the four reviewers. In this version, we provide what we intended to change/modify as you ask us in your comments, as it was unclear in the first version. In the revised version of the manuscript we have used the 'track changes' mode with different color for each reviewer to better assess the progress of the manuscript. As reviewers commented some identical points, we have performed changes with the color of the reviewer who firstly provided the comment. In addition, the text has been corrected by an English native speaker.

Moreover, in the revised version of the manuscript we have provided a new section in the discussion dedicated to the possible underestimation of N$_2$ fixation rates due to the utilization of the bubble method.

We thank you for your attention to our manuscript and we hope that you will find this new version ready for publication in *Biogeosciences*.

Sincerely, and on behalf of all co-authors,

Mathieu Caffin

**Response to Anonymous Referee #1**

We thank Anonymous Referee #1 for the time and effort devoted to the review of the manuscript. Below, we reproduce the reviewer's comments and address their concerns point by point. The reviewer's comments are copied below in regular font with our responses in blue.

We are responding to this review a long time after it was published on the *Biogeosciences Discussion* online forum because we had to submit the companion paper by Caffin et al. (this issue), cited in this article, before the closure of OUTPACE's special issue initially scheduled on December 31, 2017.

**General comments:**

In their study, Caffin *et al*. use a Lagrangian approach to determine N sources ($N_2$ fixation rates, $NO_3^-$ supply from vertical diffusion and N from atmospheric depositions) and sinks (particulate N export) at 3 stations in the Western South Tropical Ocean. They also measured primary production using $14C$ tracer incubations. Their main findings are that 1) $N_2$ fixation is the major source of new N (>90%) at all stations, regardless of whether the diazotroph community was dominated by *Trichodesmium* or unicellular cyanobacteria, 2) carbon export relative to primary production is high in this region, and 3) the sum of N input in the photic zone exceeds particulate N export. Overall, this study is interesting and timely as this region has recently been identified as a $N_2$ fixation hotspot. However, I have some concerns.

First, they report a low input from atmospheric depositions but did not consider atmospheric sources other than $NO_3^-$ and $NO_2^-$ (e.g., $NH_4^+$ DON, PON).

> Reviewer #1 is right that we did not consider atmospheric sources other than $NO_3^-$ and $NO_2^-$. Our flux could be underestimated as it represents dry deposition and because gas and organic forms were not measured. At the global scale, and depending on the location, organic nitrogen can represent up to 90 % of N atmospheric deposition (Kanakidou et al., 2012), and $NH_4^+$ could account for ~40% (Dentener et al. 2006). Even if we double our estimated deposition flux, the contribution of atmospheric deposition to N input remained low (<1.5 %) and consequently represented a minor contribution. This has been added in the discussion section page 14 line 3.
>
> Kanakidou, M., Duce, R. A., Prospero, J. M., Baker, A. R., Benitez-Nelson, C., Dentener, F. J., ... & Sarin, M. Atmospheric fluxes of organic N and P to the global ocean. Global Biogeochemical Cycles, 26(3), 2017

They also measured relatively high $N_2$ fixation rates and I am wondering if they considered the possibility of a contamination of their $15N-N_2$ stock with $15N-NO_3^-$ and $NH_4^+$, which would artificially increase their $N_2$ fixation rates, as recently reported by Dabundo *et al*. (2014).

> We are aware that Dabundo et al. (2014) reports potential contamination of some commercial $^{15}N_2$ gas stocks with $^{15}N$-enriched $NH_4^+$, $NO_3^-$ and/or $NO_2^-$, and nitrous oxide ($N_2O$). In their study, Dabundo et al. (2014) analysed various brands of $^{15}N_2$ (Sigma, Cambridge Isotopes, Campro Scientific) and found that the Cambridge Isotopes brand (i.e., the one used in these studies) contained low concentrations of $^{15}N$ contaminants, and the potential overestimated $N_2$ fixation rates modeled using this contamination level would range from undetectable to 0.02 nmol N $L^{-1}$ $d^{-1}$. The rates measured in this study

**were on average ~10 nmol N L$^{-1}$ d$^{-1}$, suggesting that stock contamination would be too low to affect the results reported here.**

**To verify this, the Cambridge Isotopes batches that are routinely used by our team has been analyzed for potential contamination in Julie Granger and Richard Dabundo's lab, and this confirmed that the contamination of the $^{15}$N$_2$ gas stock was low: 1.4 x 10$^{-8}$ mol of $^{15}$NO$_3$$^{-}$ per mol of $^{15}$N$_2$, and 1.1x10$^{-8}$ mol NH$_4$$^{+}$ per mol of $^{15}$N$_2$. The application of this contamination level to our samples using the model described in Dabundo et al. (2014) indicates that our rates could only be overestimated by 0.01 to 0.12 %. We thus confirmed that the stock contamination issue did not affect the results reported here. This has been added to the method section page 6 line 5 as following "The purity of the $^{15}$N$_2$ Cambridge isotopes stocks was previously checked by Dabundo et al. (2014) and more recently by (Benavides et al., 2015) and (Bonnet et al., 2016a). They were found to be lower than 2 x 10$^{-8}$ mol:mol of $^{15}$N$_2$, leading to a potential N$_2$ fixation rates overestimation of <1 %"**

Some of the references cited (this issue) were unavailable on the *Biogeosciences Discussions* online forum at the time of this review, making it impossible to evaluate these parts of the manuscript.

**We are aware that some of the references cited here (this issue) were unavailable on the *Biogeosciences Discussions* online forum at the time of this review, but most of them (except Bouruet-Aubertot et al., this issue; expected before March 31) are available now.**

I was also a bit confused regarding the novelty of their dataset: were the same N2 fixation rates, qPCR or any other data collected at the same stations during the OUTPACE cruise already published in previous studies? The authors should make a clear distinction of the new data contributed by their study versus the data already published elsewhere in other manuscripts in the special issue.

**We acknowledge that it must be difficult to review a paper which refers to many other papers in a Special issue, especially as some of the referred papers were not available online on the BG website at the time of the review and we would like to apologize for that. The objective of the OUTPACE special issue is to provide a unique opportunity for a group of researchers to focus on the "Interactions between planktonic organisms and biogeochemical cycles across trophic and N$_2$ fixation gradients in the western tropical South Pacific Ocean". It is a multidisciplinary approach with a tight time schedule, and with the main aim of sharing the data in order to provide best study in a relatively short time of a new (exceptional) set of data. The data may be used several times in different papers of the special issue focusing on different scientific questions. In the present case, the N$_2$ fixation data reported here with in situ incubations are not reported in any other paper, nor the qPCR data in the sediment traps. All the export data are also original and not reported anywhere in the special issue. Finally such N budgets including all potential N sources is not reported elsewhere neither.**

**Specific comments:**
**Title**
The title is a bit long and not focused on the main point of the study. I suggest changing for: "N2 fixation as the dominant new N source in the Western Tropical South Pacific Ocean (OUTPACE cruise)"

**We had chosen another title containing the Lagrangian term, but because two reviewers suggested the same change, we have changed the title in accordance with this suggestion. The new title is now: 'N$_2$ fixation as a dominant new N source in the Western Tropical South Pacific Ocean (OUTPACE cruise)'**

**Introduction**

Page 2, line 15: Knapp *et al.* (2008) and Bourbonnais *et al.* (2009) also observed a low δ15N of NO3- (relative to δ18O-NO3-) in surface waters in the western and eastern subtropical Atlantic Ocean, supporting the role of N2-fixers in these regions.

**We have added these missing references in accordance with this suggestion, thus we have modified the text page 3 line 3 in the following way " Knapp et al. (2008) and Bourbonnais et al. (2009) also observed a low δ$^{15}$N of NO$_3^-$ (relative to δ$^{18}$O-NO$_3^-$) in surface waters in the western and eastern subtropical Atlantic Ocean, supporting the role of N$_2$-fixers in these regions."**

Page 4, lines 6: What factors influence the distribution of *Trichodesmium* or UCYN? I believe temperature is an important factor (see Moisander *et al*., 2010). This point should be discussed a bit more.

**We are aware that the activity and distribution of diazotrophs have been hypothesized to be controlled by several environmental variables in the open ocean, such as light (Fu and Bell, 2003; Breitbarth et *al*., 2008; Levitan et *al*., 2010), temperature (Capone et *al*., 1997; Staal et *al*., 2003; Breitbarth et *al*., 2007; Moisander et *al*., 2010) or nutrient availability (Van den Broeck et *al*., 2004; Moutin et *al*., 2005; Mills et *al*., 2004; Ho, 2013). However, we did not focus on this in our study. A niche partitioning between *Trichodesmium* and UCYN-A has been observed in this region (Moisander et al., 2010; Bonnet et al., 2015) but further west of the studied region. In the context of the OUTPACE project, Stenegren et al., (This issue) and Bonnet et al. (this issue) have shown a correlation between diazotrophs abundance and temperature, which is mainly due the temperature changes with depth. The longitudinal pattern observed here in surface waters has been attributed to an important decrease of iron concentrations in SPG waters as compared to MA waters (Bonnet et al.;, This issue; Guieu et al., in review). All this would be too long to explain in the introduction section so we decided to only refer to the above mentioned papers in the following sentence page 4 line 8 'This west to east N$_2$ fixation gradient has been mainly attributed to a decrease of iron availability in SPG waters as compared to MA waters (Bonnet et al., This issue; Guieu et al., in rev.).'**

**Experimental procedures**

Page 5, line 19: They only considered NO3- and NO2- when quantifying N atmospheric depositions. They should also consider NH4+ or organic nitrogen (particulate or dissolved). For instance, Cornell *et al*. (1995) estimated that organic nitrogen was a significant component of atmospheric N depositions even in remove marine regions.

**We understand and accept this suggestion that NH$_4^+$ and organic N (particulate and dissolved) should also be considered when quantifying atmospheric deposition. Our quantification of the atmospheric deposition could be underestimated as it only represents dry deposition, and gas and organic forms were not measured. At global scale,**

and depending on the location, organic nitrogen could represent up to 90 % of N atmospheric deposition (Kanakidou et al., 2012), and $NH_4^+$ could account for ~40% (Dentener et al. 2006. Even if we double our estimated deposition flux, atmospheric deposition remained low (< 1.5 %) and consequently represented a minor contribution of the new N input.

Thus we have discussed about this in the section 'Discussion – Contribution of $N_2$ fixation to new N input in the WTSP' and modified the text page 12 line 3 in the following way "Extrapolated NOx deposition from the atmosphere during OUTPACE (range: 0.34 – 1.05 µmol $m^{-2}$ $d^{-1}$) were one order of magnitude lower than predicted with major uncertainties by global models that include wet and gas deposition for that region (Kanakidou et al., 2012). Our flux could be an underestimation as it represents only dry deposition and as gas and organic forms were not measured. At the global scale and depending on the location, organic nitrogen could represent up to 90 % of N atmospheric deposition (Kanakidou et al., 2012), and $NH_4^+$ could account for ~40% (Dentener et al. 2006). Even if we double our estimated deposition flux, atmospheric deposition still remained low (< 1.5 %) and consequently represented a minor contribution of the new N input (Table 2). This negligible contribution of atmospheric input to the overall N budget (less than 1.5 %) therefore implies an important contribution of other terms, such as $N_2$ fixation."

Page 5, line 26: Did they check their commercial 15N Eurisotop gas for possible contamination with 15N-labeled dissolved inorganic nitrogen (NO3-, NO2- and NH4+)? Microbial assimilation of contaminant 15N labeled dissolved inorganic nitrogen would artificially increase N2 fixation rates. Dabundo *et al*. (2014) recently reported significant concentrations of 15N contaminants in 15N-labelled N2 gas supplied by Sigma-Aldrich and Campro Scientific.

Please see response above.

Page 6, lines 5-7: A better way to assess whether equilibration was complete would be to try different treatments in triplicate, i.e., shake the bottles for different times and intensity before the *in-situ* incubations.

We agree that it would be a useful complementary study and will think about it for the future. In the present study, we performed MIMS analyses to quantify the final $^{15}$N-enrichment in the $N_2$ pool to minimize any potential underestimation due to an insufficient equilibration process.

Page 6, line 6: Add "incomplete" before equilibration.

This has been corrected in the new version

Page 6, lines 7-10: I assume the 12 mL subsample was collected without contact with the atmosphere?

The 12 mL subsample was collected rapidly from the 4.5 L to the Exetainers with as less contact as possible with the atmosphere. We assume that the sampling was rapid enough to minimize any sea-air exchanges that can affect the $^{15}N_2$ enrichment of the sample. In any case, this is more precise than using the theoretical $^{15}N_2$ enrichment, as revealed in Bonnet et al. (this issue).

Page 6, lines 16-24: What is the detection limit for their N2 fixation rates?

**The minimum quantifiable rate calculated using standard propagation of errors via the observed variability between replicate samples measured according to Gradoville et al. (2017) was 0.035 nmol N L$^{-1}$ d$^{-1}$.**

**This has been specified in the method section of the new version of the paper, page 8 line 6, in the following way "The minimum quantifiable rate calculated using standard propagation of errors via the observed variability between replicate samples measured according to Gradoville et al. (2017) was 0.035 nmol N L$^{-1}$ d$^{-1}$.".**

**Gradoville, M. R., D. Bombar, B. C. Crump, R. M. Letelier, J. P. Zehr and A. E. White Diversity and activity of nitrogen fixing communities across ocean basins. Limnol. Oceanogr. 62: 1895-1909, 2017**

Page 7, line 21: Define UCYN-B, UCYN-A1, het-1 and het-2. Which bacteria are represented by these different groups?

**The sentence has been modified page 9 line 14 in the following way "Diazotroph abundance for *Trichodesmium* spp., UCYN-B (*Crocosphaera watsonii*), UCYN-A1 (*Candidatus Atelocyanobacterium thalassa*), het-1 (*Richelia intracellularis* in symbiosis with *Rhizosolenia*), and het-2 (*Richelia intracellularis* in symbiosis with *Hemiaulus*) were quantified by qPCR analyses on the nifH gene using previously described oligonucleotides and assays (Foster et al. 2007; Church et al. 2005)."**

Page 8, line 1: Were samples with qPCR reaction efficiency below 95% reported? Why not repeat analysis for these samples?

**The samples with qPCR reaction efficiency below 95% were excluded. This sentence has been added page 9 line 29 in the new version of the paper.**

Why not repeat analysis for these samples?

**As the amount of water was restricted for each parameter measurement, we were not able to repeat the analysis for these samples.**

**Results**
Page 9, lines 24 to 27: How are these rates different from the one measured in Bonnet *et al.*, 2017 (This issue). Are they the same rates as reported in Bonnet *et al.* (2017)? The Bonnet *et al.* paper was not yet available at the time of this review, making it impossible to effectively evaluate this part of the manuscript.

**We understand that it was impossible for Reviewer #1 to evaluate this part of the manuscript as the Bonnet et al. paper was not available at the time of the review. The Bonnet el al. paper is now available in the *Biogeosciences Discussion* online forum. In the present paper, we report rates from in-situ incubations at the 3 long duration station in order to perform N budgets. The Bonnet et al. (this issue) paper reports rates from on-deck incubation at 15 short duration stations (that were not included in the Caffin et al. study) and the 3 long duration stations with the aim to provide group-specific N$_2$ fixation rates using single-cell analyses (nanoSIMS). So the goal of each paper is totally different. However, it has to be noted that the integrated N$_2$ fixation rates from the Bonnet et al.,**

**(this issue) paper have been used in the database published in Bonnet et al. (2017), together with 6 other cruises.**

Page 11, line 11: Were *Trichodesmium* data for LD A (150 m) not available or below detection limit (as stated on page 10, line 32)?

**Trichodesmium data for LD A (150 m) were not available as the qPCR efficiency was below 95 %.**

**This has been modified on page 13 line 5 in the following way "… sediment material (< QL at LD C 330 m and not available at LD A 150 m) and …"**

**Discussion**

Page 11, lines 27-29: Perhaps atmospheric deposition measured during OUTPACE are low because they neglected contributions from organic nitrogen and NH4+. This possibility should be discussed (see my previous comment, page 5, line 19).

**We agree with this comment of Reviewer #1, thus we have discussed the possibility of underestimation of the atmospheric input in the new version and modified the text page 14 line 3 in the following way: "Extrapolated NOx deposition from the atmosphere during OUTPACE (range: 0.34 – 1.05 µmol m$^{-2}$ d$^{-1}$) were one order of magnitude lower than predicted with major uncertainties by global models that include wet and gas deposition for that region (Kanakidou et al., 2012). Our flux could be an underestimation as it represents only dry deposition and as gas and organic forms were not measured. At the global scale and depending on the location, organic nitrogen could represent up to 90 % of N atmospheric deposition (Kanakidou et al., 2012), and NH$_4^+$ could account for ~40% (Dentener et al. 2006). Even if we double our estimated deposition flux, atmospheric deposition still remained low (< 1.5 %) and consequently represented a minor contribution of the new N input (Table 2). This negligible contribution of atmospheric input to the overall N budget (less than 1.5 %) therefore implies an important contribution of other terms, such as N$_2$ fixation."**

Page 12, lines 24-27: Again, it would be relevant to check for possible contamination of their 15N2 Eurisotop stock by 15N-labelled dissolved inorganic nitrogen (see Dabundo *et al*., 2014).

**Please see response above.**

Page 14, line 22-23: The Berman-Frank paper was not submitted at the time of this review. Also, define PCD.

**The Berman-Frank paper has been replaced by Spungin et al. (This issue) (her student), available in the *Biogeosciences Discussion* online forum at this time.**

**PCD (Programmed cell death) has been defined in the new version of the paper page 17 line 8 in the following way: "Programmed cell death (PCD) was detected at LD B (Spungin et al., this issue)…"**

**Spungin, D., Belkin, N., Foster, R., Stenegren, M., Caputo, A., Pujo-Pay, M., Leblond, N., Dupouy, C., Bonnet, S., and Berman-Frank, I.: Programmed cell death in diazotrophs**

**and the fate of organic matter in the Western Tropical South Pacific Ocean during the OUTPACE cruise, Biogeosciences Discuss., in review, 2018.**

Page 16, lines 1-6: This paragraph is not clear. Do they mean the dead and live "swimmers" zooplankton were not distinguishable? Rewrite accordingly.

**This paragraph has been rewritten to explain more clearly that the dead and live "swimmers" zooplankton were not distinguishable. We have modified the text page 18 line 19 in the following way "Finally, the zooplankton themselves are sampled by the traps but dead zooplankton were not distinguishable from live swimmers. In the present study, the zooplankton…"**

**References**
The following cited references were not accessible on the *Biogeosciences Discussion* online forum at the time of this review:

Berman-Frank *et al.* (This issue)
Bonnet *et al.* (This issue)
Bouruet-Aubertot *et al.* (This issue)
Caffin *et al.* (This issue)
Moutin *et al.* (This issue) – there is a Moutin *et al.* submitted but with a different title
Van Wambeke *et al.* (This issue)

**To date, the Bouruet-Aubertot et al. (this issue) paper is not available yet, but is expected before March 31. The Berman-Frank et al. (this issue) paper was submitted as Spungin et al. (this issue); all the other references are available in the *Biogeosciences Discussion* online forum.**

**Tables**
Table 5: Include contributions from atmospheric depositions in this table.

**In most of the studies presented in Table 5, the N input associated with atmospheric deposition was not quantified. Only in the Mediterranean Sea, which is strongly affected by dust deposition, the atmospheric deposition was measured. Atmospheric models and global tropospheric budgets were performed (Duce et al., 2008; Kanakidou et al., 2012) that can give us a global picture of atmospheric deposition, however to be consistent with our study we have decided not to include those data in Table 5. In addition, the main objective of this table is to compare the contribution of $N_2$ fixation as a new N input in different regions of the world Ocean. Thus, we prefer not to add atmospheric input in the table.**

**Figures**
Figure 3: Why PAR and DCM are decoupled at station LD B?

**The decoupling between PAR and DCM observed at LD B remains in significant chlorophyll a concentration observed at the surface that was stirred, deformed and transported by the mesocscale circulation, as explained in de Verneil et al. (this issue) who focused their study on the significant surface chlorophyll a bloom sampled at LD B.**

de Verneil, A., Rousselet, L., Doglioli, A. M., Petrenko, A. A., and Moutin, T.: The fate of a southwest Pacific bloom: gauging the impact of submesoscale vs. mesoscale circulation on biological gradients in the subtropics, Biogeosciences, 14, 3471-3486, https://doi.org/10.5194/bg-14-3471-2017, 2017.

**Technical considerations:**

Review the manuscript for grammatical errors and typos. Here are a few examples:

Page 1, line 21: replace "Thanks to a Lagrangian…" for "Using a Lagrangian…".

>**This has been corrected in the new version**

Page 1, line 34: replace "while there contribution…" for "while their contribution…"

>**This has been corrected in the new version**

Page 9, line 13: replace for "… in-situ incubation method ranged from 0.0 – 19.3 nmol N L$_{-1}$ d$_{-1}$ …"

>**This has been corrected in the new version**

Page 10, line 11: replace by "was strongly influenced by the vertical diffusion coefficient".

>**This has been corrected in the new version**

Page 11, line11: replace "LDA" by "LD A".

>**This has been corrected in the new version**

Page 11, line 21: replace "whatever" for "regardless of", i.e., "… N$_2$ fixation was the major external source of N to the WTSP regardless of the degree of oligotrophy, …"

>**This has been corrected in the new version**

Page 12, lines 22-3: change for "… NO$_3^-$ input by turbulence always represented a minor contribution to the N budget."

>**This has been corrected in the new version**

Page 13, line 16: replace for "… and a clear dominance of …"

>**This has been corrected in the new version**

Page 14, lines 9-10: Add a period after "in oligotrophic open ocean regions." Start a new sentence with "To date, few qPCR *nifH* data from sediment traps are available…".

>**This has been corrected in the new version**

Perhaps the proportion of dead versus live zooplankton could be estimated from the flask not filled with formaldehyde collected on the fifth day of sampling and used for diazotroph quantification.

>**Reviewer #1's suggestion is interesting as a way to estimate the proportion of dead versus live zooplankton. In fact, all the zooplankton (dead and alive) were recovered from the**

**flask filled with formaldehyde, while only dead zooplankton could be recovered from the flask that was not filled with formaldehyde. However, zooplankton were not recovered from the poisoned and unpoisoned flask on the same day; that is a problem, as we observed daily variability in the amount of zooplankton recovered from the flask.**

Page 15, line 4: change for: "… in different oligotrophic regions of the ocean, for instance, the SPG …"

**This has been corrected in the new version**

**Response to Anonymous Referee #2**

**We thank Anonymous Referee #2 for the time and effort devoted to the review of the manuscript. Below, we reproduce the reviewer's comments and address their concerns point by point. The reviewer's comments are copied below in regular font, with our responses in red.**

**We are responding to this review a long time after it was published on the *Biogeosciences Discussion* online forum because we had to submit a companion paper by Caffin et al. (this issue), cited in this article, before the closure of OUTPACE's special issue initially scheduled on December 31, 2017.**

**Summary and Overall Impressions**

This is a well-written, interesting, enjoyable paper that quantifies the source of "new" nitrogen to the euphotic zone as well as the flux of nitrogen derived from N2 fixation with respect to diazotroph community and other factors. Data was collected from three stations in the western tropical South Pacific ocean. Two of these stations were oligotrophic while the third was "ultra" oligotrophic. Nitrogen fixation was overwhelmingly the largest source of N input due in part to very high rates of nitrogen fixation and lower than typical rates of atmospheric deposition. N flux from nitrogen fixation was uncoupled from N inputs, and possible reasons for this are well discussed. This paper is well presented and the "story" is clearly told. Everything is well organized and easy to follow. The conclusions are interesting and important, and I really liked the 5 day averaging and well resolved vertical profiles. Daily variability is high and important to measure! In addition, the comparison between in situ and on deck incubations is useful and important, as most researchers are unable to use in situ arrays. The introduction and discussion sections are very well done, but I do have some concerns about some of the methods and the presentation of the results. My primary concerns are methodological and regard 1) the nitrogen fixation rate calculations including the absence of a reported detection limit and a failure to completely address the failings of the "bubble method" and 2) propogation of error throughout calculations and some additional statistical comparisons. These issues can all using data already in hand, and I think that these changes are unlikely to impact the conclusions of the manuscript. Nevertheless statistical rigor and proper calculations are important. I have also listed some line item concerns below.

**General Comments**

Throughout the paper, nitrate transport is referred to as a "diffusive flux", but "diffusion" brings to mind molecular diffusion, whereas this flux is really a result of vertical mixing. Please rename this term or, if this is standard terminology, indicate clearly that this is not molecular diffusion but is a physical mixing process.

> **We agree with this comment and nitrate transport is now referred to as "turbulent diffusive flux" to avoid any confusion with molecular diffusion. We have modified the term throughout the manuscript in accordance with this suggestion.**

Regarding atmospheric deposition, was atmospheric deposition of NH4 considered? What about DON? - These would be "new" N that look like regenerated N.

> **We agree that $NH_4^+$ and organic N (particulate and dissolved) should also be considered when quantifying atmospheric deposition. Our quantification of the atmospheric deposition could be underestimated as it only represents dry deposition and that gas and organic forms were not measured. At the global scale and depending on the location,**

organic nitrogen could represent up to 90 % of N atmospheric deposition (Kanakidou et al., 2012), and $NH_4^+$ could account for ~40% of total N deposition (Dentener et al. 2006). Even if we double our estimated deposition flux, atmospheric deposition remained low (< 1.5 %) and consequently represented a minor contribution of the new N input.

Thus we have discussed about this in the section 'Discussion – Contribution of $N_2$ fixation to new N input in the WTSP' and modified the text page 14 line 3 in the following way "Extrapolated NOx deposition from the atmosphere during OUTPACE (range: 0.34 – 1.05 µmol $m^{-2}$ $d^{-1}$) were one order of magnitude lower than predicted with large uncertainties by global models that include wet and gas deposition for that region (Kanakidou et al., 2012). Our flux could be underestimation as it represents dry deposition and that gas and organic forms were not measured. At the global scale and depending on the location, organic nitrogen could represent up to 90 % of N atmospheric deposition (Kanakidou et al., 2012), and $NH_4^+$ could account for ~40% (Dentener et al. 2006). Even if we double our estimated deposition flux, atmospheric deposition still remained low (< 1.5 %) and consequently represented a minor contribution of the new N input (Table 4). This negligible contribution of atmospheric input to the overall N budget (less than 1.5 %), therefore imply an important contribution of other terms, such as $N_2$ fixation."

Kanakidou, M., Duce, R. A., Prospero, J. M., Baker, A. R., Benitez-Nelson, C., Dentener, F. J., ... & Sarin, M. Atmospheric fluxes of organic N and P to the global ocean. Global Biogeochemical Cycles, 26(3), 2017

Also, throughout the manuscript, "atmospheric deposition" reflects dry deposition only (as opposed to wet). Please modify the text throughout so that this distinction is clear.

We have changed "atmospheric deposition" to "dry atmospheric deposition" throughout the text, as recommended.

Some of the notation was a bit confusing was used both for integrated N2 fixation rates (i-N2) and density and nitracline density (NO3). I know that these are established conventions, but use of the same abbreviation is a bit confusing and these should be clarified somehow.

To clarify this confusion, "ρ" is now used to refer only to density and "ρNO3" to the isopycne associated to nitracline. Thus, we have modified this throughout the manuscript, as recommended. The special notation of integrated $N_2$ fixation rates has been deleted as it was not used in the text.

The methods section could use a bit more detail and/or additional references. I have indicated specific problems below.

We have added more detail and additional references in the new version. The atmospheric deposition section has been modified page 6 line 19 in the following way: "N atmospheric deposition ($NO_3^-$ and $NO_2^-$ (nitrite), hereafter called NOx) was quantified along the transect after dissolution of aerosols collected continuously during the transect as described in Guieu et al. (in rev.). Briefly, the sampling device, designed to avoid ship contamination was installed at the look-out post in the front of the ship, collected aerosols at ~20 L $min^{-1}$ on onto polycarbonate, 47-mm diameter, 0.45-µm porosity (previously acid-cleaned with a 2% solution of HCl (Merck, Ultrapur, Germany) and

**thoroughly rinsed with ultra-pure water and dried under a laminar flow bench and stored in acid-cleaned Petri dishes). Dissolution experiments to determine NOx released in surface seawater after deposition were performed on board using acid-cleaned Sartorius filtration units (volume 0.250 L) and filtered surface (5m) seawater. Each sample was subjected to two contact times: the first contact was at one minute, and the second contact was at 24 hours. NOx was analysed using a 1-m long Liquid Waveguide Capillary Cells (LWCC) made of quartz capillary tubing, following the protocol described in Louis et al., 2015. An extrapolated NOx release from dry deposition was estimated on the basis of a deposition velocity of submicronic particles (0.4 m s$^{-1}$; Vong et al., 2010)."**

In section 2.1, a more clear description of which parameters led to the designation of the three stations would be useful, i.e. LD A was oligotrophic and Tricho dominated, etc. A map of station locations would be helpful, to be referenced on p. 4 lines 11-14, which could be Figure 1 or similar.

**A clearer description of which parameters led to the designation of the three stations is now proposed, but it appears in the Introduction because most of the parameters concerning the diazotroph domination (i.e. *Trichodesmium* vs UCYN), the trophic gradient (i.e. oligotrophy vs ultra-oligotrophy) and the global N$_2$ fixation rates of each region (i.e. MA vs SPG) have been explained and published in other papers (Stenegren et al., this issue; Bonnet et al., this issue; Bonnet et al., 2017; Moutin et al., 2017). The detailed description of the whole transect is beyond the scope of this paper. Thus, we refrain from changing the text.**

**See Introduction section:**

**"The WTSP Ocean has recently been identified as a hotspot of N2 fixation, including N$_2$ fixation rates >500 μmol N m$^{-2}$ d$^{-1}$ (Bonnet et al., 2017). The region covered by the OUTPACE cruise is characterized by trophic and N$_2$ fixation gradients (Moutin et al., 2017). The region covered by the OUTPACE cruise encompasses contrasting trophic regimes characterized by strong differences in top nitracline depths, from 46 to 141 m (Moutin et al., this issue) and representing a large part of the oligotrophic gradient at the scale of the world Ocean (Moutin and Prieur., 2012; their Fig. 9). The westward oligotrophic waters are characterized by high N$_2$ fixation rates (631 ± 286 μmol N m$^{-2}$ d$^{-1}$) mainly associated with Trichodesmium (i.e. within the hotspot around Melanesian archipelago waters, hereafter named MA), and the eastward ultra-oligotrophic waters (in the eastern border of the South Pacific gyre, hereafter named SPG waters) are characterized by low N$_2$ fixation rates (85 ± 79 μmol N m$^{-2}$ d$^{-1}$), mainly associated with UCYN (Bonnet et al., this issue; Stenegren et al., this issue)."**

**Then, in the section 2.1 we had mentioned that LD A and LD B were positioned in the MA waters, and LD C in the SPG waters; and referred to Fig. 1.**

Regarding the nitrogen fixation rate calculations, while I understand the reluctance to add equilibrated seawater to the incubations, especially given the oligotrophic nature of the samples, the compromise proposed here (to measure 15N2 at the conclusion of the incubation) does not adequately reconcile the rates calculated here with the problems associated with the bubble method. It has been shown that the fraction label of the dissolved N2 "source pool" changes during the course of the incubation (Mohr 2010 and others). However, the method used in this manuscript assumes a steady fraction label of that pool, based on the fraction label measured at the end of the incubation, inevitably resulting in an

underestimation of nitrogen fixation rates. This is problematic because a) different organisms fix at different times of day and b) the rate of change likely varies based on physical and chemical parameters. Problem (a) is especially relevant to this study as the researchers have specifically set out to compare regions with different diazotroph assemblages. At this stage, this problem cannot be addressed directly, but some discussion of the implications of this in light of the conclusions is warranted. Problem (b) can and should be directly addressed by correcting the rate calculations for the rate of bubble dissolution. Examples of this correction can be found in Figure 1 of Mohr (2010) and the Supplemental Figure in Jayakumar (2017). Since these were incubated on an in situ array, differences in temperature with depth could have variably impacted rates of 15N2 dissolution. This is addressed to some degree in those two publications and should be discussed in this manuscript.

**We agree with these comments and try to give below some more explanations regarding our methodological choice. We decided to use the 'bubble' method to avoid any trace metal and DOM contaminations (frequent in the study area, Benavides et al., 2017; Moisander et al., 2011) in the incubation bottles and avoid any potential over-estimation of rates. However, we are aware that this method potentially underestimates $N_2$ fixation rates (Großkopf et al., 2012; Mohr et al., 2010; Wilson et al., 2012) due to the incomplete (and gradually increasing during the incubation period) equilibration of the $^{15}N_2$ in the incubation bottles when injected as a bubble. This results in a lower $^{15}N/^{14}N$ ratio of the $N_2$ pool available for $N_2$ fixation (the term $A_{N2}$ used in the Montoya et al. (1996) equation) as compared to the theoretical value calculated based on gas constants, and therefore potentially leads to underestimated rates in some studies, whereas other studies do not see any significant differences between both methods (Bonnet et al., 2016c; Shiozaki et al., 2015). Here we paid careful attention to accurately measure the term $A_{N2}$ to minimize any potential underestimation. The average values at the end of the incubation period was 7.548 ± 0.557 atom%, which is lower than the theoretical value of ~8.2 atom% based on gas constants calculations (Weiss, 1970). We are aware the dissolution kinetics of $^{15}N_2$ in the incubation bottles is progressive along the 24 h of incubation (Mohr et al., 2010a). Therefore, the $^{15}N$ enrichment of the $N_2$ pool measured with the MIMS likely represent maximum values, and the $N_2$ fixation rates provided in this study represent minimum values. We decided not to perform new calculations based on examples given in Jayakumar 2017 but clearly specifify that the rates presented here have to be considered as minimum values. This has been added in the discussion section page 18 line 30 'Methodological underestimation leads to a possible higher contribution of $N_2$ fixation':**

**"In this study we intentionally used the 'bubble method' to measure $N_2$ fixation rates, considering the small differences observed between the two methods in Pacific waters (Bonnet et al. 2016, Shiozaki et al., 2015) and the high risk of sample contamination involved when manipulating sample seawater to prepare dissolved $^{15}N_2$ (Klawonn et al., 2015). In addition to the contamination issues, preparing dissolved $^{15}N_2$ on board represents additional time with samples sitting on the bench or rosette before incubation, which is especially critical in tropical environments. To reduce any potential underestimation, we measured the $^{15}N$ enrichment of the $N_2$ pool at the end of the incubation (7.548 ± 0.557 atom%, Bonnet et al., this issue), which was lower than the theoretical value of ~8.2 atom% based on gas constants calculations (Weiss, 1970). We are aware the dissolution kinetics of $^{15}N_2$ in the incubation bottles is progressive along the 24 h of incubation (Mohr et al., 2010a). Therefore, the $^{15}N$ enrichment of the $N_2$ pool measured with the MIMS likely represent maximum values, and the $N_2$ fixation rates**

**provided in this study represent minimum values. This reinforces the conclusions of this study regarding the prominent role of $N_2$ fixation in this region. This reinforces the conclusions of this study regarding the prominent role of $N_2$ fixation in this region"**

**Regarding the second potential issue (effect of the diazotroph community composition), we are aware that Großkopft and colleagues (2012) found lower discrepancies between both methods when Trichodesmium was dominating the community than when UCYN were dominating. Despite that this potential underestimation was kept minimal in this study due to the measurement of the A2 term, it is likely that the underestimation was lower in MA waters (where Trichodesmium dominated) than in SPG waters (where UCYN dominated). We have added the following sentence in the discussion section page 19 line 10:**

**"Großkopft et al., (2012) found that the discrepancy between both methods was more important when UCYN dominates the diazotroph community as compared to when Trichodesmium dominates. Consequently, N2 fixation rates in this study are potentially more underestimated in SPG waters than in MA waters. By applying the maximum factor of underestimation found by Grosskopft et al (1.7), $N_2$ fixation in SPG waters would have been higher (100 µmol N m$^{-1}$ d$^{-1}$ instead of 59), which is still far lower than in MA waters and does not change de conclusions of this study".**

Atmospheric deposition is really a flux to the mixed layer. Depending on mixed layer depth relative to the nitracline, mixing speeds, and biological uptake rates the upward flux of NO3 may reflect a flux to the sub-mixed layer euphotic zone only. Nitrogen fixation rates also show vertical structure. I assume that diazotroph communities have vertical structure (might be nice to put this on Figure 2 if these data were collected, as well as density to show mixed layer depth); Tricho floats (if much of the Tricho is floating on the surface, was it even sampled?). Diatoms can adjust their bouyancy. The 200 m over which everything is integrated is a fairly large region. How does one account for these mini-environments within that 200 m in the budget? Could the vertical structure in N fixation reflect higher NO3 concentrations (even though they're undetectable – but the d.l. in this study was pretty high for oligotrophic waters) closer to the nitracline? Some discussion of the vertical structure within the euphotic zone may be informative.

**Here, we focused on the photic layer (from the surface to 0.1 % of surface irradiance), and on what comes from above and below the photic layer, and what is produced within the photic layer. The photic layer that we studied goes down to 125 m, 100 m and 200 m for LD A, LD B and LD C, respectively. We presented integrated rates (per m$^2$) and considered the whole photic layer, and we are aware of the heterogeneity of the diazotroph communities' vertical structure and the processes that happened in the entire photic layer. Data of diazotroph communities' vertical structure was collected and is available on the OUTPACE database (http://www.obs-vlfr.fr/proof/php/outpace/outpace.php). In the study we focus on $N_2$ fixation, a specific paper (Stenegren et al., this issue) focus on diazotroph communities' structure during the OUTPACE transect, which was beyond the scope of the main focus of this paper. Thus we refrain from changing the text.**

The abstract could use more contextual information and conclusions. In its current form it reads as a bit of a data dump. The interesting conclusions of the paper could be better showcased in this section.

In accordance with this comment and that of reviewer #4, we have rewritten the Abstract in the following way:

"We performed nitrogen (N) budgets in the photic layer at three contrasting stations representing different trophic conditions in the western tropical South Pacific (WTSP) Ocean during austral summer conditions (Feb. Mar. 2015). Using a Lagrangian strategy, we sampled the same water mass for the entire duration of each long duration (5 days) station, allowing us to consider only vertical exchanges for the budgets. We quantified all major vertical N fluxes both entering the system ($N_2$ fixation, nitrate turbulent diffusion, atmospheric deposition) and leaving the system (particulate N export). The three stations were characterized by strong nitracline and contrasted deep chlorophyll maximum depths, which was lower in the oligotrophic Melanesian archipelago (MA, stations LD A and LD B) than in the ultra-oligotrophic waters of the South Pacific gyre (SPG, station LD C). $N_2$ fixation rates were extremely high at both LD A ($593 \pm 51$ µmol N m$^{-2}$ d$^{-1}$) and LD B ($706 \pm 302$ µmol N m$^{-2}$ d$^{-1}$), and the diazotroph community was dominated by *Trichodesmium*. $N_2$ fixation rates were lower ($59 \pm 16$ µmol N m$^{-2}$ d$^{-1}$) at LD C, and the diazotroph community was dominated by unicellular $N_2$-fixing cyanobacteria (UCYN). At all stations, $N_2$ fixation was the major source of new N (> 90 %) before atmospheric deposition and upward nitrate fluxes induced by turbulence. $N_2$ fixation contributed circa 13-18 % of primary production in the MA region and 3 % in the SPG water and sustained nearly all new primary production at all stations. The e-ratio (e-ratio = particulate carbon export / primary production) was maximum at LD A (9.7 %) and was higher than the e-ratio in most studied oligotrophic regions (<5 %), indicating a high efficiency of the WTSP to export carbon relative to primary production. The direct export of diazotrophs assessed by qPCR of the *nifH* gene in sediment traps represented up to 30.6 % of the PC export at LD A, while their contribution was 5 and < 0.1 % at LD B and LD C, respectively. At the three studied stations, the sum of all N input to the photic layer exceeded the N output through organic matter export. This disequilibrium leading to N accumulation in the upper layer appears as a characteristic of the WTSP during the summer season, although the role of zooplankton in export fluxes should be further investigated."

**Technical and Line Item Comments**

**Abstract**

Line 20: "...all major fluxes..." could be "...all major vertical fluxes..."

**This has been corrected in the new version**

Line 21: Instead of "Thanks to a Lagrangian..." this would be more clear as "Using a Lagrangian..."

**This has been corrected in the new version**

Line 22: "...allowing to consider..." Should this read "allowing us to consider"?

**We have changed this to "allowing us to consider"**

Line 23: Might it be easier to refer to stations as A, B, and C without the 'LD', as it does not seem to have any necessary significance.

**The reference "LD" has been kept in the interests of consistency with all the papers of the OUTPACE special issue.**

Line 26: The N2 fixation measurements appear to have been depth integrated (units umol N/m2/d), but what depth region were these integrated over? Also, what is meant by "extremely high"? In comparison to what, are these high?

**$N_2$ fixation rates were integrated over the photic layer (i.e. from the surface to 0.1% of surface irradiance, as described in the section 'Nitrogen fixation rates' of the Material and Methods), corresponding to surface to 105, 80 and 180 m for LD A, LD B and LD C, respectively. The "extremely high" $N_2$ fixation rates measured here are in the upper range of the rates reported in the global $N_2$ fixation Marine Ecosystem Data (MAREDAT) database (Luo et al., 2012), that we have compared our findings with, which has been in the text line 15 page 5.**

Line 30-31: "N2 fixation...at all stations." Does this refer only to the LD A, B, and C stations? If not, please state how many stations were sampled in each region for these data.

**In this sentence, we only referred to the three stations LD A, LD B and LD C, on which we focused in this study. Thus we did not change the text.**

Line 31: PC and PP have not been defined.

**This has been corrected page 2 line 14 as follows: "The e-ratio (e-ratio = particulate carbon export / primary production) was maximum at…"**

Line 34: "there contribution" should be "their contribution"

**This has been corrected in the new version**

Line 36-37: "This disequilibrium...summer season..." I don't understand this sentence. Does this mean that this disequilibrium is generally held to be true (confirmed by other studies) or that you found this here and that it was consistent across all stations?

**This disequilibrium leading to N accumulation in the upper layer during the summer season was consistent across all stations.**

Line 37: The mention of zooplankton seems strange. I would delete it.

**Reviewer #2 is right, thus we have delete the mention of zooplankton.**

**Introduction**

p 2 Line 5: "di-nitrogen" Is this journal format? I have generally seen it written as "dinitrogen".

**It has been corrected to "dinitrogen" in the new version**

p 2 Line 5: "ammonia" At seawater and physiological pH, it is primarily "ammonium" that is present.

**Rev. 2 is correct, because although ammonia ($NH_3$) is the direct product of this reaction, it is quickly ionized to ammonium ($NH_4^+$)**

**Here, we referred to the equation of the N2 fixation process which is:**

$$N_2 + 8\ H^+ + 8\ e^- + 16ATP \rightarrow 2\ NH_3 + H_2 + 16\ ADP + 16\ Pi$$

p 2 Line 7-11: "In the oligotrophic...photic layer." Run-on sentence.

**The sentence has been rewritten page 2 line 25 as follows: "In the oligotrophic ocean, N availability often limits phytoplankton growth (e.g. Moore et al., 2013) and $N_2$ fixation sustains a significant part of new primary production (PP, i.e. the production unrelated to internal recycling of organic matter in the photic layer), such as in the North (Karl et al., 1997) and South Pacific Ocean (Moutin et al., 2008), the western Mediterranean Sea (Garcia et al., 2006), or the tropical North Atlantic (Capone et al., 2005)."**

p 3 Line 8: "(see below)" This is not necessary.

**It has been deleted in the new version**

p 3 Lines 9-13: What region are these N2 fixation rates integrated over?

**In their study, Bonnet et al. (2017) have integrated the $N_2$ fixation rates over the photic layer, i.e. from surface to 0.1% of surface irradiance, as we did in this study.**

**Materials and Methods**

p 3 Line 22: "strong thermal stratification" Please provide data or a reference for this statement or delete it. What qualifies as "strong" thermal stratification?

**Reviewer #2 is right that austral summer condition is sufficient to describe the stratification occurring in this area, thus we deleted the statement.**

p 3 Line 23: "...along a west-east...French Polynesia." Please also indicate the location using lat/long, as the precise locations of New Caledonia and French Polynesia do not spring immediately to mind.

**We understand that the precise locations of New Caledonia and French Polynesia do not immediately spring to mind, thus we have added the location using lat/long in the text page 4 line 17 "…from New-Caledonia (22°00'S – 166°00'E) to French Polynesia (17°30'S – 149°30'W)."**

p 3 Line 25: "diazotrophs" should be "diazotroph"

**This has been corrected in the new version**

p 3 Line 29: "firstly" should be "first"

**This has been corrected in the new version**

p 4 Line 19: Doesn't PAR stand for "Photosynthetically Active Radiation", not "Photosynthetically Available Radiation", as written here?

**We confirm that PAR stands for "Photosynthetically Available Radiation" as written in the manuscript.**

p 4 Line 19: Does "fluorescence" refer to chlorophyll a fluorescence, or some other set of wavelengths? Please be specific.

> **Here, fluorescence referred to chlorophyll a fluorescence. We have specified this in the new version page 5 line 13 as following "…and chlorophyll a fluorescence during the station's occupation…"**

p 5 Line 3: "In situ Chl a concentration was..." Should be "In situ Chl a concentrations were..."

> **This has been corrected in the new version**

p 5 Line 3: Was the AquaTraka III an in situ sensor attached to and deployed with the CTD package? So, this was used instead of the SeaBird chl sensor? Or is the AquaTraka for shipboard measurements? This is a little unclear because I expected to see the SeaBird chl sensor used with the SeaBird package. Please clarify.

> **Yes, the AquaTraka III is an in situ sensor attached to and deployed with the CTD package. No, it was not used instead of a SeaBird chl sensor. It is one of the sensors that could be used with the SeaBird CTD/Rosette System. The CTD configuration is proposed there: https://outpace.mio.univ-amu.fr/spip.php?article137. This has been clarified in the new version page 5 line 28 in the following way: "…fluorescence measurements performed with a AquaTraka III (Chelsea Technologies Group Ltd) sensor mounted on the CTD…"**

p 5 Line 4-7: Please include more details on collection of the NO3 and PO4 samples were they filtered? Were they stored or run immediately at sea?

> **Two samples for $NO_3^-$ and $PO_4^{3-}$ concentration measurements were collected from Niskin bottles in 20-mL Polyethylene bottles. After filtration, one sample was directly analyzed on-board and the other poisoned with 50 μl $HgCl_2$ (20 g $L^{-1}$) and stored for analysis after the cruise in the laboratory.**

> **The details have been added in the new version page 5 line 32 in the following way: "Phosphate ($PO_4^{3-}$) and $NO_3^-$ concentrations were measured daily at 12 depths from the surface to 200 m on each nutrient CTD cast using standard colorimetric procedures (Aminot and Kérouel, 2007) on a AA3 AutoAnalyzer (Seal-Analytical). After filtration, one sample was directly analyzed on-board and the other poisoned with 50μl HgCl2 (20 g $L^{-1}$) and stored for analysis after the cruise in the laboratory. The quantification limits were 0.05 μmol $L^{-1}$ for $PO_4^{3-}$ and $NO_3^-$."**

p 5 Line 9: Were these incubations in glass bottles? please indicate. How many replicates were used?

> **The PP rates were measured in triplicate using the [14]C tracer method (Moutin and Raimbault, 2002) in 150-mL polycarbonate bottles.**

> **The details have been added in the new version page 6 line 4 in the following way: "PP was measured in triplicate using the [14]C tracer method (Moutin and Raimbault, 2002). Samples were incubated in 320 mL polycarbonate bottles on the in situ drifting production line…"**

p 5 Line 13-17: Were these integrated over the upper 200 m? It looks like that's so, but please state it specifically.

> **The integrations were performed from the surface to 20 m below the deepest depth. As the depths of sampling were chosen according to surface irradiance levels to cover the**

**entire photic layer, the deepest depth (0.1 % of surface irradiance) was different at each station: 105 m, 80 m and 180 m for LD A, LD Band LD C, respectively (as mentioned p5 line 10-12). Thus, the integration was performed from surface to 125 m, 100 m and 200 m for LD A, LD B and LD C, respectively.**

**This has been specified in the new version of the manuscript page 6 line 9 in the following way: "Integrated N-PP (iN-PP) over the studied (surface to 125m, 100 m and 200 m for LD A, LD B and LD C, respectively) layer was calculated by using…"**

p 5 Line 19: "DIN" should include ammonium, which can account for ~40% of total N deposition (Dentener et al. 2006). I suggest renaming the combined NO3/NO2 term to NOx or similar to avoid confusion.

**In accordance with this comment, we have renamed the combined NO3/NO2 term as NOx.**

p 5 Line 19-22: Since the reference is "submitted", these methods should be explained in greater detail or additional references given.

**We completely agree with this comment. Thus, we have completed this section (page 6 line 19) in collaboration with Cécile Guieu who performed the measurements. Cécile Guieu will therefore be added as co-author of the manuscript in its final version.**

**The new version of the section is:**

**"N atmospheric deposition ($NO_3^-$ and $NO_2^-$ (nitrite), hereafter called NOx) was quantified along the transect after dissolution of aerosols collected continuously during the transect, as described in Guieu et al. (in rev.). Briefly, the sampling device, designed to avoid ship contamination, was installed at the look-out post in the front of the ship, collected aerosols at ~20 L min$^{-1}$ on onto polycarbonate, 47-mm diameter, 0.45-µm porosity (previously acid-cleaned with a 2% solution of HCl (Merck, Ultrapur, Germany) and thoroughly rinsed with ultra-pure water and dried under a laminar flow bench and stored in acid-cleaned Petri dishes). Dissolution experiments to determine NOx released in surface seawater after deposition were performed on board using acid-cleaned Sartorius filtration units (volume 0.250 L) and filtered surface (5m) seawater. Each sample was subjected to two contact times: the first contact was at one minute, and the second contact was at 24 hours. NOx was analyzed using a 1-m long Liquid Waveguide Capillary Cells (LWCC) made of quartz capillary tubing, following the protocol described in Louis et al., 2015. An extrapolated NOx from dry deposition was estimated on the basis of a deposition velocity of submicronic particles (0.4 m s$^{-1}$; Vong et al., 2010)."**

**Louis, J., Bressac, M., Pedrotti, M. L., & Guieu, C. (2015). Dissolved inorganic nitrogen and phosphorus dynamics in seawater following an artificial Saharan dust deposition event. Frontiers in Marine Science, 2, 27.**

**Vong, R. J., Vong, I. J., Vickers, D., and Covert, D. S.: Size-dependent aerosol deposition velocities during BEARPEX'07, Atmos. Chem. Phys., 10, 5749-5758, doi:10.5194/acp-10-5749-2010, 2010**

p 5 Line 26: "same depths as for NPP" Were these 15N2 incubations performed in the same bottles as the 14C incubations?

**The $^{15}N_2$ incubations were not performed in the same bottles as the $^{14}C$ incubations.**

**This should be clearer in the new version of the manuscript as we mentioned that "Samples were incubated in 320 mL polycarbonate bottles on the in situ drifting production line…" in the method section for the PP, in the new version page 6 line 4.**

p 6 Line 4: The danger of "trace metal contamination" is mentioned. Was the water for incubations collected using trace metal clean methods?

**Indeed, we mentioned the danger of "trace metal contamination", referring to this danger while preparing the $^{15}N_2$-enriched seawater using the method of Mohr et al. (2010). We did not collect water using in the trace metal clean container for logistical reasons (no space for that) but all bottles, tubing were carefully washed with 10% HCl to keep potential contaminations at their minimum.**

p 6 Line 13: "blue screening" I assume that the purpose of blue screening was to alter the quality of incident light on the bottles. Please provide additional information on the change in quality to the incident light achieved with the blue screening. Also, was the quantity of incident light altered (i.e. by using different sizes mesh screen or some other neutral density filter) in the deckboard incubators?

**By "blue screening", we mean transparent blue sheet filters from light blue to dark blue which were fixed on the faces of the incubator and filtered the incident light from 75 to 0.1 % of the incident light, respectively to the depth of sampling. Thus, the quantity of light that reached the bottle was deliberately altered to copy the in-situ light. Information concerning incubators used in the study are available on 'https://outpace.mio.univ-amu.fr/spip.php?article135'.**

**Thus we have added this reference in the text page 7 line 27 as following "…bottles were incubated in on-deck incubators (see details on https://outpace.mio.univ-amu.fr/spip.php?article135) equipped with circulating seawater…"**

p 6 Line 16: What is meant by "gentle filtration"?

**By "gentle filtration" we mean low pressure (<0.2 bar) filtration that does not damage and blow up the cells. This has been specified in the new version page 7 line 32 as following: "…were stopped by gentle filtration (<0.2 bar) of the samples…"**

p 6 Line 21-22: Why was the initial PN only measured at two depths rather than for each rate measurement? Has it been determined that two depths are sufficient? If so, please be specific. How exactly were these two measurements used as 'initial' measurements for each rate measurement? Were they averaged and then used for all rates at that stations or for all stations or some other method? please specify. How many replicates were collected per depth for the initial measurements?

**The natural PN $\delta^{15}N$ did not vary within the photic layer, that is why we used the average of the 2 values for our rate calculations. In the companion paper Benavides et al. (this issue) in which we measured aphotic $N_2$ fixation, we performed $\delta^{15}N$ measurements of PN at each depth because the values were much more variable and could influence our rate calculations, which was not the case in the photic layer.**

**Thus we have modified the text page 8 line 5 as following: "…the [15]N-enrichement of the ambient (unlabeled) PN was measured in one replicate at each station at the DCM and the subsurface…"**

p 6 Line 25: How was the nitracline depth calculated? I am unclear on what NO3 is. Is this the density where the nitracline occurs?

**The top nitracline depth was graphically determined at the depth where the NO3 concentration was detectable. As there were internal waves (Bouruet-Aubertot et al., this issue) at LD A, the top nitracline depth varied daily, but the slope (nitrate concentration against density) was constant. We observed that using the nitrate vs. density profiles instead of nitrate vs. depth profiles, the top of the nitracline was always at the same iso-density. For that purpose, we decided to work with density instead of depth as is usually the case. Thus, $\rho_{NO3}$ corresponds to the isopicnal associated with the top of the nitracline.**

p 6 Line 25- p 7 Line 3: Is there a reference for this flux calculation?

**There is no reference for this flux calculation, this is why we have detailed it in the method section.**

p 7 Line 6: Why were these depths chosen?

**First, we decide to use the same configuration at the 3 LD stations. The first trap was deployed at 150 m because it is always below the base of the photic layer, and therefore below the productivity layer. 520 m was used because most of the diel zooplankton vertical migrations stopped above this depth, and 330 m was chosen as an intermediary depth.**

**Thus, we have specified this in the new version page 8 line 23 in the following way: "Particulate matter export was quantified with three PPS5 sediment traps (1 m$^2$ surface collection, Technicap, France) deployed for 5 days at 150, 330 and 520 m at each LD station (Fig. 1). We decide to use the same configuration at the 3 LD stations. The first trap was deployed at 150 m as it is always below the base of the photic layer, and therefore below the productivity layer. 520 m was used because most of the diel zooplankton vertical migrations stopped above this depth, and 330 m was chosen as an intermediary depth…"**

p 7 Line 8: "...buffered solution of formaldehyde..." Please include a reference for this statement.

**The sentence: "The flasks were previously filled with a buffered solution of formaldehyde (final conc. 2 %) and were stored at 4 °C after collection until analysis to prevent degradation of the collected material." page 8 line 29 was changed by:" The flasks were previously filled with a buffered solution (Sodium borate) of formaldehyde (final conc. 2 % and pH=8) and were stored at 4 °C after collection until analysis to prevent degradation of the collected material."**

p 7 Line 12: Please specify what a "swimmer" is. Were these living organisms found in the 'fresh' trap, or were they in the preserved samples too? If the latter, how were they separated from the rest of the material? Were these a certain size class of organisms? Why were they analyzed separately?

**Swimmers are used to describe zooplankton organisms recovered in the traps (both fresh and preserved samples). The word "swimmers" is used because it is thought that mainly living zooplankton was poisoned and recovered in the traps. They were manually handpicked and analyzed separately because they cannot be considered as settling particulate matter.**

p 8 Line 6: What is meant by "gently filtered"?

**By "gentle filtration" we mean low pressure filtration (<0.2 bar) that do not damage and blow up the cells. This has been specified page 9 line 33.**

p 8 Line 11-12: "The C contents...Luo et al. 2012)." This belongs in the results section.

**As this sentence is necessary to validate our method, we have included this sentence in the Methods section. Thus, we refrain to change the text.**

p 8 Line 11-16: "As DDAs...asymbiotic)." It is unclear why this information is present; I thought the cell C content was only determined for Tricho and UCYN. Please add a topic sentence to this paragraph explaining for which organisms biovolume and cell C content were determined and which were determined directly v indirectly.

**The cell C content was directly determined for *Trichodesmium* and UCYN, and indirectly for DDAs. This has been specified in the new version page 9 line 31 in the following way: "To determine directly the biovolume and C content of diazotrophs, cell sizes of *Trichodesmium* and UCYN-B were determined in samples from the photic layer of each LD station." and page 10 line 8 in the following way "As DDAs were not easily identified on the filters the C content was indirectly estimated. We used a C content of…"**

**Results**

p 9 Line 2: "...with the minimum concentration located at 60 m." It's best to refrain from calling this a minimum because the difference in concentration is slight and there are no replicates.

**We agree with this comment, this part of the sentence has been deleted. The new sentence in the new version page 10 line 26 is "At LD C, $PO_4^{3-}$ concentrations were always above the quantification limit and varied from 0.11 to 0.17 µmol $L^{-1}$ in the 0-120 m layer."**

p 9 Line 5: "...while the DCM was deepening from 25 m to 70 m..." I suggest restating this as "...the depth of the DCM increased from 25 to 70 m during the five days that the station was occupied..." or something similar

**We agree with this comment, thus we have rewritten the sentence page 10 line 30 as proposed by reviewer #2 in the following way: "At LD B, the $\rho_{NO3}$ was located between 100 and 120 m, while the depth of the DCM increased from 25 to 70 m during the five days that the station was occupied"**

p 9 Line 6: "...varying simultaneously between 115 and 155 m." This wording is confusing.

**We have changed "simultaneously" to "concurrently" in the new version.**

p 9 Line 11: "...below 0.2..." Is it below 0.2 or equal to 0.2, as Table 4 indicates? Where is the comparison of total N deposition and NO3+NO2 deposition, as suggested in the Methods?

**We have rewritten this section page 11 line 7 in the following way:**

**"Nitrate dissolution from aerosols occurred rapidly releasing in seawater on average 1.8 nmol.m$^{-3}$ (26 ng m$^{-3}$) dissolved inorganic nitrogen. Nitrate appeared to be as nitrate aerosol since no correlation was observed between nitrate and Fe, Si, Na, Cl (K. Desboeufs, pers. Com., 2017), precluding a mixing with ash or sea salt. Extrapolated dry deposition flux (Table 4) was on average 630 ± 329 nmol.m$^{-2}$.d$^{-1}$ (3.22 ± 1.7 mg m$^{-2}$ yr$^{-1}$)."**

**The comparison of total N deposition and NO3+NO2 deposition is discussed in the Discussion section in the new version. Thus we have changed have changed the text in the 'Discussion- Contribution of N$_2$ fixation to new N input in the WTSP' section page 14 line 3 in the following way: "Extrapolated NOx deposition from the atmosphere during OUTPACE (range: 0.34 – 1.05 µmol m$^{-2}$ d$^{-1}$) were one order of magnitude lower than predicted with major uncertainties by global models that include wet and gas deposition for that region (Kanakidou et al., 2012). Our flux could be an underestimation as it represents only dry deposition and as gas and organic forms were not measured. At the global scale and depending on the location, organic nitrogen could represent up to 90 % of N atmospheric deposition (Kanakidou et al., 2012), and NH$_4^+$ could account for ~40% (Dentener et al. 2006). Even if we double our estimated deposition flux, atmospheric deposition still remained low (< 1.5 %) and consequently represented a minor contribution of the new N input (Table 2). This negligible contribution of atmospheric input to the overall N budget (less than 1.5 %) therefore implies an important contribution of other terms, such as N$_2$ fixation."**

p 9 Line 13: "...0.0-19.3..." What was the detection limit of this method? Instead of reporting "0.0" please report rates as below the limit of detection if they are. Two recent publications depict methods for calculating the detection limit of these rates (Gradoville et al. 2017; Jayakumar et al. 2017). Accounting for a DL will be especially important for LDC where the rates were very low and may be undetectable.

**The minimum quantifiable rate calculated using standard propagation of errors via the observed variability between replicate samples, measured according to Gradoville et al. (2017), was 0.035 nmol N L$^{-1}$ d$^{-1}$. We have indicated <QL instead of "0.0" in the new version of the manuscript.**

p 9 Line 17: "...below the quantification limit..." So there is a quantification limit! How was it calculated and what was it?

**We have calculated minimum quantifiable rate according to Gradoville et al. (2017) that was 0.035 nmol N L$^{-1}$ d$^{-1}$in our study. This has been added in the new version (Method section) page 8 line 6 in the following way: "The minimum quantifiable rate calculated using standard propagation of errors via the observed variability between replicate samples, measured according to Gradoville et al. (2017), was 0.035 nmol N L$^{-1}$ d$^{-1}$."**

p 9 Line 17-19: Please compare the N2 fixation rates from days 1-3 using the Mann Whitney test, as was done comparing the in situ and shipboard incubations.

**This comparison has been performed, and the rates from day 1 and day 3 were not statistically different. This comparison has been performed for all daily rates at each station, and no statistical difference was found. We have specified this in the new version page 11 line 24 in the following way "In addition, N₂ fixation rates from days 1-2-3 were not statistically different each other (Mann-Whitney paired test, p<0.05)."**

p 9 Line 23: "...the maximum at 40 m at LDC..." Was this small rate actually above the detection limit though?

**The maximum of 2.6 nmol N L⁻¹ d⁻¹ measured at 40 m at LD C was above the minimum quantifiable rate of 0.035 nmol N L⁻¹ d⁻¹. This should be clearer in the new version as we specified the minimum quantifiable rate in the Method section page 8 line 6.**

p 9 Line 24-27: It is nice to see this comparison! It makes me feel better about all the shipboard measurements in the literature!

**We appreciate the comment**

p 9 Line 31-p10 Line 1: "Strong time...(table 1)." This is repetitive with the previous sentence.

**This sentence has been deleted and merged with the previous sentence page 12 line 2 in the following way: "The averaged NO₃⁻ input through vertical turbulent diffusion showed strong time variability with a typical standard deviation of the same order as the mean value (Table 3), and a strong contrast between the western station LD A and the two other stations, with mean values equal to 24.4 ± 24.4 µmol N m⁻² d⁻¹ at LD A and 6.7 ± 5.3 and 4.8 ± 2.2 µmol N m⁻² d⁻¹ at LD B and LD C, respectively (Fig. 4)."**

p 10 Line 22: "...maximum export...at LD C." PC export for LDB and LDC do not look significantly different, based on the overlap in standard deviation (Table 2).

**We agree with this comment, thus the sentence has been rewritten page 12 line 27 in the following way: "…maximum export at 150 m at LD A, minimum export at LD B and LD C…"**

p 10 Line 21-24: Why is there a discussion of trends in PC but not PN and PP? Line 24: "...averaged C:N ratios...LDC." Are these significantly different?

**In this study, we decided to focus on the C and N export, associated with N₂ fixation. We agree with the comment of Reviewer #2 and we thus deleted the PP and Zoo-P data from the study.**

p 10 Line 31: "below detection limit" What was the detection limit? Elsewhere in the paper, (i.e. for NO3 and PO4 concentrations), a "quantification limit" is referenced. What is the difference between that quantification limit and a detection limit? Please define these terms in the Methods section and be consistent with their usage.

**The limit of detection refers to the lowest concentration that can be detected at a specified level of confidence. The limit of quantification refers to the lowest concentration at which the performance of a method/measurement system is acceptable for a specified use.**

**The limit here is a quantification limit, as for NO₃⁻ and PO₄³⁻ concentrations and N₂ fixation measurements. The quantification limit was referenced in the Methods section page 6 line 1 in the following way: "The quantification limits were 0.05 μmol L⁻¹ for PO₄³⁻ and NO₃⁻.". All the terms have been corrected in the new version.**

**Discussion**

p 11 Line 27: "...atmospheric deposition...lower end of fluxes..." I am curious what the mixed layer depth was at these stations, as atmospheric deposition is really a flux only to the mixed layer, not the entire euphotic zone. Since the flux is so small, it likely doesn't matter for this study, but it would be nice to see those depths in a Table for readers who study atmospheric deposition to the mixed layer.

Depths of mixed layers were around 20 m and are described in a companion paper by Moutin et al. (this issue). We focus on the productive layers and prefer not to add additionnal values not necessary for our calculation. p 12 Line 8-17: "In this study...to significant bias." This is a really good point, and it's nice to see a snapshot of this variability. It does make one wonder, however, if the snapshot had been even longer, how would the results have varied, given the large differences seen between the use of an average Kz and the instantaneous Kz values and the seeming randomness of the spikes. Is 5 days long enough? A sentence or two addressing this concern would be useful.

**The differences in estimated fluxes between the stations is mainly attributable to differences in Kz between the stations. The gradients of concentration varied little between stations (table 1). The turbulent diffusion contrast is a robust result despite the variability of the signal (Kz and therefore the flux). This is explained by the fact that shear instability, the dominant process triggering turbulence at long stations, is more intense in LDA because of higher shear, by a factor of about 3. Due to this, we suppose that if this had been studied longer than 5 days, the results would have been in the same order of magnitude. This point has been studied in depth in Bouruet-Aubertot et al. (this issue), that we refer to: "The contrasted NO₃⁻ input observed at the three stations results from the high variability in turbulence along the west-east transects (Bouruet-Aubertot et al., this issue)."**

p 12 Line 22: "N m-4" What is a m-4?

**The unit of the gradient is unit of concentration divided per unit of depth, expressed in [μmol N m⁻³]/[m] and simplified as μmol N m⁻⁴, thus we keep this unit.**

p 15 Line 11: "such as DON and/or zooplankton export" Please elaborate. Dissolved compounds are not exported like particles are; they do not sink. Or is this a reference to conversion of fixed N to the DON pool, followed up uptake and subsequent export? Both a direct DON flux and a delayed N flux following DON uptake are worth mentioning. Also, what about fixed N that is released as ammonium? Ammonium and many simple DON compounds (amino acids, urea...) cycle very quickly and likely would be taken up before they could be mixed downward. Also, it is unclear why DON and zooplankton export are lumped together like this... Is this referring to an active zooplankton flux (which is a completely different process than the DON pathway) or to the sinking out of dead zooplankton later?

**In this sentence, we assessed the hypothesis that could explain the imbalance between new and exported N. We presented 3 hypotheses, and the third one is "processes other than particle export, such as DON and/or zooplankton export". Those two come within the same hypothesis because they refer to an "export", but yes, they are not exported via**

the same process. They are lumped together in this sentence, but below in the section, DON and zooplankton export are discussed separately.

We are aware that DON is not exported as particles and does not sink. The export of DON (whatever the source of DON: i.e. phytoplankton or zooplankton) by turbulent diffusion was estimated by Moutin et al. (this issue) and the important result is that it cannot explain the imbalance. As reviewer #2 refers to the conversion of fixed N to the DON pool, note that Caffin et al. (this issue) have studied the transfer of the fixed N to the dissolved pool (NH4 and DON) and to plankton communities. This N transferred to the dissolved pool, was re-uptaken and subsequently exported as reviewer #2 mentioned, and finally recovered in the sediment traps particulate matter.

p 16 Line 1: "zooplankton itself is" should be "zooplankton themselves are" Please make similar corrections throughout.

**This has been corrected in the new version.**

**Conclusion**

p 16 Line 16-17: "contributed to ~15-21%" and "...and to ~3%..." Delete "to".

**This has been corrected in the new version**

**Tables and Figures**

The tables seem to be out of the order that they are mentioned in the text. Please check this.

**This has been corrected in the new version**

Table 1. Please define Kz in the figure legend. For this and all figures and tables, please indicate what the error is. Standard deviation? Standard error? n = ?

**The legend has been rewritten in the following way: "Mean turbulent diffusion coefficient (Kz), mean nitracline gradient, mean $NO_3^-$ flux and associated standard deviations (n=3) over the station occupation at LD A, LD B and LD C at the top nitracline and at the maximum gradient."**

Table 2. What is "DWmatter"? Please indicate this in the table legend or with a footnote to the table. Why are the errors italicized in this table and Table 3 but not in Table 1? Please standardize this across tables. If PC, PN, and PP should be PC flux, PN flux, and PP flux, please indicate that. Please propogate the error of PC, N, and P into the C:N:P ratio calculation.

**We do not present the Phosphorus values anymore as we do not use them in the study. We have specified dry weight (DW) in the table legend. The legend has been rewritten in the following way: "Sediment trap data at the three LD stations. Depth of collection, mean mass flux of dry weight (DW) matter, particulate C and N flux, mean C:N molar ratio. No data was collected at LD C at 520 m. No data was collected at LD C at 520 m." The errors have been standardized across tables in regular font. We have indicated that PC and PN are PC flux and PN flux in the table. The C:N ratio were presented without propagation error to keep the table and the message clear and to give a global overview of the ratio at each depth. We allowed ourselves to do that as the standard deviation was presented in each of the previous terms.**

Table 3. Please apply the comments from Tables 1 and 2 to this table. Also, please use a standard number of significant digits for all measurements (i.e. LD A 300 m is inconsistent).

**This has been corrected in the new version. The legend has been rewritten in the following way: "Zooplankton sediment traps data at the three LD stations. Depth of sampling, mean dry weight (DW) zooplankton recovered in the traps, C and N associated to zooplankton". Also, we do not present the Phosphorus values anymore as we do not use them in the study.**

Table 4: Please include the standard deviation or error of these values by propogating the error from the measurements. please define all non-obvious terms (i.e. d[NO3]/dp, NO3) and the calculation for the e-ratio in a footnote so that the reader does not have to dig through the text to find them. Is N2 fixation the in situ rate or the shipboard rate? Please use consistent significant digits (i.e. for NO3 diffusion and export N 150 m).

**The objective of this table is to show a global overview of the budget and the different fluxes, that is why we do not include standard deviations. All the standard deviations are given in the text or/and in other tables. We have deleted the term d[NO3]/dp from the table as it is not a useful information. Here, we present in situ rates, this has been specified in the new version of the table.**

Fig 1: This figure seems needlessly complicated and the vertical component does not seem spaceworthy. The advantage of the vertical component is to show where the production arrays and traps were deployed, but it's difficult to tell the exact depths in this figure. A simple map may be preferable.

**The purpose of this figure is to give a global map of the surface chl a, show the station position and show where the production arrays and traps were deployed, in a same picture. We understand that it's difficult to tell the exact depths, that is why we have added the exact depth in the legend. We have decided to keep this figure in this version.**

Fig. 2: Please indicate on the figure itself which units correspond to which parameter, as it is a bit confusing in current form (particularly for phosphate). It may be instructive to use a different scale for the N fixation rates in the third panel, since they can't be seen on this scale.

**This has been corrected in the new version, we have presented a separate axe for each unit/parameter.**

Fig 3: This is a cool figure! Nice dataset! I am a little confused on how 1% of surface PAR was calculated at night. Should there be breaks in the dotted line for nighttime?

**The 1 % of PAR was calculated only at midday (yellow crosses) and then linked to the previous and following day by the yellow dotted line. Even during the night 1 % of PAR is present at depth.**

**Response to Carolin Löscher (Referee #3)**

**We thank Carolin Löscher for the time and effort devoted to the review of the manuscript. Below, we reproduce the reviewer's comments and address her concerns point by point. The reviewer's comments are copied in regular font, with our responses in green.**

**We are responding to this review a long time after it was published on the *Biogeosciences Discussion* online forum because we had to submit the companion paper by Caffin et al. (this issue), cited in this article, before the closure of OUTPACE's special issue on December 31, 2017. We recently obtained a 3 months extension.**

The manuscript by Caffin et al. describes budgets of nitrogen at three stations in the oligotrophic western tropical South Pacific using a Lagrangian strategy thus being able to track the same water mass over time. The study reports exceptionally high N2 fixation rates and a corresponding high contribution of N2 fixation impacted material to export production. The study is very interesting to me particularly because of an approach that is more innovative than what is classically used when it comes to N budgets and N2 fixation. Overall, the paper doesn't need much changes to get into shape for publication, the study is clear and well presented. I personally think the title is not the best choice, it could make a statement on what the prominent role of N2 fixation is.

**In accordance with the comments by Anonymous Referee #1, we have changed the title to "N$_2$ fixation as a dominant new N source in the Western Tropical South Pacific Ocean (OUTPACE cruise)"**

In order to make the study entirely convincing I have some main aspects, which should be and easily could be clarified:

1. The good old topic on using the bubble method: It is not convincing to just measure the dissolved fraction and not give any ranges. There are concerns with that method, everyone knows that, if you claim it is ok to use it you should have done a comparative measurement at least for some of your samples using both methods. In this context, I either need to see the data on the dissolved vs. particulate phase, or the rates have to be presented as potential rates.

**We agree with this comment and try to give below some more explanations regarding our methodological choice. We only report particulate N2 fixation rates in this study. We decided to use the 'bubble' method to avoid any trace metal and DOM contaminations (frequent in the study area, Benavides et al., 2017; Moisander et al., 2011) in the incubation bottles and avoid any potential over-estimation of rates. However, we are aware that this method potentially underestimates N$_2$ fixation rates (Großkopf et al., 2012; Mohr et al., 2010; Wilson et al., 2012) due to the incomplete (and gradually increasing during the incubation period) equilibration of the $^{15}$N$_2$ in the incubation bottles when injected as a bubble. This results in a lower $^{15}$N/$^{14}$N ratio of the N$_2$ pool available for N$_2$ fixation (the term A$_{N2}$ used in the Montoya et al. (1996) equation) as compared to the theoretical value calculated based on gas constants, and therefore potentially leads to underestimated rates in some studies, whereas other studies do not see any significant differences between both methods (Bonnet et al., 2016c; Shiozaki et al., 2015). Here we paid careful attention to accurately measure the term A$_{N2}$ to minimize any potential underestimation. The average values at the end of the incubation period was 7.548 ± 0.557 atom%, which is lower than the theoretical value of ~8.2 atom% based on gas constants calculations (Weiss, 1970). We are aware the dissolution kinetics of**

15N2 in the incubation bottles is progressive along the 24 h of incubation (Mohr et al., 2010a). Therefore, the $^{15}N$ enrichment of the $N_2$ pool measured with the MIMS likely represent maximum values, and the $N_2$ fixation rates provided in this study represent minimum values. This has been added in the discussion section page 18 line 30 'Methodological underestimation leads to a possible higher contribution of $N_2$ fixation' as following:

"In this study we intentionally used the 'bubble method' to measure $N_2$ fixation rates, considering the small differences observed between the two methods in Pacific waters (Bonnet et al. 2016b, Shiozaki et al., 2015) and the high risk of sample contamination involved when manipulating sample seawater to prepare dissolved $^{15}N_2$ (Klawonn et al., 2015). In addition to the contamination issues, preparing dissolved $^{15}N_2$ on board represents additional time with samples sitting on the bench or rosette before incubation, which is especially critical in tropical environments. To reduce any potential underestimation, we measured the $^{15}N$ enrichment of the $N_2$ pool at the end of the incubation (7.548 ± 0.557 atom%, Bonnet et al., this issue), which was lower than the theoretical value of ~8.2 atom% based on gas constants calculations (Weiss, 1970). We are aware that the dissolution kinetics of $^{15}N_2$ in the incubation bottles is progressive along the 24 h of incubation (Mohr et al., 2010a). Therefore, the $^{15}N$ enrichment of the $N_2$ pool measured with the MIMS likely represent maximum values, and the $N_2$ fixation rates provided in this study represent minimum values. This reinforces the conclusions of this study regarding the prominent role of $N_2$ fixation in this region. Großkopft et al. (2012) found that the discrepancy between both methods was more important when UCYN dominates the diazotroph community as compared to when *Trichodesmium* dominates. Consequently, $N_2$ fixation rates in this study are potentially more underestimated in SPG waters than in MA waters. By applying the maximum factor of underestimation found by Großkopft et al (1.7), $N_2$ fixation in SPG waters would have been higher (100 μmol N m$^{-1}$ d$^{-1}$ instead of 59), which is still far lower than in MA waters and does not change de conclusions of this study."

2. In the same context, I don't know the gas quality of the company you bought from, but I assume you checked for purity as recommended in the Dabundo paper. Otherwise the high rates may as well come from an ammonia incorporation or similar. Please present your quality check, here.

We are aware that Dabundo et al. (2014) reports potential contamination of some commercial $^{15}N_2$ gas stocks with $^{15}N$-enriched $NH_4^+$, $NO_3^-$ and/or $NO_2^-$, and nitrous oxide ($N_2O$). In their study, Dabundo et al. (2014) analysed various brands of $^{15}N_2$ (Sigma, Cambridge Isotopes, Campro Scientific) and found that the Cambridge Isotopes brand (i.e., the one used in these studies) contained low concentrations of $^{15}N$ contaminants, and the potential overestimated $N_2$ fixation rates modeled using this contamination level would range from undetectable to 0.02 nmol N L$^{-1}$ d$^{-1}$. The rates measured in this study were on average ~10 nmol N L$^{-1}$ d$^{-1}$, suggesting that stock contamination would be too low to affect the results reported here.

To verify this, the Cambridge Isotopes batches that are routinely used by our team has been analyzed for potential contamination in Julie Granger and Richard Dabundo's lab, and this confirmed that the contamination of the $^{15}N_2$ gas stock was low: 1.4 x 10$^{-8}$ mol of

$^{15}NO_3^-$ per mol of $^{15}N_2$, and $1.1 \times 10^{-8}$ mol $NH_4^+$ per mol of $^{15}N_2$. The application of this contamination level to our samples using the model described in Dabundo et al. (2014) indicates that our rates could only be overestimated by 0.01 to 0.12 %. We thus confirmed that the stock contamination issue did not affect the results reported here. This has been added to the method section page 7 line 5 "The purity of the $^{15}N_2$ Cambridge isotopes stocks was previously checked by Dabundo et al. (2014) and more recently by (Benavides et al., 2015) and (Bonnet et al., 2016a). They were found to be lower than $2 \times 10^{-8}$ mol:mol of $^{15}N_2$, leading to a potential $N_2$ fixation rates overestimation of <1 %"

3. In addition, ammonia background measurements, fluxes and inputs are not mentioned- this would add enormous value to the stud, so please present if available. As you are making a suggestion on zooplankton moderated export, ammonia is a good part of this, too.

In our study, we focused on new N inputs (i.e. atmospheric deposition, $N_2$ fixation and vertical nitrate diffusion) thus associated with new production. Ammonium concentrations were measured at the three LD stations (available on http://www.obs-vlfr.fr/proof/php/outpace/outpace.php) and was low (in the nM range) in the photic layer. Thus we refrain from presenting ammonium data in our study, because ammonium fluxes were not measured during the cruise where the focus was essentially on new N budgets.

4. No sequencing was performed and no single cell rates were determined- how can you interpret on the key N2 fixers if you just look at 6 clusters via qPCR? What makes you conclude that Trichodesmium or UCYN clusters are important if you don't assess which diazotrophs are there?

We agree on this comment. Careful microscopic analyses have been performed in addition to qPCR at all stations (qPCR data are reported in Stenegren et al., 2018) and do not reveal any additional diazotroph groups compared to qPCR results at the studied stations (at least those who can be determined microscopically), except some very few *Katagnymene spiralis*. For the groups who cannot be determined microscopically, UCYN-A were determined by qPCR and were scarce. In addition, Pia Moisander and Mar Benavides did some RNA sequencing in surface at LD stations ABC and found that Trichodesmium was dominating the diazotrophic community expression at LDA and LDB and the nifH expression of heterotrophic diazotrophs was almost not detectable. Finally, Bonnet et al. (this issue) performed some group specific $N_2$ fixation rate measurements at stations LD A and B and found that Trichodesmium was the major $N_2$ fixing organisms, accounting for 47-84 % of bulk $N_2$ fixation.

**Response to Anonymous Referee #4**

**We thank Anonymous Referee #4 for the time and effort devoted to the review of the manuscript. Below, we reproduce the reviewer's comments and address their concerns in each case. The reviewer's comments are copied below in regular font, with our responses in orange. We are responding to this review a long time after it was published on the *Biogeosciences Discussion* online forum because we had to submit the companion paper by Caffin et al. (this issue), cited in this article, before the closure of OUTPACE's special issue on December 31, 2017.**

**Summary Statement**

Caffin et al. constructed a nitrogen budget for three stations in the western tropical Pacific Ocean by quantifying N2 fixation, NO3 diffusion, atmospheric deposition, and PN export. Overall, the study seems to be well-conducted, arguments are supported by data, and the paper is well-cited. There are some relatively minor issues, mostly with the presentation, as described below. The manuscript requires a thorough editing to correct awkward word choices, punctuation errors, and confusing text. The main point I found that was missing from the paper was a definition of the system being studied. When the authors attempted to describe the system and site selection choices, the text was confusing and too vague, so this area of the paper could be improved. Some additional details are also missing from the methods and should be included. The conclusions section fell a bit flat and could be bolstered by putting the study findings into a better context relative to filling information and data gaps and describing the overall importance of the study results for our understanding of the global ocean. None of these issues represent serious barriers to publication, in my view, and only minor revisions are needed.

**Again, we thank Anonymous Referee #4 who highlighted some 'relatively minor issues', 'mostly with the presentation', and we will respond point by point to the comments. Moreover, the text has been corrected by an English native speaker. We have included missing additional details regarding the method and opened up the conclusions with a view to making good the information gaps to enhance our understanding of the WTSP, that is a hot spot of $N_2$ fixation, and our understanding of extensive areas of the oligotrophic ocean.**

**Specific Comments**

**Abstract**

Overall, I found the Abstract was confusing. There is no clear direction, and the text jumps around from topic to topic without any clear context for the study or results. The concluding sentences do not place the study findings into any sort of importance relative to information and data gaps that we have for the WTSP (or other areas of the oligotrophic ocean). Why is the disequilibrium and apparent N accumulation important to describe?

P1, Lines 21-22 — Confusing sentence. Rewrite for clarity.

**We have rewritten the Abstract as suggested, see the new Abstract below.**

P1, Lines 24-25 — Is there more information on these locations, other than just DCM, that could be presented to give the reader a better idea of what these sampling locations are like?

We have rewritten the Abstract as suggested and indicate the strong nitracline depth differences, allowing the reader to understand that we sampled a strong oligotrophic gradient. The depth values were not indicated in the Abstract because they do not represent major findings.

New Abstract:

We performed nitrogen (N) budgets in the photic layer at three contrasting stations representing different trophic conditions in the western tropical South Pacific (WTSP) Ocean during austral summer conditions (Feb. Mar. 2015). Using a Lagrangian strategy, we sampled the same water mass for the entire duration of each long duration (5 days) station, allowing us to consider only vertical exchanges for the budgets. We quantified all major vertical N fluxes both entering ($N_2$ fixation, nitrate turbulent diffusion, atmospheric deposition) and leaving the system (particulate N export). The 3 stations were characterized by strong nitracline and contrasted deep chlorophyll maximum depths, which was lower in the oligotrophic Melanesian archipelago (MA, stations LD A and LD B) than in the ultra-oligotrophic waters of the South Pacific gyre (SPG, station LD C). $N_2$ fixation rates were extremely high at both LD A ($593 \pm 51$ µmol N $m^{-2}$ $d^{-1}$) and LD B ($706 \pm 302$ µmol N $m^{-2}$ $d^{-1}$), and the diazotroph community was dominated by Trichodesmium. $N_2$ fixation rates were lower ($59 \pm 16$ µmol N $m^{-2}$ $d^{-1}$) at LD C, and the diazotroph community was dominated by unicellular N2-fixing cyanobacteria (UCYN). At all stations, $N_2$ fixation was the major source of new N (> 90 %) before atmospheric deposition and upward nitrate fluxes induced by turbulence. N2 fixation contributed circa 13-18 % of primary production in the MA region and 3 % in the SPG water and sustained nearly all new primary production at all stations. The e-ratio (e-ratio = particulate carbon export / primary production) was maximum at LD A (9.7 %) and was higher than the e-ratio in most studied oligotrophic regions (<5%), indicating a high efficiency of the WTSP to export carbon relative to primary production. The direct export of diazotrophs assessed by qPCR of the nifH gene in sediment traps represented up to 30.6 % of the PC export at LD A, while their contribution was 5 and < 0.1 % at LD B and LD C, respectively. At the three studied stations, the sum of all N input to the photic layer exceeded the N output through organic matter export. This disequilibrium leading to N accumulation in the upper layer appears as a characteristic of the WTSP during the summer season, although the role of zooplankton in export fluxes should be further investigated.

**Introduction**

P3, Lines 1-8 — The authors need to define the "system" they are talking about. What are the boundaries of the "system"?

Reviewer #4 is right and we have specified that our systems were the photic layers. Thus we modified the text page 3 line 23 on the following way: ". It requires the measurement of all major N fluxes both entering the photic layer ($N_2$ fixation, nitrate ($NO_3^-$) eddy diffusion, atmospheric deposition) and leaving the photic layer (PN export) with an adequate time frame (i.e. linking production and export)."

Are sediments included?

**The sediments were not included in our open ocean areas. It should be clearer now with the previous definition of the system**

What does "...with an adequate time frame under contrasting diazotroph communities' composition" mean?

**The adequate time frame is required to link export and production. The contrasting diazotroph communities' composition means if the diazotrophs are dominated by *Trichodesmium* or UCYN.**

**The sentence: "Studying the impact of $N_2$ fixation on PN export in the ocean and the relative role of each diazotroph group in this process are technically challenging. It requires the measurement of all major N fluxes both entering the system ($N_2$ fixation, nitrate ($NO_3^-$) eddy diffusion, atmospheric deposition) and leaving the system (PN export) with an adequate time frame under contrasting diazotroph communities' composition." page 3 line 22 has been rewritten as follows: "Studying the impact of $N_2$ fixation on PN export in the ocean and the relative role of each diazotroph group in this process is technically challenging. It requires the measurement of all major N fluxes both entering the photic layer ($N_2$ fixation, nitrate ($NO_3^-$) eddy diffusion, atmospheric deposition) and leaving the photic layer (PN export) with an adequate time frame (i.e. linking production and export). In addition, the sampling has to been performed under contrasting situations, for example when either *Trichodesmium* or UCYN dominate the diazotroph community, hence allowing to assess the potential role of each diazotroph group."**

Does "the same water mass" mean that horizontal water movement is not present/considered?

**There were little horizontal movement in the low horizontal advection areas chosen, and our strategy was to sample along the flow, thus minimizing horizontal advection fluxes. Note that fluxes need gradients of properties and they are really low horizontally in open ocean areas, particularly as station locations were specifically chosen in low horizontal current areas.**

Are there processes occurring within (or beyond) the boundaries of this "system" that could confound the approach?

**As mentioned, vertical movements of zooplankton may probably play a significant role but are difficult to quantify.**

P3, Lines 9-16 — The authors should provide more information on the trophic gradient and how 'oligotrophic' and 'ultra-oligotrophic' are defined. What are the physical factors causing the gradient?

**The paragraph: "The WTSP Ocean has recently been identified as a hot spot of $N_2$ fixation, harbouring $N_2$ fixation rates >500 µmol N m$^{-2}$ d$^{-1}$ (Bonnet et al., 2017). The region covered by the OUTPACE cruise is characterized by trophic and $N_2$ fixation gradients (Moutin et al., 2017), with oligotrophic waters characterized by high $N_2$ fixation rates (631 ± 286 µmol N m$^{-2}$ d$^{-1}$) mainly associated with Trichodesmium in the western part (i.e. within the hot spot around Melanesian archipelago waters, hereafter named MA), and ultra-oligotrophic waters characterized by low $N_2$ fixation rates (85 ± 79 µmol N m$^{-2}$ d$^{-1}$) mainly associated with UCYN in the eastern part (South Pacific gyre, hereafter named SPG) (Bonnet et al., this issue; Stenegren et al., this issue)." page 3 line**

32 has been rewritten as follows: "The WTSP Ocean has recently been identified as a hotspot of $N_2$ fixation, including $N_2$ fixation rates >500 µmol N m$^{-2}$ d$^{-1}$ (Bonnet et al., 2017). The region covered by the OUTPACE cruise encompasses contrasting trophic regimes characterized by strong differences in top nitracline depths, from 46 to 141 m (Moutin et al., this issue), and representing a large part of the oligotrophic gradient at the scale of the world Ocean (Moutin and Prieur, 2012; their Fig. 9). The westward oligotrophic waters are characterized by high $N_2$ fixation rates (631 ± 286 µmol N m$^{-2}$ d$^{-1}$) mainly associated with Trichodesmium (i.e. within the hotspot around the Melanesian archipelago waters, hereafter named MA), and the eastward ultra-oligotrophic waters (in the eastern boarder of the South Pacific gyre, hereafter named SPG waters) are characterized by low $N_2$ fixation rates (85 ± 79 µmol N m$^{-2}$ d$^{-1}$) mainly associated with UCYN (Bonnet et al., this issue; Stenegren et al., this issue). This west to east $N_2$ fixation gradient has been mainly attributed to a decrease of iron availability in SPG waters as compared to MA waters (Bonnet et al., this issue; Guieu et al., in rev.)."

The strong ultra-oligotrophy of the South Pacific gyre has previously been described in the BIOSOPE special issue (https://www.biogeosciences.net/special_issue19.html).

P3, Lines 17-19 — The points of focus are great, but were there hypotheses to be tested? Why was it important to focus the study on these three points? What information/data gaps were being filled by conducting the study?

The sentence is: "In the present study we focus on (i) the contribution of $N_2$ fixation to new N inputs in the WTSP during the summer season; (ii) the coupling between $N_2$ fixation and export; and (iii) the equilibrium versus disequilibrium between $N_2$ fixation and particulate N export in the WSTP." We consider that the plan of our paper has to be placed at the end of the Introduction. It is important to focus the study on these three points because as stated by Reviewer #4, the 3 'points of focus are great'. The hypotheses we tested were "Is nitrogen fixation a predominant flux in the photic layer of the WTSP?", and "Are we able to link specificity in export with the dominant diazotroph groups?"

We provided the first N budgets following a Lagrangian strategy allowing confirmation of the predominant role of nitrogen fixation in the WTSP, as well as a large dataset of new data from the poorly studied South Pacific Ocean (http://www.obs-vlfr.fr/proof/php/outpace/outpace.php).

**Methods**

P3, Line 27 - P4, Line 14 — There are not enough details on the 3 criteria for site selection. What were the parameters of "local minima of surface current intensity" used to determine if conditions were suitable? How much surface current was considered acceptable? Were deeper currents considered? How was trophic status defined? In terms of chlorophyll or something else? If so, what were the thresholds used for oligotrophic, ultra-oligotrophic, etc.?

We were looking for local minima of surface current intensity in order to find adequate locations for our Lagrangian strategy, and we found them. The oligotrophic gradient was necessarily sampled when we crossed 4000 km of the WTSP from the Melanesian archipelagoes to the South Pacific gyre. We understand the reviewer's comment about the choice of sampling sites and we apologize for this. The question is so important that a

specific paper was devoted to that question by Alain de Verneil et al. (this issue), and unfortunately, it was not available at the time of Reviewer #4's response. The paper is now available here: https://www.biogeosciences-discuss.net/bg-2017-455/).

P5, Line 3—Was the chlorophyll fluorescence sensor calibrated to simultaneous samples analyzed for chlorophyll using more conventional extraction techniques?

To answer Reviewer #4's question, we have added the following sentence page 5 line 28 in the text "The chlorophyll fluorescence sensor was calibrated prior to the cruise and post-calibration was conducted using all HPLC measurements undertaken during the cruise.". Unfortunately, the relationship was not good, indicating that we sampled really different communities along the 4000 km transect. Nevertheless, considering the poor relationship between chl a and biomass, particularly with depth in the South Pacific Ocean (Duhamel et al., 2007), we considered that our pre-calibrated sensor was good enough to represent the oligotrophic gradient sampled.

Duhamel, S., T. Moutin, F. Van Wambeke, B. Van Mooy, P. Rimmelin, P. Raimbault, and H. Claustre. Biogeosciences, 4, 941-956, https://doi.org/10.5194/bg-4-941-2007, 2007

P5, Lines 4-7—Were nutrient samples analyzed immediately, or filtered and stored for analysis later (if so, provide details on procedures used), or not filtered at all...? Why wasn't ammonium included in the nutrient measurements?

Samples were filtered and analyzed both immediately and in the laboratory after poisoning. We have modified the text page 5 line 30 in the following way: "Phosphate ($PO_4^{3-}$) and $NO_3^-$ concentrations were measured daily at 12 depths from the surface to 200 m on each nutrient CTD cast using standard colorimetric procedures (Aminot and Kérouel, 2007) on a AA3 AutoAnalyzer (Seal-Analytical). After filtration, one sample was directly analyzed on-board and the other poisoned with 50µl HgCl2 (20 g $L^{-1}$) and stored for analysis after the cruise in the laboratory. The quantification limits were 0.05 µmol $L^{-1}$ for $PO_4^{3-}$ and $NO_3^-$."

Ammonium was measured on board and data are available but we focus here on new N budgets in the photic layer and were not interested in regeneration.

P5, Lines 9-17 — It is unclear where the "associated N uptake" part of this section is evaluated. More details are needed describing sample handling and analyses for the PP incubations.

The "associated N uptake" corresponds to the N-derived PP (N-PP) mentioned in the paper. To be clearer, we have rewritten the sentence page 6 line 8 as follows: "A N-derived PP (N-PP), i.e. the associated N uptake, was obtained…"

Further details have been added in the final version concerning PP incubation methods page 6 line 4: "PP was measured in triplicate using the $^{14}$C tracer method (Moutin and Raimbault, 2002). Samples were incubated in 320 mL polycarbonate bottles on the in situ drifting production line…"

P5, Lines 19-22—More details are needed on the aerosols sampling, especially since the reference given for the method is only a submitted paper. Is there a reason why ammonia was not included in the atmospheric deposition measurements?

We completely agree with this remark. We have thus completed this section in collaboration with Cécile Guieu, who performed the measurements. Thus, Cécile Guieu will be added as co-author of the paper in its final version.

The new version of the section is:

"N atmospheric deposition ($NO_3^-$ and $NO_2^-$ (nitrite), hereafter called NOx) was quantified along the transect after dissolution of aerosols collected continuously during the transect, as described in Guieu et al. (in rev.). Briefly, the sampling device, designed to avoid ship contamination, was installed at the look-out post in the front of the ship, collected aerosols at ~20 L min$^{-1}$ onto polycarbonate, 47-mm diameter, 0.45-µm porosity (previously acid-cleaned with a 2% solution of HCl (Merck, Ultrapur, Germany) and thoroughly rinsed with ultra-pure water and dried under a laminar flow bench and stored in acid-cleaned Petri dishes). Dissolution experiments to determine NOx released in surface seawater after deposition were performed on board using acid-cleaned Sartorius filtration units (volume 0.250 L) and filtered surface (5 m) seawater. Each sample was subjected to two contact times: the first contact was at one minute, and the second contact was at 24 hours. NOx was analyzed using 1-m long Liquid Waveguide Capillary Cells (LWCC) made of quartz capillary tubing, following the protocol described in Louis et al., 2015. An extrapolated NOx from dry deposition was estimated on the basis of a deposition velocity of submicronic particles (0.4 m s$^{-1}$; Vong et al., 2010)."

P5, Line 29 – P6, Line 3 — Very confusing sentence. Rewrite for clarity.

The sentence has been rewritten page 7 line 11 as follows: "It has been previously shown that the bubble method was potentially underestimating $N_2$ fixation rates (Großkopf et al., 2012; Mohr et al., 2010) compared to methods consisting in adding the $^{15}N_2$ as dissolved in a subset of seawater previously $N_2$ degassed (Mohr et al., 2010). This underestimation is due to incomplete equilibration of the $^{15}N_2$ gas with surrounding seawater. However, other studies did not find any significant difference between the two methods (Bonnet et al., 2016b; Shiozaki et al., 2015)."

P6, Lines 7-10 — How were dissolved gas samples transferred from the bottles to Exetainers? Kana et al. (1994) does not cover 15N2 measurements/analyses using MIMS. Is there another citation for the 15N2 analyses using MIMS?

The dissolved gas analyzed was contained in the seawater, thus 12 mL of seawater was rapidly sub-sampled with an eyedropper from the bottles to the Exetainers to avoid contamination. In their study, Kana et al. (1994) measured dissolved $N_2$, $O_2$ and Ar in water using MIMS. For that purpose, they detected masses 28, 32 and 40, corresponding to $N_2$, $O_2$ and Ar, respectively. In our study, we detected masses 28 and 30 corresponding to $^{14}N_2$ and $^{15}N_2$ respectively.

P7, Lines 1-3 — perhaps add "and" before daily? Something is missing in this sentence.

The sentence has been rewritten page 8 line 18 as follows: "The time series of the $NO_3^-$ diffusive flux was calculated using an hourly temporal interpolation of Kz over the entire duration of each LD station. Also, daily averages and 5-day averages were computed."

P7, Lines 12-15 — Define "swimmers". PP was previously defined as primary production, so also using it for particulate phosphorus is confusing.

**Swimmers correspond to zooplankton, as we considered in a first approximation that all zooplankton were alive. We delete the particulate phosphorus data, outside of the scope of the manuscript, and therefore the confusion disappears.**

**Results & Discussion**

P9, Lines 24-27—What was the integration depth used for these rates? It is odd to see areal rates reported for a depth-integration that apparently does not include sediments.

**The rates were integrated from the surface to 20 m below the last measurement (considered as nil) using the trapezoidal method for integration according to the classical JGOFS protocol. This allows us to be consistent within the three LD stations which presented different photic layer depth.**

P9, Line 30—Is there really a strong contrast between LD A and the other two stations given the very large variability around the mean at LD A (24.4 ± 24.4)?

**The mean $NO_3^-$ flux at LD A was around 3.5 and 5 times higher than at LD B and LD C, respectively, that is it is clearly contrasted. The wide variability around the mean is characteristic of the pulse that was observed at the end of the LD A survey, and which strongly increased the mean Kz at LD A. The differences in estimated fluxes between the stations is mainly attributable to differences in Kz between the stations. The gradients of concentration varied little between stations (table 1). The turbulent diffusion contrast is a robust result despite the variability of the signal (Kz and therefore the flux). This is explained by the fact that shear instability, the dominant process triggering turbulence at long stations, is more intense in LDA because of higher shear, by a factor of about 3. Thus, the strong contrast between LD A and the other two stations is confirmed.**

P11, Line 11 — Perhaps the authors should use LD-A, LD-B, and LD-C to denote their stations, instead of LD A, LD B, and LD C. There have been a few cases like here (LDA) where the site abbreviations have not been consistent.

**The references LD A, LD B and LD C have been kept in the interests of consistency with all the papers of the OUTPACE special issue.**

P12, Lines 3-23 — I found this narrative confusing. Perhaps the authors could streamline this text to focus it on the most important points?

**In accordance with this remark, we have 'streamlined' the text has much as possible. However, much important information is mentioned in this section that it is not possible to delete. We explain that we observed an unreported high contribution of N2 fixation in this region compared to low atmospheric deposition and vertical turbulent diffusion. In addition, we discuss both of those inputs in a general context, and therefore think this discussion is important and has its place in this section.**

**The text has been rewritten page 14 line 3 to page 15 line 8 as follows:**

- " Extrapolated NOx deposition from the atmosphere during OUTPACE (range: 0.34 – 1.05 µmol m$^{-2}$ d$^{-1}$) were one order of magnitude lower than predicted with major uncertainties by global models that include wet and gas deposition for that region (Kanakidou et al., 2012). Our flux could be an underestimation as it represents only dry deposition and as gas and organic forms were not measured. At the global scale and depending on the location, organic nitrogen could represent up to 90 % of N atmospheric deposition (Kanakidou et al., 2012), and $NH_4^+$ could account for ~40% (Dentener et al. 2006). Even if we double our estimated deposition flux, atmospheric deposition still remained low (< 1.5 %) and consequently represented a minor contribution of the new N input (Table 2). This negligible contribution of atmospheric input to the overall N budget (less than 1.5 %) therefore implies an important contribution of other terms, such as $N_2$ fixation.

- Then, $NO_3^-$ input by vertical turbulent diffusion appeared as the second source (1 to 8 %) of new N at the three stations. This contribution was lower than in previous studies in other oligotrophic regions (Table 5), where $NO_3^-$ input by vertical turbulent diffusion contributes ~ 18 % of new N in the Indian South Subtropical Gyre (Fernández-Castro et al., 2015), and ~ 50 % in the Tropical North Atlantic (Capone et al., 2005). In most studies (Fernández-Castro et al., 2015; Moutin and Prieur, 2012; Painter et al., 2013), an average Kz value is used (i.e. averaged over the cruise, over a station or over depth) to determine $NO_3^-$ input by turbulence in the photic layer. In this study we performed high frequency direct measurements of Kz and highlighted the importance of turbulent event pulses on diffusive $NO_3^-$ input. Using a constant Kz of 10$^{-5}$ m$^2$ s$^{-1}$ at the 3 stations decreases the $NO_3^-$ input down to 22.9 µmol N m$^{-2}$ d$^{-1}$ at LD A and increases $NO_3^-$ input up to 19.9 and 25.5 µmol N m$^{-2}$ d$^{-1}$ at LD B and LD C, that is 2.7 and 4.8 times higher than using a high frequency Kz for the latter two stations. The contrasted $NO_3^-$ input observed at the three stations results from the high variability in turbulence along the west-east transects (Bouruet-Aubertot et al., this issue). Thus, using a constant Kz removes the contrasted $NO_3^-$ input between the 3 stations (~ 4 times higher at LD A than at LDB and LD C). Consequently, using average Kz values for the turbulent diffusive flux computation can lead to significant bias. In our study, $NO_3^-$ input was calculated at the top of the nitracline. Painter et al. (2013) have demonstrated the variability that may be introduced into the estimated $NO_3^-$ input by the depth of the defined nitracline. With a constant Kz in the 2 cases, they estimated that $NO_3^-$ input was 5 times lower at the top of nitracline depth than at the maximum gradient depth. In our study, the $NO_3^-$ input would also be ~ 3-4 times higher if calculated at the maximum gradient depth rather than at the top nitracline, mainly due to the increase of the nitracline gradient up to 48

µmol N m$^{-4}$ (Table 3). However, in all cases, the NO$_3^-$ input by turbulence always represented a minor contribution to the N budget.

- Finally, the high contribution of N$_2$ fixation to new N input in the photic layer results from the intrinsically high N$_2$ fixation rates we measured in the WTSP (especially in MA waters), that are part of the hotspot of N$_2$ fixation reported by Bonnet et al., (2017), with rates being in the upper range of rates reported in the global N$_2$ fixation Marine Ecosystem Data (MAREDAT) database (Luo et al., 2012). Those high N$_2$ fixation rates are as high as westward in the Salomon Sea (Berthelot et al., 2017; Bonnet et al., 2015), extending the hotspot of N$_2$ fixation to the whole of the WTSP (Bonnet et al., 2017)."

P13, Line 13 — (and elsewhere) primary production or particulate phosphorus?

In this sentence, PP is used for primary production. We agree with this remark, it should be clearer as we do not present particulate phosphorus data anymore.

P13, Lines 15-16 — correct these scientific notations for gene copies

"nifH copies L$^{-1}$" has been changed to "nifH gene copies L$^{-1}$" in the new version

P13, Lines 20-21 — Why use the areal rates here instead of the volumetric rates?

We used the areal rates to be consistent with the units of all the N inputs in the photic layer and N export, as reported in Table 4.

P13, Line 32 – P14, Line 2 — Awkward sentence. Rewrite for clarity.

It has been rewritten page 16 line 15 as follows: "The export efficiency of UCYN-B (2.3 % on average) and het-1 (4.0 % on average) was higher than that on of *Trichodesmium*, which is consistent with Bonnet et al., (2016b) and Karl et al., (2012). In a mesocosm experiment performed in the coastal waters of New Caledonia, Bonnet et al., (2016b) revealed that UCYN-C were efficiently exported thanks to aggregation processes."

P14, Lines 7-14 — Confusing text. Rewrite for clarity.

It has been rewritten page 16 line 26 as follows: "To date, few qPCR data on nifH from sediment traps are available (Karl et al., 2012) to compare with our study. However, it must be noted that we measured the highest export and e-ratio at LD A, where Trichodesmium dominated the diazotroph community. This suggests that most of the export was probably indirect, i.e. after the transfer of diazotroph-derived N (DDN) to the surrounding bacterial, phytoplankton and zooplankton communities, as revealed by Caffin et al., (this issue) during the same cruise, that are subsequently exported."

P14, Line 22 — What is "PCD"?

Programmed cell death (PCD) has been defined in the new version of the manuscript page 17 line 8 in the following way: "Programmed cell death (PCD) was detected at LD B (Spungin et al., this issue),..."

P16, Lines 12-26—The conclusion section is a little flat. The authors could do a better job of placing their study into a better context in terms of the global N budget and C export in the oceans.

We understand reviewer #4's comment. However, we have opened up the discussion of the fact that the oligotrophic ocean, which represents 60 % of the global ocean surface, could play a more significant role in C export than initially considered. We think that it is not consistent to perform larger C and N budget, at higher spatial scale, with the dataset that we have mentioned here at only 3 stations during one season of the year. Thus, to fill information and data gaps relative to our study and to enhance understanding of the global ocean, we think that a time-series study should be established in this region of the world ocean. This would give us the 'big picture' of the role of N2 fixation in export, as Bonnet et al. (2017) have shown that the WTSP is a hotspot of N2 fixation, and we have mentioned the important role of this process with regard to export, that is a major issue in oceanography today. In addition, this would complete the present time-series HOT, BATS and DYFAMED that were established in the North Pacific, North Atlantic Oceans and Mediterranean Sea, respectively.

Thus we have added new sentences at the end of the conclusion page 19 line 31 in the following way: "Finally, as Bonnet et al. (2017) have recently shown that the WTSP is a hot spot of $N_2$ fixation, and as we have shown the importance of this process on the N input and N and C export, we suggest that this region of the world ocean should be further investigated by oceanographic cruise and time-series establishment. This would give us a "big picture" of the role of $N_2$ fixation on the export in the oligotrophic ocean."

Table 1 — Are these Kz values supposed to be in scientific notation? Are the units for the nitracline correct?

This has been corrected to "$1.11 \times 10^{-5} \pm 1.00 \times 10^{-5}$" in the new version. We confirm that the units for the nitracline are correct.

**Technical Corrections**

P1, Line 24 — add "respectively" after "LD B"

This has been corrected in the new version

P1, Line 28 — add a comma after "LD C"

This has been corrected in the new version

P1, Line 31 — PC and PP not defined (or PN earlier)

This has been corrected in the new version

P1, Line 34 — change "there" to "their"

This has been corrected in the new version

P2, Line 5 — add comma after "ammonia"

It has been corrected in the new version

P2, Line 7 — "before" is an awkward word choice

P2, Lines 7-11 — Long, run on sentence. Rewrite for clarity.

**This has been rewritten page 2 line 25 as follows: "In the oligotrophic ocean, N availability often limits phytoplankton growth (e.g. Moore et al., 2013) and $N_2$ fixation sustains a significant part of new primary production (PP, i.e. the production unrelated to internal recycling of organic matter in the photic layer) such as in the North (Karl et al., 1997) and South Pacific Ocean (Moutin et al., 2008), the western Mediterranean Sea (Garcia et al., 2006), or the tropical North Atlantic (Capone et al., 2005)."**

P2, Line 21 — add comma after "...et al., 2008)"

**This has been corrected in the new version**

P2, Line 24 — add comma after "large"

**This has been corrected in the new version**

P2, Line 25 — add comma after "phytoplankton"

**This has been corrected in the new version**

P2, Line 29 — add comma after "ocean"

**This has been corrected in the new version**

P3, Line 9 — "harbouring" is an awkward word choice

**This has been replaced by "including" in the new version**

P3, Lines 10-15 — Long, run on sentence. Rewrite for clarity.

**This has been rewritten page 3 line 34 as follows: "The region covered by the OUTPACE cruise is characterized by trophic and $N_2$ fixation gradients (Moutin et al., 2017). The westward oligotrophic waters are characterized by high $N_2$ fixation rates (631 ± 286 μmol N $m^{-2}$ $d^{-1}$) mainly associated with *Trichodesmium* (i.e. within the hotspot around Melanesian archipelago waters, hereafter named MA), and the eastward ultra-oligotrophic waters (in the eastern border of the South Pacific gyre, hereafter named SPG waters) are characterized by low $N_2$ fixation rates (85 ± 79 μmol N $m^{-2}$ $d^{-1}$) mainly associated with UCYN (Bonnet et al., this issue; Stenegren et al., this issue).**

P3, Line 17 — add comma after "study"

**This has been corrected in the new version**

P4, Line 22 — "every day"

**This has been corrected in the new version**

P7, Line 13 — "weighed"

**This has been corrected in the new version**

P9, Line 13 — add "from" after "ranged"

**It has been corrected in the new version**

P10, Line 32 — "2.67 x 104"

**This has been corrected in the new version**

P11, Line 2 — change "from" to "for"

**This has been corrected in the new version**

The paper requires a thorough editing for grammar, word choice, and punctuation.

**The new version of the manuscript has been reviewed and corrected by an English native speaker.**

**References added**

Gradoville, M. R., D. Bombar, B. C. Crump, R. M. Letelier, J. P. Zehr and A. E. White Diversity and activity of nitrogen fixing communities across ocean basins. Limnol. Oceanogr. 62: 1895-1909, 2017

[revised manuscript text omitted]
 | $1.11 \times 10^{-5} \pm 1.00 \times 10^{-5}$ | $23 \pm 13$ | $24.4 \pm 24.4$ |
| LD B | $3.59 \times 10^{-6} \pm 3.11 \times 10^{-6}$ | $21 \pm 12$ | $6.7 \pm 5.3$ |
| LD C | $2.04 \times 10^{-6} \pm 1.11 \times 10^{-6}$ | $27 \pm 13$ | $4.8 \pm 2.2$ |
| *Maximum gradient* | | | |
| LD A | $1.69 \times 10^{-5} \pm 1.15 \times 10^{-5}$ | $53 \pm 10$ | $79 \pm 56$ |
| LD B | $4.52 \times 10^{-6} \pm 3.22 \times 10^{-6}$ | $48 \pm 6$ | $19 \pm 14$ |
| LD C | $2.96 \times 10^{-6} \pm 1.84 \times 10^{-6}$ | $48 \pm 14$ | $21 \pm 11$ |

**Table 4: Zooplankton sediment traps data at the three LD stations. Depth of sampling, mean dry weight (DW) zooplankton recovered in the traps, C and N associated to zooplankton.**

| Station | depth m | Zooplankton DW (Swimmers) $mg\ m^{-2}\ d^{-1}$ | Zoo-C $mg\ C\ m^{-2}\ d^{-1}$ | Zoo-N $mg\ N\ m^{-2}\ d^{-1}$ |   | C:N ratio 106:x |
|---|---|---|---|---|---|---|
| | 150 | $82.1 \pm 18.0$ | $42.3 \pm 7.6$ | $8.2 \pm 1.2$ |  | 106:18 |
| LD A | 330 | $376.1 \pm 26.1$ | $129.2 \pm 15.5$ | $19.5 \pm 4.0$ |  | 106:14 |
| | 520 | $14.1 \pm 8.0$ | $5.3 \pm 2.2$ | $1.3 \pm 0.4$ |  | 106:22 |
| | 150 | $112.9 \pm 38.1$ | $57.1 \pm 21.7$ | $11.3 \pm 4.7$ |  | 106:18 |
| LD B | 330 | $62.3 \pm 31.6$ | $27.3 \pm 15.5$ | $4.9 \pm 2.8$ |  | 106:16 |
| | 520 | $10.5 \pm 3.0$ | $4.9 \pm 1.2$ | $1.1 \pm 0.3$ |  | 106:20 |
| | 150 | $121.3 \pm 37.5$ | $41.7 \pm 14.8$ | $10.3 \pm 5.0$ |  | 106:22 |
| LD C | 330 | $31.0 \pm 5.1$ | $14.3 \pm 5.0$ | $2.9 \pm 1.3$ |  | 106:18 |
| | 520 | - | - | - | - | |

**Table 5: Contribution of N$_2$ fixation and NO$_3^-$ vertical diffusion to new N inputs in oligotrophic region.**

| Location | Contribution to new N | | Source |
|---|---|---|---|
| | N$_2$ fixation | NO$_3^-$ diffusion | |
| Tropical North Atlantic | 50 % | 50 % | Capone et al., 2005 |
| Subtropical North Atlantic | 2 % | - | Mourino-Carballido et al., 2011 |
| Subtropical South Atlantic | 44 % | - | Mourino-Carballido et al., 2011 |
| South Atlantic Gyre | 21 % | 24 % | Fernández-Castro et al., 2015 |
| Indian South Subtropical Gyre | 12 % | 18 % | Fernández-Castro et al., 2015 |
| Mediterranean Sea | 0 – 32 % | 21 – 53 % | Moutin and Prieur, 2012 |
| | | | Bonnet et al., 2011 |
| North Pacific subtropical Gyre | 30 – 50 % | - | Karl et al., 2003 |
| North Pacific subtropical Gyre | 48 % | 52 % | Dore et al., 2002 |
| Western Tropical South Pacific | 92 – 99 % | 1 – 8 % | This study |

**Table 6: Contribution of N export by active zooplankton migration to total PN export.**

| Location | % of PN export | Source |
|---|---|---|
| Subtropical and tropical Atlantic | 7.6 | Longhurst et al., 1989, 1990 |
| North Atlantic BATS | 37.3 | Dam et al., 1995 |
| Equatorial Pacific | 4.9 | Le Borgne and Rodier, 1997 |
| Equatorial Pacific | 19.8 | Le Borgne and Rodier, 1997 |
| North Atlantic BATS | 19.9 | Al-Mutairi and Landry, 2001 |
| North Pacific subtropical Gyre | 38 | Hannides et al., 2009 |
| California Current Ecosystem | 20 | Stukel et al., 2013 |
| Cost Rica Dome | 38 | Stukel et al., 2015 |

**Figure captions**

**Figure 1: Position of the long duration stations sampled in this study (OUTPACE cruise): LD A in green, LD B in red and LD C in blue on a quasi-Lagrangian surface Chl *a* concentrations map. The in situ production lines were deployed in the photic layer (from 5 to 105, 80 and 180 m for LD A, LD B and LD C, respectively) and the PPS5 sediment traps were deployed at 150, 330 and 520 m.**

**Figure 2: Vertical profiles of net $N_2$ fixation rates (nmol N $L^{-1}$ $d^{-1}$) estimated using in situ incubations at day 1 (in situ 1: red circles), day 3 (in situ 2 : orange circles) and day 5 (in situ 3 :purple) and using on-deck incubations (purple filled area) at stations LD A (left), LD B (middle) and LD C (right). The $NO_3^-$ concentrations averaged over the 5 days of station occupation are also reported (light blue squares : $\mu$mol $L^{-1}$), as well as $PO_4^{3-}$ concentrations (dark blue squares: $\mu$mol $L^{-1}$), and fluorescence/chlorophyll (green dots : $\mu$g $L^{-1}$)**

**Figure 3: Temporal evolution of PAR, DCM (green line), $\rho_{NO3}$ (red line), and 1 % of surface PAR (yellow dots and crosses) during the three stations' occupation period (LD A: top panel, LD B: middle panel, LD C: bottom panel).**

**Figure 4: Temporal evolution of upward vertical $NO_3^-$ flux ($\mu$mol N $m^{-2}$ $d^{-1}$) calculated at the top of the nitracline for each station (LD A: top panel, LD B: middle panel and LD C: bottom panel) after temporal interpolation (blue). Daily mean from noon to noon in dashed orange line and occupation period mean in green line.**

**Figure 5: Relative contribution of each diazotroph (*Trichodesmium* in red, UCYN-B in orange and Het-1 in yellow) to the total PC associated with diazotrophs (diazotroph-PC) in the sediment traps at 150 m (top), 330 m (middle) and 520 m (bottom), at the three stations LD A (left panel), LD B (middle panel) and LD C (right panel). Values in blue correspond to the contribution of diazotroph-PC to total PC measured in the traps. No *Trichodesmium* valid data available at LD A 150 m.**

[Figure]

**Figure 1**

[Figure]

**Figure 2**

[Figure]

**Figure 3**

[Figure]

**Figure 4**

[Figure]

5    **Figure 5**